# METADATA MATTERS FOR TIME SERIES: INFORMATIVE FORECASTING WITH TRANSFORMERS

## ABSTRACT

Time series forecasting is prevalent in extensive real-world applications, such as financial analysis and energy planning. Previous studies primarily focus on time series modality, endeavoring to capture the intricate variations and dependencies inherent in time series. Beyond numerical time series data, we notice that metadata (e.g. dataset and variate descriptions) also carries valuable information essential for forecasting, which can be used to identify the application scenario and provide more interpretable knowledge than digit sequences. Inspired by this observation, we propose a **Meta**data-informed **T**ime **S**eries **T**ransformer (MetaTST), which incorporates multiple levels of context-specific metadata into Transformer forecasting models to enable informative time series forecasting. To tackle the unstructured nature of metadata, MetaTST formalizes them into natural languages by pre-designed templates and leverages large language models (LLMs) to encode these texts into metadata tokens as a supplement to classic series tokens, resulting in an informative embedding. Further, a Transformer encoder is employed to communicate series and metadata tokens, which can extend series representations by metadata information for more accurate forecasting. This design also allows the model to adaptively learn context-specific patterns across various scenarios, which is particularly effective in handling large-scale, diverse-scenario forecasting tasks. Experimentally, MetaTST achieves state-of-the-art compared to advanced time series models and LLM-based methods on widely acknowledged short- and long-term forecasting benchmarks, covering both single-dataset individual and multi-dataset joint training settings.

## 1 INTRODUCTION

Time series forecasting is of increasing demand in real-world scenarios encompassing diverse domains, including energy, transportation, and meteorology (Weron, 2014; Lv et al., 2014; Wu et al., 2021; Wang et al., 2024b). Motivated by the substantial practical value, deep time series models have been widely explored and achieved significant advancements, where diverse techniques are developed to capture temporal variations from historical observations for future prediction (Salinas et al., 2020; Nie et al., 2023; Liu et al., 2024b; Dong et al., 2024). Despite the success in uncovering intricate temporal patterns, relying solely on the sequence of observation values can be insufficient to guarantee accurate forecasting. Taking the example of traffic forecasting, two crossroads may exhibit similar patterns in the morning peak but will present disparate future trends due to the closing times of nearby companies. Although there may exist some slightest clues in observations, it requires the model to identify very subtle differences of the past or consider a sufficiently long period for identification, bringing challenges in model capacity or efficiency. More direct and evident information is expected.

In the spirit of informing the forecasting model of a more direct context, we notice metadata, which is referred to as "descriptions about data", holds significant value in time series analysis. In time series databases, metadata records information such as data source details and statistical summaries, which is crucial in facilitating efficient data organization and enhancing query efficiency. Beyond data management, descriptive metainformation enriches the context in time series forecasting, providing a more comprehensive understanding of the scenario and enabling accurate predictions. Notably, metadata is usually unstructured since it contains information on the data from heterogeneous views. Therefore, despite the potential benefits, incorporating metadata into prediction remains unexplored.

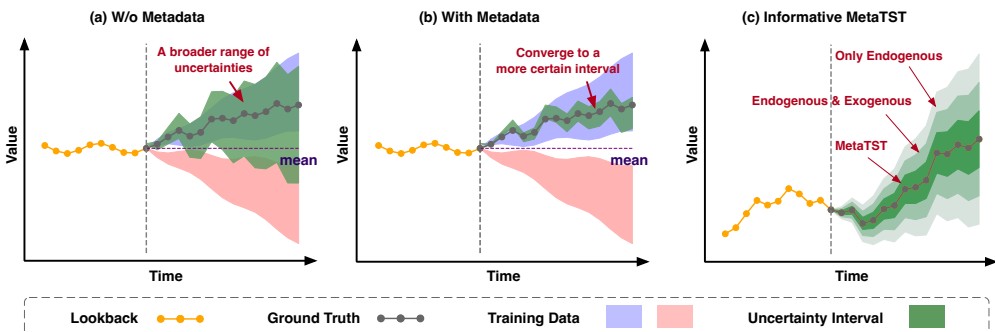

Figure 1: A conceptual illustration for different forecasting paradigms. (a) Canonical time series forecasting without metadata. (b) Metadata-informed time series forecasting and (c) MetaTST utilizes more informative inputs, especially context-specific metadata, to achieve highly certain forecasts.

After comprehensively analyzing the factors that influence the predictability of time series, we summarize three key elements crucial for accurate forecasting: (1) accurately capturing the intrinsic temporal variations of the target time series (*Endogenous series*); (2) fully understanding the external factors influencing the target series (*Exogenous series*); and (3) properly introducing reasonable and context-specific information of the forecasting scenario (*Metadata*). However, most contemporary researches solely focus on developing models to learn intrinsic temporal dependencies, (Zhou et al., 2021; Nie et al., 2023), and only a limited branch of works have emphasized the incorporation of exogenous factors (Lim et al., 2021; Olivares et al., 2023). As illustrated in Figure 1, the absence of metadata regarding the forecasting scenario causes models to consider a broader range of uncertain future possibilities, hindering them from generating reliable predictions. Furthermore, without detailed information on the prediction scenarios, models may become perplexed when confronted with similar temporal patterns, resulting in an increased uncertainty range of predictions. Recalling the aforementioned traffic forecasting example, the flow is intricately related to factors such as holidays, control policies, and even sensor locations. By incorporating these informative metadata, models can better distinguish between distinct scenarios and achieve more precise and certain forecasts.

Inspired by the above insights, we propose a **Meta**data-informed **T**ime **S**eries forecasting method with **T**ransformers (MetaTST). To handle the unstructured nature of metadata, MetaTST incorporates metadata by describing them in well-formalized natural languages from three different levels of granularity, including dataset, task, and sample aspects, which provides a multifaceted view of the data, enabling more informed predictions. Unlike previous LLM4TS works that utilize pre-trained large language models (LLMs) through fine-tuning model parameters and aligning representations (Zhou et al., 2023; Jin et al., 2024), MetaTST leverages well-trained LLMs as the fixed metadata encoder to maintain their original understanding capability. By combining metadata tokens encoded from texts with patch-wise endogenous and series-wise exogenous series tokens, MetaTST significantly enriches the representation learning of endogenous series, resulting in more informative and reliable predictions. Besides, MetaTST demonstrates its adaptability to diverse forecasting scenarios by learning and distinguishing context-specific patterns, which allows MetaTST to handle large-scale, diverse-scenario forecasting tasks, posing a potential solution for time series foundation models. Equipped with informative metadata, MetaTST consistently achieves state-of-the-art performance on both short- and long-term time series forecasting tasks, covering single-dataset-individual and multi-dataset-joint training settings. Our contributions are summarized as follows:

- Rethinking the key factors that drive accurate time series forecasting, we propose MetaTST, an informative time series forecasting method with Transformers that incorporates multi-level metadata to enhance series representations for more accurate time series forecasting.

- Unlike previous usage of LLMs, MetaTST proposes to integrate them as the fixed metadata encoder. This design can fully utilize LLMs' original semantic understanding capability to better capture context-specific forecasting preferences of diverse scenarios.

- MetaTST consistently achieves state-of-the-art performance on extensive real-world time series forecasting tasks, encompassing both single-dataset individual and multi-dataset joint training settings on twelve well-established short- and long-term benchmarks.

## 2 RELATED WORK

**Native Time Series Models**    In recent years, deep models have been widely studied for time series analysis, particularly for forecasting (Wang et al., 2024b). Diverse architectures are proposed to capture temporal variations in time series, including CNN-based (Liu et al., 2022; Wu et al., 2023) and RNN-based models (Zhao et al., 2017; Lai et al., 2018; Wang et al., 2019). However, these models often struggle with limited receptive fields, making it challenging to capture long-term dependencies. Besides, several MLP-based forecasters with temporally fully connected layers (Oreshkin et al., 2020; Das et al., 2023; Wang et al., 2024a) have demonstrated remarkable performance, but they fall short in model capacity, which may degenerate in handling diverse and complex data (Wu et al., 2023). As a milestone of foundation backbones, Transformers have also been extensively explored in time series to capture long-term dependencies and unearth complex intricate temporal patterns (Wen et al., 2022). The classic usage is to apply the attention mechanism or its variants along the time dimension to uncover temporal variations (Zhou et al., 2021; Wu et al., 2021). Subsequently, PatchTST (Nie et al., 2023) proposes to capture temporal dependencies among series patches and achieves notable performance. In addition, some research has adapted the attention mechanism to capture the multivariate correlations (Zhang & Yan, 2022). iTransformer (Liu et al., 2024b) inverts the conventional duties of the attention mechanism and the feed-forward network by encoding the entire time series to one variate token. Furthermore, TimeXer (Wang et al., 2024c) leverages different levels of representation to capture temporal and variate dependencies simultaneously. However, none of these methods consider incorporating metadata, which is a foundation insight of MetaTST.

Motivated by recent advances in large models, large-scale pre-trained time series models have gained increasing interest (Das et al., 2024; Liu et al., 2024d; Cao et al., 2024; Woo et al., 2024; Gao et al., 2024). Existing works (Liu et al., 2024a; Das et al., 2024) have constructed large-scale datasets collected from diverse domains, encapsulating as many varied temporal patterns as possible. However, these models are predominantly limited to the time series modality and overlook the essential metadata information. This limitation hampers the ability of the consequent models to discern the differences in temporal patterns across various domains, which can be well addressed in MetaTST.

Table 1: Related work comparison. "TS" is short for time series. "/" refers to without LLMs.

| Methods | **MetaTST** (**Ours**) | TimeLLM (2024) | GPT4TS (2023) | TimeXer (2024c) | iTransformer (2024b) | PatchTST (2023) | DLinear (2023) |
|---|---|---|---|---|---|---|---|
| Input Modality | TS + Language | TS + Language | TS + Language | TS | TS | TS | TS |
| LLMs Usage | Encoder | Backbone | Backbone | / | / | / | / |

**Large Language Models for Time Series**    With the rapid advancement of large language models (LLMs) (Devlin et al., 2018; Radford et al., 2019; Gao et al., 2020; Touvron et al., 2023), there has been growing interest in leveraging LLMs for time series analysis (Jin et al., 2023). One key challenge lies in bridging the gap between these two distinct modalities. One line of approaches focuses on fine-tuning LLMs with specialized designs to empower them with time series analysis capabilities. The pilot work, GPT4TS (Zhou et al., 2023) introduces a unified framework for various time series analysis tasks based on GPT-2 (Radford et al., 2019) by fine-tuning its positional embeddings and layer normalization layers. Similarly, LLM4TS (Chang et al., 2023) proposes a two-stage fine-tuning strategy, encompassing time series alignment and forecasting fine-tuning to adapt LLMs to time series data. Others have explored keeping the LLMs frozen and aligning time series data with natural language. For instance, TimeLLM (Jin et al., 2024) reprograms the input time series with text prototypes to align the two modalities and AutoTimes (Liu et al., 2024c) independently embeds time series segments into the latent space of the LLM and train new projection layers of time series.

Despite the popularity of LLM4TS, Tan et al. argue that existing methods have yet to fully harness the powerful potential of LLMs, which limits their effectiveness in time series. As listed in Table 1, rather than previous works that take LLMs as the dominant backbone for prediction which is statistically ineffective but computationally expensive, MetaTST leverages LLMs as plug-in encoders for context-specific metadata, which can fully utilize the original capability of LLMs in semantic understanding.

## 3 METHOD

As aforementioned, to make up for the deficiency of the previous forecasting paradigm, this paper proposes to conduct informative time series forecasting. Instead of solely considering the time series

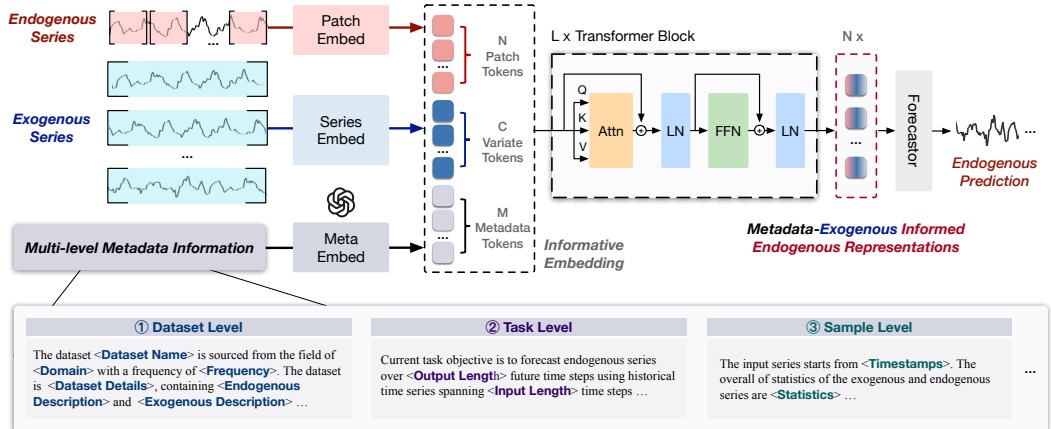

Figure 2: The overall design of MetaTST, which integrates endogenous series, exogenous series, and context-specific textual metadata to enable informative time series forecasting with Transformers.

modality, we propose MetaTST to incorporate valuable metadata information into the forecasting process to enable a comprehensive and direct understanding of the forecasting scenario. Technically, MetaTST consists of a well-designed metadata embedding mechanism to obtain multi-level metadata tokens. These metadata tokens are subsequently used along with exogenous series tokens to enrich target endogenous representations through a Transformer encoder.

## 3.1 INFORMATIVE TIME SERIES FORECASTING

In this paper, we highlight a new paradigm as informative time series forecasting, whose objective is to predict the future values of endogenous series based on information as sufficient as possible.

Considering the practicability, we study an essential and informative factor set as inputs, which includes historical observations $\mathbf{x}_{\mathrm{en}} \in \mathbb{R}^{T_{\mathrm{en}}}$ of endogenous series along with multiple relevant exogenous series $\mathbf{x}_{\mathrm{ex}} = \{\mathbf{x}_{\mathrm{ex},1}, \mathbf{x}_{\mathrm{ex},2}, \ldots, \mathbf{x}_{\mathrm{ex},C}\} \in \mathbb{R}^{T_{\mathrm{ex}} \times C}$ and corresponding metadata $\mathbf{x}_{\mathrm{meta}}$. $T_{\mathrm{en}}$ and $T_{\mathrm{ex}}$ denote the look-back lengths of endogenous and exogenous series respectively, and $C$ denotes the number of exogenous series. Noteworthily, metadata, referring to the information on the forecasting context (e.g. task description and variate meaning), is readily available in real-world applications, which just inherently maintained in the forecasting task definition. This means that our proposed informative forecasting paradigm can be seamlessly extended from canonical settings without the cost of newly collecting or labeling data. Thus, different from canonical formalization, the goal of informative forecasting in this paper is defined as learning deep models to accurately predict the future $S$ time steps of the endogenous series $\mathbf{y}_{\mathrm{en}} \in \mathbb{R}^S$ based on multiple inputs:

$$\arg\min_{\theta} \|\mathbf{y}_{\mathrm{en}} - \mathcal{F}_{\theta}(\mathbf{x}_{\mathrm{en}}, \mathbf{x}_{\mathrm{ex}}, \mathbf{x}_{\mathrm{meta}})\|_2^2. \tag{1}$$

where $\mathcal{F}_{\theta}(\cdot)$ represents the learned time series forecasting model parameterized by $\theta$.

## 3.2 METADATA EMBEDDING

Given the unstructured nature of metadata, we devise a multi-level metadata parser to structure it with well-designed natural language templates and further utilize large language models (LLMs) as the metadata encoder to exploit their vast prior knowledge of the world to facilitate a comprehensive understanding of the time series data and forecasting scenario from multi-level aspects.

**Multi-level Metadata Parser**  As shown in Figure 2, MetaTST introduces three types of tokens to incorporate metadata from three distinct perspectives: (1) providing essential properties about the *dataset*, such as domain and sampling frequency, empowering the model with external prior knowledge relevant to the forecasting scenario; (2) incorporating a description of the *task*, such as the target of interest, the length of input and output series, enhancing the model's understanding of the specific predictive behavior; (3) revealing dynamic statistics of time series *sample*, such as start timestamps, mean, and standard deviation, allowing the model to consider fine-grained differences

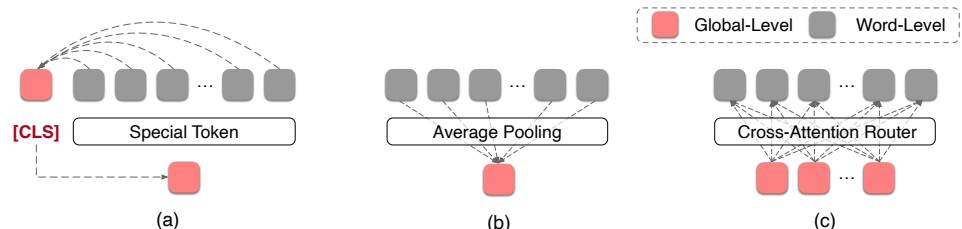

Figure 3: Different aggregating methods to transform word-level token sequences to global-level.

across samples. To incorporate diverse metadata information, MetaTST firstly introduces a metadata parsing module $\mathrm{MetaParser}(\cdot)$, which utilizes pre-defined language templates to structure the raw metadata into well-formalized, language-based metadata information across three distinct levels of granularity. Notably, each level of metadata can provide distinct perspectives on the prediction, enriching the understanding of the forecasting context. The process can be summarized as follows:

$$\{\widehat{\mathbf{x}}_{\mathrm{meta},k}\}_{k=1}^M = \mathrm{MetaParser}\left(\mathbf{x}_{\mathrm{meta}}\right). \tag{2}$$

$M$ is a hyperparameter for information levels, which is set as 3 for dataset, task and sample aspects.

**LLMs as the Metadata Encoder**  To further utilize the multi-level metadata information, MetaTST employs LLMs as the metadata encoder $\mathrm{LLMEncoder}(\cdot)$, where the LLM can be of any architecture, ranging from auto-regressive LLMs (e.g. Llama-3-8B [1], GPT-2 (2019)) to encoder-type LLMs like T5 (2020) and BERT (2018). As aforementioned, we introduce three language templates from different perspectives. Consequently, a descriptive paragraph is created for each point of view (bottom of Figure 2) and fed into the LLM encoder, resulting in multiple word-level language token sequences. To effectively incorporate these word-level language tokens into forecasting, we aggregate them into a global-level token for each paragraph, ultimately yielding three distinct metadata tokens.

Concretely, we explore three types of token aggregating methods as detailed in Figure 3: (a) employing the special global token, which is specially designed to encapsulate the entire sentence, like [CLS] token in BERT (Devlin et al., 2018); (b) using an average pooling layer to calculating the mean of all word-level token to generate a single global token; and (c) applying a router mechanism based on cross-attention mechanisms following (Zhang & Yan, 2022; Wang et al., 2024c) that define a small, fixed number of latent tokens as routers to aggregate information from all word-level tokens. Experimentally, we observe that an average pooling layer $\mathrm{AvgPooling}(\cdot)$ can achieve the best performance in most cases (see results in Figure 6(a)), which also presents favorable efficiency. Thus, we choose average pooling as the final design. Additionally, we employ a simple but effective modality alignment module $\mathrm{ModalAlign}(\cdot)$, which contains two linear layers with an in-between activation function to ensure alignment in both modality and latent dimensionality between LLMs and native time series models. The overall process can be formalized as follows:

$$\begin{aligned}\{\widetilde{\mathbf{h}}_{\mathrm{meta},k}\}_{k=1}^M &= \mathrm{AvgPooling}\left(\mathrm{LLMEncoder}\left(\{\widehat{\mathbf{x}}_{\mathrm{meta},k}\}_{k=1}^M\right)\right),\\ \{\mathbf{h}_{\mathrm{meta},k}\}_{k=1}^M &= \mathrm{ModalAlign}\left(\{\widetilde{\mathbf{h}}_{\mathrm{meta},k}\}_{k=1}^M\right).\end{aligned} \tag{3}$$

We summarize the process of metadata embedding as $\{\mathbf{h}_{\mathrm{meta},k}\}_{k=1}^M = \mathrm{MetaEmbed}\left(\{\widehat{\mathbf{x}}_{\mathrm{meta},k}\}_{k=1}^M\right)$, where $M$ represents the total numbers of metadata tokens.

### 3.3 METATST

MetaTST boosts the forecasting performance by employing informative embedding which aggregates endogenous, exogenous, and metadata tokens, and further utilizes Transformer Encoder to generate metadata-exogenous informed endogenous representations for informative time series forecasting.

**Informative Embedding**  Following the well-acknowledged time series modeling approaches (Nie et al., 2023; Dong et al., 2023), MetaTST splits the endogenous series $\mathbf{x}_{\mathrm{en}}$ into $N = \lfloor \frac{T_{\mathrm{en}}}{P} \rfloor$ non-overlapping patches, where $P$ is patch length. For the $i$-th patch, it is embedded into a $D$-dimensional endogenous token $\mathbf{h}_{\mathrm{en},i}$ through a trainable linear projection $\mathrm{PatchEmbed}(\cdot): \mathbb{R}^P \to \mathbb{R}^D$. We also adopt variate-wise embedding $\mathrm{SeriesEmbed}(\cdot): \mathbb{R}^{T_{\mathrm{ex}}} \to \mathbb{R}^D$ for related exogenous series, which is

---

[1]https://llama.meta.com/llama3

implemented by a temporal linear layer to map the whole exogenous series $\mathbf{x}_{\text{ex},j}$ into a $D$-dimensional exogenous token $\mathbf{h}_{\text{ex},j}$. The above-described design can highlight the temporal information of endogenous series and avoid the potential temporally mismatch problems w.r.t. exogenous series (Wang et al., 2024c). These two embedding processes are formalized as follows:

$$\{\mathbf{h}_{\text{en},i}\}_{i=1}^{N} = \text{PatchEmbed}\left(\mathbf{x}_{\text{en}}\right), \quad \{\mathbf{h}_{\text{ex},j}\}_{j=1}^{C} = \text{SeriesEmbed}\left(\{\mathbf{x}_{\text{ex},j}\}_{j=1}^{C}\right). \quad (4)$$

In addition to the above two types of series tokens, metadata tokens have already been aligned to time series modality as formalized in Eq. (3). Thus, MetaTST directly concatenates these three types of tokens to construct the informative embedding $\mathbf{h}^0$, including $N$ patch-wise endogenous tokens, $C$ series-wise exogenous tokens, and $M$ metadata tokens, which can be formalized as follows:

$$\mathbf{h}^0 = \text{Concat}\left(\{\mathbf{h}_{\text{en},i}\}_{i=1}^{N}, \{\mathbf{h}_{\text{ex},j}\}_{j=1}^{C}, \{\mathbf{h}_{\text{meta},k}\}_{k=1}^{M}\right). \quad (5)$$

**Informative Forecasting**   To communicate three types of tokens, we employ a Transformer encoder with $L$ layers for representation learning, whose attention mechanism can progressively fuse meta and exogenous information to the first $N$ endogenous tokens. As a result, we obtain $N$ metadata-exogenous informed endogenous representations $\mathbf{h}_{\text{en}}^{L}$ to ensure informative forecasting, which is:

$$\mathbf{h}^{l+1} = \text{TransformerBlock}\left(\mathbf{h}^l\right), \ l \in \{1, \cdots, L\}, \quad \widehat{\mathbf{y}}_{\text{en}} = \text{Forecastor}\left(\mathbf{h}_{\text{en}}^{L}\right), \quad (6)$$

where $\text{Forecastor}(\cdot)$ is instantiated as a linear layer to regress the prediction of endogenous series $\widehat{\mathbf{y}}_{\text{en}} \in \mathbb{R}^{S_{\text{en}}}$ from informative endogenous representations $\mathbf{h}_{\text{en}}^{L}$. Finally, as formalized in Eq. (1), MetaTST is trained using the L2 loss between the prediction and the ground truth.

## 4 EXPERIMENTS

We conduct extensive experiments on two well-established time series forecasting tasks to evaluate MetaTST: short- and long-term forecasting with exogenous series, covering twelve benchmarks. In addition to the conventional single-dataset individual training protocol, we also experiment with the new multi-dataset joint training to test the model capability in diverse forecasting scenarios.

**Benchmarks and Baselines**   In the experiments, we include twelve widely used benchmarks in total. Specifically, for the short-term forecasting with exogenous series task, we employ the EPF benchmark (Lago et al., 2021), which comprises five electricity price forecasting subsets derived from real-world power markets. Meanwhile, we conduct long-term forecasting with exogenous series based on seven well-established public datasets from diverse domains (Wu et al., 2021). As for baselines, we extensively compare MetaTST with ten well-acknowledged forecasting models, including LLM4TS models: GPT4TS (2023), TimeLLM (2024) and advanced native time series models: Autoformer (2021), Crossformer (2022), DLinear (2023), TimesNet (2023), PatchTST (2023), iTransformer (2024b), Timer (2024d), and TimeXer (2024c). More details are in the Appendix A.

**Individual and Joint Training Settings**   Previous methods mainly experiment with single-dataset individual training setups (Wu et al., 2021; Nie et al., 2023), which means the training set only contains data from one single domain. This conventional setting can well test the model's capacity to handle one specific task. Recently, in pursuing the foundation time series model, handling diverse forecasting scenarios has become an indispensable capability. Thus, in this paper, we further test MetaTST in the multi-dataset joint training setting. Compared to individual training, this joint training strategy requires the model to have enough capacity to cover diverse training sets and generalize well in shifted data distribution, inconsistent variate numbers, and varied semantic meanings. It is worth noticing that not all the baselines can handle varied variate numbers. Thus, we only compare with PatchTST (2023), iTransformer (2024b), and TimeXer (2024c) in joint training experiments.

**Model Implementations**   To ensure a fair comparison, for the individual training, we search hyperparameters in model configurations of all baselines in different benchmarks following the experiment strategy in (Nie et al., 2023; Wang et al., 2024a). However, this search protocol will lead to inconsistent model size among different benchmarks, which is contradictory to the unified model joint training setting. Thus, for the joint training experiments, we adjust the hyperparameters to ensure all the models have a comparable parameter size and keep consistent for all sub-datasets.

Table 2: Short-term forecasting results under single-dataset individual training. The input and output lengths are set to 168 and 24 following (Olivares et al., 2023). For clarity, the best result is in **bold**. *Avg.* is the average forecasting performance among all benchmarks.

| Datasets | | NP | | PJM | | BE | | FR | | DE | | Avg. | |
|---|---|---|---|---|---|---|---|---|---|---|---|---|---|
| Models | | MSE | MAE | MSE | MAE | MSE | MAE | MSE | MAE | MSE | MAE | MSE | MAE |
| TS Native | Autoformer (2021) | 0.402 | 0.398 | 0.168 | 0.267 | 0.500 | 0.333 | 0.519 | 0.295 | 0.674 | 0.544 | 0.453 | 0.367 |
| | DLinear (2023) | 0.309 | 0.321 | 0.108 | 0.215 | 0.463 | 0.431 | 0.429 | 0.260 | 0.520 | 0.463 | 0.366 | 0.338 |
| | TimesNet (2023) | 0.250 | 0.289 | 0.097 | 0.195 | 0.419 | 0.288 | 0.431 | 0.234 | 0.502 | 0.446 | 0.340 | 0.290 |
| | Crossformer (2022) | 0.240 | 0.285 | 0.101 | 0.199 | 0.420 | 0.290 | 0.434 | 0.208 | 0.461 | 0.432 | 0.331 | 0.283 |
| | PatchTST (2023) | 0.267 | 0.284 | 0.106 | 0.209 | 0.400 | 0.262 | 0.411 | 0.220 | 0.574 | 0.498 | 0.352 | 0.295 |
| | iTransformer (2024b) | 0.265 | 0.300 | 0.097 | 0.197 | 0.394 | 0.270 | 0.439 | 0.233 | 0.479 | 0.443 | 0.335 | 0.289 |
| | Timer (2024d) | 0.275 | 0.294 | 0.095 | 0.193 | 0.380 | 0.254 | 0.437 | 0.211 | 0.469 | 0.432 | 0.331 | 0.277 |
| | TimeXer (2024c) | 0.236 | 0.268 | 0.093 | 0.192 | 0.379 | **0.243** | 0.385 | 0.208 | 0.440 | 0.415 | 0.307 | 0.265 |
| LLM4TS | GPT4TS (2023) | 0.282 | 0.302 | 0.109 | 0.219 | 0.421 | 0.281 | 0.395 | 0.220 | 0.513 | 0.459 | 0.344 | 0.296 |
| | TimeLLM (2024) | 0.330 | 0.330 | 0.134 | 0.248 | 0.448 | 0.290 | 0.455 | 0.253 | 0.542 | 0.472 | 0.382 | 0.319 |
| **MetaTST (Ours)** | | 0.239 | **0.267** | **0.089** | **0.188** | **0.364** | 0.244 | **0.384** | **0.210** | **0.423** | **0.409** | **0.300** | **0.264** |

Table 3: Long-term forecasting results under individual training. The input length is set to 96. Results are averaged from 4 different prediction lengths {96, 192, 336, 720}. See Table 16 for full results.

| Datasets | | ETTh1 | | ETTh2 | | ETTm1 | | ETTm2 | | Weather | | Traffic | | ECL | | Avg. | |
|---|---|---|---|---|---|---|---|---|---|---|---|---|---|---|---|---|---|
| Models | | MSE | MAE | MSE | MAE | MSE | MAE | MSE | MAE | MSE | MAE | MSE | MAE | MSE | MAE | MSE | MAE |
| TS Native | Autoformer (2021) | 0.130 | 0.282 | 0.242 | 0.386 | 0.085 | 0.230 | 0.154 | 0.305 | 0.006 | 0.060 | 0.302 | 0.353 | 0.495 | 0.528 | 0.202 | 0.306 |
| | DLinear (2023) | 0.116 | 0.259 | 0.224 | 0.369 | 0.066 | 0.188 | 0.126 | 0.263 | 0.006 | 0.066 | 0.323 | 0.404 | 0.393 | 0.457 | 0.179 | 0.287 |
| | TimesNet (2023) | 0.076 | 0.215 | 0.210 | 0.369 | 0.054 | 0.175 | 0.129 | 0.271 | 0.097 | 0.115 | 0.171 | 0.264 | 0.410 | 0.476 | 0.164 | 0.269 |
| | Crossformer (2022) | 0.285 | 0.447 | 1.027 | 0.873 | 0.411 | 0.548 | 0.976 | 0.769 | 0.005 | 0.055 | 0.182 | 0.268 | 0.344 | 0.412 | 0.461 | 0.482 |
| | PatchTST (2023) | 0.078 | 0.215 | 0.192 | 0.345 | 0.053 | 0.173 | 0.120 | 0.258 | **0.002** | 0.031 | 0.173 | 0.253 | 0.394 | 0.446 | 0.145 | 0.246 |
| | iTransformer (2024b) | 0.075 | 0.211 | 0.199 | 0.352 | 0.053 | 0.175 | 0.127 | 0.267 | **0.002** | 0.031 | 0.161 | 0.246 | 0.365 | 0.442 | 0.140 | 0.246 |
| | Timer (2024d) | 0.081 | 0.220 | 0.186 | 0.344 | 0.053 | 0.173 | 0.139 | 0.280 | **0.002** | 0.034 | 0.340 | 0.409 | 0.364 | 0.425 | 0.166 | 0.269 |
| | TimeXer (2024c) | 0.073 | 0.209 | 0.189 | 0.342 | 0.052 | 0.171 | 0.120 | 0.258 | **0.002** | 0.031 | 0.156 | 0.234 | 0.327 | 0.408 | 0.132 | 0.236 |
| LLM4TS | GPT4TS (2023) | 0.077 | 0.214 | 0.189 | 0.341 | 0.052 | 0.171 | 0.120 | 0.256 | **0.002** | 0.031 | 0.185 | 0.286 | 0.362 | 0.429 | 0.141 | 0.247 |
| | TimeLLM (2024) | 0.077 | 0.215 | 0.199 | 0.352 | 0.053 | 0.173 | 0.122 | 0.261 | 0.003 | 0.036 | 0.186 | 0.271 | 0.365 | 0.413 | 0.144 | 0.246 |
| **MetaTST (Ours)** | | **0.069** | **0.203** | **0.182** | **0.335** | **0.051** | **0.170** | **0.118** | **0.254** | **0.002** | **0.029** | **0.146** | **0.227** | **0.308** | **0.402** | **0.125** | **0.231** |

## 4.1 SINGLE-DATASET INDIVIDUAL TRAINING

**Short-term Forecasting**   As shown in Table 2, MetaTST consistently delivers state-of-the-art performance on most of the datasets. Compared to advanced LLM4TS works GPT4TS (2023) and TimeLLM (2024), MetaTST achieves average MSE reductions of 12.8% (0.300 vs. 0.344) and 21.5% (0.300 vs. 0.382) respectively, demonstrating the effectiveness of encoder-type LLM usage in MetaTST. Notably, TimeXer (2024c), the latest model in forecasting with exogenous series, achieves comparable performance with MetaTST on NP and FR datasets. This may be attributed to the fact these datasets exhibit highly correlated variates, thereby solely including exogenous series can already enable a relatively informative prediction. Nonetheless, MetaTST still achieves the best average performance across all datasets, highlighting the effectiveness of metadata in enhancing prediction accuracy, whose contribution will be more significant in more complex joint training settings.

**Long-term Forecasting**   We evaluate MetaTST on long-term forecasting benchmarks in Table 3, where MetaTST achieves consistent state-of-the-art performance across four prediction lengths. On the average of all benchmarks, MeTaTST achieves 4.9% MSE reduction compared to TimeXer (2024c), 11.3% MSE reduction compared to the LLM4TS baseline GPT4TS (2023). This indicates that MetaTST effectively captures valuable information from language-based metadata to informative time series predictions, uniformly benefiting extensive prediction tasks.

## 4.2 MULTI-DATASET JOINT TRAINING

Going beyond training a dataset-specific model, we develop a multi-dataset joint training setting that trains models based on mixing datasets. Larger-scale data from various datasets not only provide more

Table 4: Short-term forecasting results under multi-dataset joint training. Promotion refers to the relative error reduction of joint training w.r.t. individual training ($1 - \frac{\text{Joint error}}{\text{Individual error}}$). ↑ and ↓ indicate the positive and negative effects brought by joint training respectively.

| Models | | NP | | PJM | | BE | | FR | | DE | | Avg. | |
|---|---|---|---|---|---|---|---|---|---|---|---|---|---|
| Scenarios | | MSE | MAE | MSE | MAE | MSE | MAE | MSE | MAE | MSE | MAE | MSE | MAE |
| Individual | PatchTST (2023) | 0.263 | 0.278 | 0.095 | 0.206 | 0.399 | 0.259 | 0.415 | 0.222 | 0.462 | 0.429 | 0.327 | 0.279 |
| | iTransformer (2024b) | 0.415 | 0.391 | 0.163 | 0.273 | 0.560 | 0.368 | 0.530 | 0.298 | 0.656 | 0.538 | 0.465 | 0.374 |
| | TimeXer (2024c) | 0.275 | 0.289 | 0.090 | 0.217 | 0.408 | 0.259 | 0.424 | 0.225 | 0.465 | 0.432 | 0.332 | 0.284 |
| | **MetaTST (Ours)** | 0.244 | 0.273 | 0.101 | 0.200 | 0.377 | 0.246 | 0.405 | 0.220 | 0.446 | 0.419 | 0.315 | 0.272 |
| Joint | PatchTST (2023) | 0.256↑ | 0.273↑ | 0.088↑ | 0.190↑ | 0.342↑ | 0.240↑ | 0.360↑ | 0.194↑ | 0.466↓ | 0.430↓ | 0.302↑ | 0.265↑ |
| | iTransformer (2024b) | 0.376↑ | 0.377↑ | 0.154↑ | 0.260↑ | 0.516↑ | 0.337↑ | 0.531↓ | 0.298 | 0.661↓ | 0.550↓ | 0.448↑ | 0.364↑ |
| | TimeXer (2024c) | 0.262↑ | 0.276↑ | **0.085**↑ | **0.181**↑ | 0.358↑ | 0.242↑ | 0.384↑ | 0.196↑ | 0.464↑ | 0.430↑ | 0.311↑ | 0.265↑ |
| | **MetaTST (Ours)** | **0.234**↑ | **0.263**↑ | 0.087↑ | 0.186↑ | **0.318**↑ | **0.234**↑ | **0.329**↑ | **0.193**↑ | **0.435**↑ | **0.415**↑ | **0.281**↑ | **0.258**↑ |
| Promotion | | 4.1% | 3.7% | 13.9% | 7.0% | 15.6% | 4.9% | 18.8% | 12.3% | 2.5% | 1.0% | 10.8% | 4.9% |

Table 5: Long-term forecasting under multi-dataset joint training. Look-back length is fixed to 96. Results are averaged from four prediction lengths $\{96, 192, 336, 720\}$. See Table 17 for full results.

| Models | | ETTh1 | | ETTh2 | | ETTm1 | | ETTm2 | | Weather | | Traffic | | ECL | | Avg. | |
|---|---|---|---|---|---|---|---|---|---|---|---|---|---|---|---|---|---|
| Scenarios | | MSE | MAE | MSE | MAE | MSE | MAE | MSE | MAE | MSE | MAE | MSE | MAE | MSE | MAE | MSE | MAE |
| Individual | PatchTST (2023) | 0.077 | 0.214 | 0.201 | 0.353 | 0.054 | 0.173 | **0.120** | **0.257** | 0.002 | 0.031 | 0.191 | 0.283 | 0.382 | 0.443 | 0.147 | 0.250 |
| | iTransformer (2024b) | 0.079 | 0.217 | 0.204 | 0.357 | 0.055 | 0.179 | 0.131 | 0.275 | 0.002 | 0.032 | 0.270 | 0.365 | 0.444 | 0.495 | 0.169 | 0.274 |
| | TimeXer (2024c) | 0.078 | 0.216 | 0.197 | 0.353 | 0.055 | 0.174 | 0.121 | 0.258 | 0.002 | 0.031 | 0.190 | 0.281 | 0.368 | 0.438 | 0.144 | 0.250 |
| | **MetaTST (Ours)** | 0.078 | 0.215 | 0.202 | 0.353 | 0.054 | 0.174 | 0.130 | 0.267 | **0.002** | **0.030** | 0.180 | 0.267 | 0.343 | 0.422 | 0.141 | 0.247 |
| Joint | PatchTST (2023) | 0.078↓ | 0.216↓ | 0.203↓ | 0.351↑ | 0.052↑ | **0.171**↑ | 0.127↓ | 0.263↓ | 0.002 | 0.031↓ | 0.218↓ | 0.302↓ | 0.366↑ | 0.437↑ | 0.149↓ | 0.253↓ |
| | iTransformer (2024b) | 0.083↓ | 0.224↓ | 0.223↓ | 0.376↓ | 0.055 | 0.179 | 0.139↓ | 0.286↓ | 0.002 | 0.032 | 0.541↓ | 0.562↓ | 0.599↓ | 0.588↓ | 0.235↓ | 0.321↓ |
| | TimeXer (2024c) | 0.078 | 0.215↑ | 0.199↓ | 0.350↓ | 0.053↑ | 0.173↑ | 0.128↓ | 0.266↓ | **0.002** | **0.030**↑ | 0.198↓ | 0.288↓ | 0.390↓ | 0.452↓ | 0.150↓ | 0.253↓ |
| | **MetaTST (Ours)** | **0.077**↑ | **0.213**↑ | **0.196**↑ | **0.349**↑ | **0.052**↑ | **0.171**↑ | 0.124↑ | 0.262↑ | **0.002** | **0.030** | **0.171**↑ | **0.261**↑ | **0.332**↑ | **0.414**↑ | **0.136**↑ | **0.243**↑ |
| Promotion | | 1.3% | 0.9% | 3.0% | 1.1% | 3.7% | 1.7% | 4.6% | 1.9% | - | - | 5.0% | 2.3% | 3.2% | 1.9% | 3.54% | 1.62% |

diverse information but also introduce more complex temporal variations. This poses a significant challenge for the forecasting model to handle complex and diverse forecasting scenarios. Note that as we described in model implementations, to train a unified model for all seven datasets, we have to use uniform model hyperparameters for different datasets, which makes the individual training results in Table 4-5 consistently inferior to Table 2-3. However, the relative promotion between individual and joint training can serve as a valuable metric for comparing model capacity and generalizability.

**Short-term Forecasting** These five short-term forecasting datasets are all about electricity price forecasting. They hold similar forecasting scenarios and a consistent number of exogenous variables. We train a unified model by mixing all five datasets and directly evaluate its zero-shot performance on each dataset. As listed in Table 4, we observe that the multi-dataset joint training from similar domains could consistently enhance model performance. Notably, MetaTST outperforms all baseline models, achieving remarkable zero-shot performance that even exceeds the searched hyperparameter results shown in Table 2. These results underscore the benefit of incorporating metadata, which significantly enhances MetaTST's understanding of domain-specific and sharing temporal patterns through context-specific information, thereby improving its adaptability to diverse forecasting scenarios.

**Long-term Forecasting** Since long-term forecasting datasets are from distinct domains with inconsistent variates, mismatched frequencies, and vastly different meanings, it is hard to directly apply zero-shot generalization. Thus, following (Goswami et al., 2024), we trained a unified model based on the data mixed from all seven datasets and linearly probed it to each dataset, which requires the model to learn generalizable representations. As shown in Table 5, linear probing results of all baselines are consistently inferior to results under individual training. This is unsurprising since the discrepancies among multiple datasets can confound the model, particularly when they exhibit contradictory temporal patterns. In contrast, enhanced by metadata-guided joint training, MetaTST benefits from joint training even under distinct datasets and achieves overall state-of-the-art.

### 4.3 ABLATIONS STUDIES

We conducted extensive ablation studies to validate the effectiveness of various designs in MetaTST, including endogenous series (*En.*), exogenous series (*Ex.*), and metadata (*Meta*). Results in Figure 4

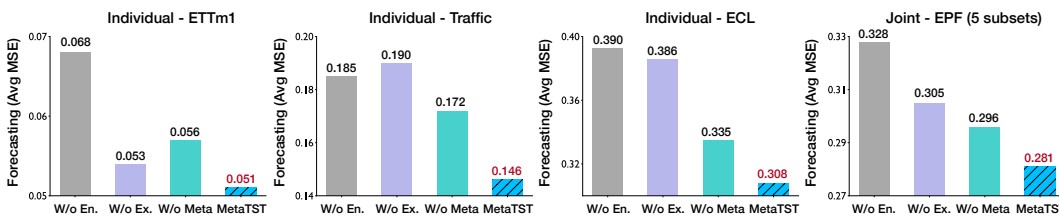

Figure 4: Ablation studies of MetaTST with various types informative forecasting, covering individual and joint training strategy in long- and short-term forecasting tasks. More details are in Appendix C.

demonstrate that all three types of inputs are favorable for the prediction, with endogenous series proving to be the most critical factor. The absence of endogenous series leads to a loss of essential temporal information, resulting in a significant degradation of forecasting performance. In datasets with a substantial proportion of exogenous series, such as Traffic and ECL, correlations between endogenous and exogenous series also play an essential role, offering valuable insights into achieving accurate results. While in datasets with a limited number of exogenous series, such as ETTm1 and EPF, the incorporation of metadata yields significant improvements in forecasting performance.

## 4.4 DIVE INTO METADATA ENCODER

**Encoder or Decoder** We investigate the use of various large language models (LLMs) as the metadata encoder for MetaTST, encompassing different architectures and scales. As illustrated in Figure 5(a), we can find that MetaTST consistently achieves excellent results across various LLMs, highlighting the generality of MetaTST. We provide the full results in Table 11 of the Appendix. Notably, we observe a preference for encoder-based LLMs, such as BERT (2018) and T5 (2020), over generative decoder-based LLMs. This may be because MetaTST leverages LLMs to process textual metadata information into latent tokens instead of generating future predictions.

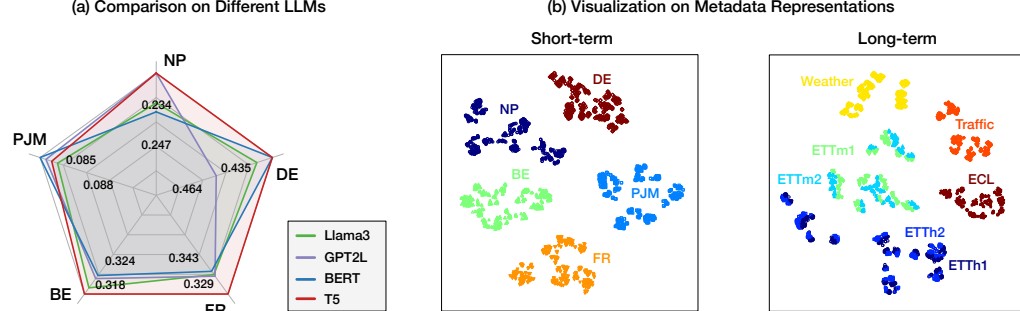

Figure 5: (a) Performance comparison on different LLMs as the metadata encoder and (b) Representation visualization of metadata on short-term (left) and long-term (right) joint training settings.

**Metadata Guided by LLMs** We visualize different metadata representations using t-SNE (der Maaten et al., 2008), as shown in Figure 5(b). Specifically, we perform a quantitative analysis of the distribution of test set representations of metadata in both short-term and long-term joint training settings. The results show that metadata representations from different datasets are distinguishable, suggesting that domain-invariant features have been successfully learned. Furthermore, we observe that metadata representations from similar datasets (e.g., ETTh1 vs. ETTh2) exhibit significantly closer clustering compared to more distinct datasets (e.g., ETTh1 vs. Traffic, Weather). This can be attributed to the metadata information, where more similar and specific contexts (e.g., domain, frequency, etc.) are constructed. This further demonstrates that valuable prior knowledge can be introduced to improve time series forecasting through reasonable metadata design.

**Metadata Token Aggregating** In the metadata encoder, we explore various token aggregation methods for incorporating different levels of metadata tokens into the prediction, including a special token, average pooling, and router mechanism. Concretely, for the router mechanism, we vary the number of routers in $\{3, 6, 12\}$. Full results are listed in Table 15 of Appendix. As presented in Figure 6(a), we find that a simple average pooling method yields better results. Therefore, we adopt it as the token aggregation method in MetaTST to transform word-level token sequences to global-level.

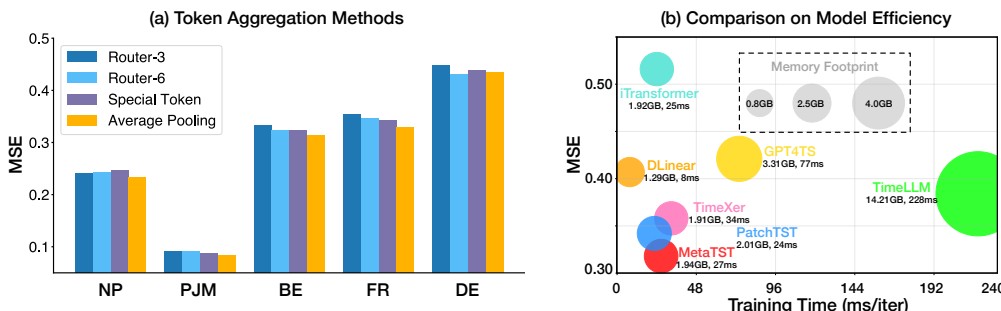

Figure 6: Comparison on (a) different token aggregation methods and (b) different model efficiency.

**Efficiency Analysis**   We conduct a comprehensive efficiency analysis of MetaTST with various baselines. Specifically, for the native time series models and our proposed MetaTST, we employ a unified model trained from a multi-dataset joint training strategy with unified model hyperparameters. For the LLM4TS models, we use a six-layer GPT-2 as the backbone for all baselines. The efficiency results under multi-dataset joint training setting are presented in Figure 6(b). We can find that under the same model configuration, MetaTST outperforms all baselines with favorable efficiency. Despite LLM-based baselines introducing elaborated fine-tuning methods, the cost of training and inference is ineffective. In contrast, MetaTST employs a fully-frozen LLM as a metadata encoder and enjoys a lower computational cost and better forecasting performance.

**Case Studies**   As illustrated in Figure 7, the attention map highlights the correlations between endogenous patches, exogenous series, and metadata. It is clear that different information contributes to the predictions with varying significance, where three types of embedding hold distinct patterns in the attention map. This observation indicates that benefiting from advanced attention mechanisms in Transformers, MetaTST effectively distinguishes the various types of information, identifies strong associations, and learns discriminative attention weights for different endogenous patches, thereby accurately predicting future variations. More case studies can be found in Figure 12 of Appendix.

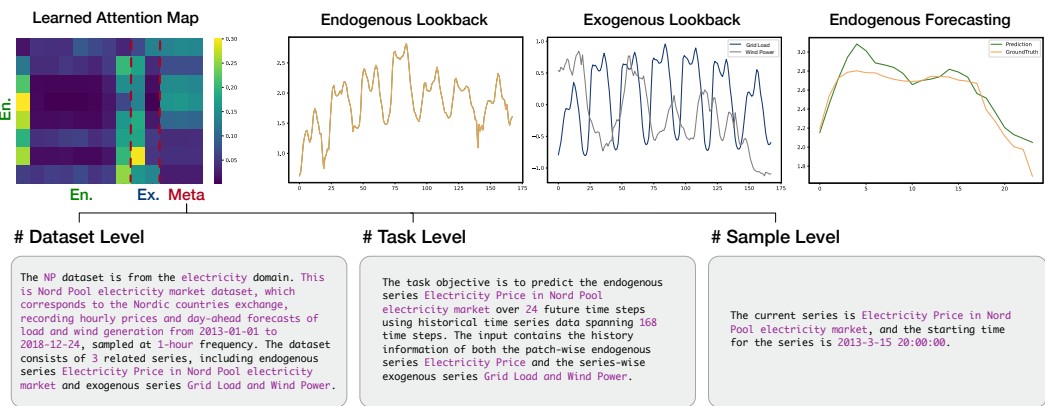

Figure 7: Visualization of raw endogenous series (*En.*), exogenous series (*Ex.*), language-based metadata (*Meta*) from the NP dataset, and the learned attention maps in MetaTST. Attention map is calculated by averaging the attention matrices over all the heads and across all the layers.

## 5   CONCLUSION

This paper highlights a new paradigm as informative time series forecasting and presents MetaTST to seamlessly incorporate multi-level metadata to facilitate the prediction. By formalizing unstructured metadata with pre-designed language templates and employing LLMs as the metadata encoder, MetaTST can provide a comprehensive understanding of forecasting scenarios, ultimately enabling more informative forecasts. Experimentally, MetaTST outperforms advanced forecasters with favorable efficiency on both short- and long-term forecasting tasks. More remarkably, MetaTST demonstrates significant adaptability to diverse scenarios and achieves state-of-the-art performance in multi-dataset joint training settings, posing a potential solution for time series foundation models.

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

## A    DATASET DESCRIPTIONS

We conduct short-term and long-term prediction experiments on real-world datasets, respectively, to evaluate the performance of our proposed MetaTST. In alignment with the forecasting setup using exogenous series introduced by TimeXer (Wang et al., 2024c), we stick to the original dataset configuration, designating the target series of the dataset as the endogenous series and all other related series as the exogenous series, covering all experimental settings comprehensively.

For short-term forecasting, we utilize real-world benchmarks for forecasting with exogenous series, derived from five major power markets (Olivares et al., 2023). The configurations of endogenous and exogenous series are summarized in Table 6, with further details provided as follows:

(1) **NP**: The Nord Pool electricity market, which records hourly electricity prices, corresponding grid load, and wind power forecasts from January 1, 2013, to December 24, 2018.

(2) **PJM**: The Pennsylvania-New Jersey-Maryland market, which contains zonal electricity prices in the Commonwealth Edison (COMED) and corresponding system load and COMED load forecasts from January 1, 2013, to December 24, 2018.

(3) **BE**: Belgium's electricity market, which records hourly electricity prices, load forecasts in Belgium, and generation forecasts in France from January 9, 2011, to December 31, 2016.

(4) **FR**: The electricity market in France, which records hourly prices along with corresponding load and generation forecasts from January 9, 2012, to December 31, 2017.

(5) **DE**: The German electricity market, which records hourly prices, zonal load forecasts in the TSO Amprion zone, and wind and solar generation forecasts from January 9, 2012, to December 31, 2017.

As for long-term forecasting, we adhere to the experimental setting of forecasting with exogenous variables outlined in (Wang et al., 2024c) where the last dimension of multivariate data is designated as the endogenous series, and the others are treated as exogenous variables. We evaluate model performance on seven well-established benchmarks across four different domains as follows:

(1) **ECL** (Li et al., 2019), comprising hourly electricity consumption data from 321 clients. We treat the consumption of the last client as the endogenous variable, while the data from the other clients serve as exogenous variables.

(2) **Weather** (Wu et al., 2021) recording 21 meteorological factors every 10 minutes from the Weather Station of the Max Planck Biogeochemistry Institute in 2020. We use the Wet Bulb factor as the endogenous variable, with the remaining indicators as exogenous variables.

(3) **ETT** (Zhou et al., 2021) including four subsets: ETTh1 and ETTh2, recorded hourly, and ETTm1 and ETTm2, recorded every 15 minutes. The oil temperature is the endogenous variable, accompanied by six power load features as exogenous variables.

(4) **Traffic** (Wu et al., 2023) recording hourly road occupancy rates measured by 862 sensors on San Francisco Bay Area freeways. The measurement from the last sensor is used as the endogenous variable, with the other sensors serving as exogenous.

We follow the same data processing and train-validation-test set split protocol in TSLib (Wang et al., 2024b), where the train, validation, and test datasets are split by the ratio of 6:2:2 for the ETT dataset and 7:1:2 for the other datasets. As for the forecasting setting, we set the look-back length of both endogenous and exogenous series to 96 for long-term forecasting tasks, and the prediction horizon varies in $\{96, 192, 336, 720\}$. For those short-term electricity price datasets, we fix the look-back length and prediction length to 168 and 24 respectively.

## B    IMPLEMENTATION DETAILS

All experiments were implemented using PyTorch (Paszke et al., 2019) and conducted on a single NVIDIA A100 40GB GPU. We employed the ADAM optimizer (Kingma, 2014) with an initial learning rate of 1e-4 and L2 loss for model optimization. Our proposed method involves several key hyperparameters: the number of Transformer blocks ($e_{\text{layers}}$) is set from $\{1, 2, 3\}$; the hidden dimension ($d_{\text{model}}$) is set from $128, 256, 512$; the dimensions of the feedforward layer ($d_{\text{ff}}$) are

Table 6: Dataset descriptions. *Ex.* and *En.* are abbreviations for the Exogenous series and Endogenous series, respectively. The dataset size is organized in (Train/Validation/Test).

| Dataset | Dim | Ex. Descriptions | En. Descriptions | Frequency | Dataset Size | Domain |
|---|---|---|---|---|---|---|
| Electricity | 321 | Electricity Consumption | Electricity Consumption | 1 Hour | 18,317/2,633/5,261 | Electricity |
| Weather | 21 | Climate Feature | CO2-Concentration | 10 Minutes | 36,792/5,271/10,540 | Weather |
| ETTh | 7 | Power Load Feature | Oil Temperature | 1 Hour | 8,545/2,881/2,881 | Electricity |
| ETTm | 7 | Power Load Feature | Oil Temperature | 15 Minutes | 34,465/11,521/11,521 | Electricity |
| Traffic | 862 | Road Occupancy Rates | Road Occupancy Rates | 1 Hour | 12,185/1,757/3,509 | Transportation |
| NP | 3 | Grid Load, Wind Power | Nord Pool Electricity Price | 1 Hour | 36,500/5,219/10,460 | Electricity |
| PJM | 2 | System Load, Zonal COMED Load | Pennsylvania-New Jersey-Maryland Electricity Price | 1 Hour | 36,500/5,219/10,460 | Electricity |
| BE | 3 | Generation, System Load | Belgium's Electricity Price | 1 Hour | 36,500/5,219/10,460 | Electricity |
| FR | 3 | Generation, System Load | France's Electricity Price | 1 Hour | 36,500/5,219/10,460 | Electricity |
| DE | 3 | Wind Power, Ampirion Zonal Load | German's Electricity Price | 1 Hour | 36,500/5,219/10,460 | Electricity |

explored from $512, 1024, 2048$; and the number of attention heads ($n_{\text{heads}}$) is tuned from $4, 8, 16$. Additionally, we carefully considered the training batch size from $\{16, 32, 64, 128\}$, and dropout from $\{0, 0.1, 0.2, 0.3\}$. Moreover, RevIN (Kim et al., 2021) are utilized in all experiments as an architecture-agnostic technique to address distribution shifts, aiming to better capture temporal dependencies. The patch length was uniformly set to 12 for long-term forecasting and 24 for short-term forecasting, with 10 training epochs across all datasets. We use only historical endogenous and exogenous series to predict future values of the endogenous series. Each series is processed through a distinct embedding layer tailored to its type. Positional encoding is then applied to the patchified endogenous series before the data is fed into a Transformer-based time series model for further processing. All compared baseline models are reproduced based on TSLib (Wang et al., 2024b).

Regarding multi-dataset joint training, we train a unified model across all datasets and a dataset-specific model using the same hyperparameters to quantify the benefits of dataset mixing. To ensure a fair comparison, both MetaTST and native time series baselines are built on identical configurations, which are detailed as follows:

Table 7: Unified hyperparameter values for all baselines in different benchmarks.

| Tasks | Model | | | | | | Training | | |
|---|---|---|---|---|---|---|---|---|---|
| | $e_{\text{layers}}$ | $d_{\text{model}}$ | $d_{\text{ff}}$ | $n_{\text{heads}}$ | patch | patch stride | learning rate | batch size | training epochs |
| Short-term | 3 | 256 | 2048 | 8 | 24 | 24 | 1e-4 | 32 | 10 |
| Long-term | 3 | 256 | 2048 | 8 | 12 | 12 | 1e-4 | 32 | 10 |

Furtherly, we explore the joint training strategy in both short- and long-term prediction tasks. To address the challenges of mismatched channel numbers and varying physical meanings across different time series datasets, we propose a batch mixing strategy. This strategy ensures that samples from the same dataset are grouped in the same training batch. The joint training strategy mitigates conflicts between different datasets in a single batch and reduces excessive padding when dealing with datasets with significant dimensional differences. Additionally, we present results from individual training to validate the capability of different models in handling diverse forecasting scenarios.

## C  ABLATION STUDIES

To verify the rationality of the design of our proposed MetaTST, we conduct detailed ablation studies by removing each component in the input tokens, covering meta information, exogenous series, and endogenous series. Due to the paper limit, we only report the average results in Figure 4 and provide detailed results and analysis here.

**Removing the Specific Types of Tokens**    We present the comprehensive ablation results under the long-term forecasting setting in Table 8, where a lower bar indicates a better performance. Notably, the removal of any component from MetaTST consistently leads to a decline in forecasting performance, underscoring the significance of each component in our proposed model. Among the three ablation designs, the removal of endogenous series results in the most pronounced reduction in forecasting performance, with an average decrease of 22.4%. This finding further reinforces the dominant role of endogenous variables in prediction, suggesting that they are the primary drivers of the forecasting performance. However, in certain datasets, such as Traffic, the exogenous variables surprisingly play a more crucial role than the endogenous variables. This phenomenon may be attributed to the unique characteristics of the Traffic dataset, which records road occupancy collected from sensors in different areas of the highway, potentially resulting in time lags between the variables. Additionally, we conduct ablation studies on the short-term forecasting dataset under a joint training setting. The ablation results in Table 9 consistently demonstrate that our design effectively leverages both temporal and metadata information, yielding improved performance. These results collectively demonstrate the effectiveness of MetaTST in harnessing the strengths of both endogenous and exogenous series, as well as temporal and metadata information, to achieve superior forecasting performance.

To further explore the effectiveness of metadata, we conduct ablation studies on different levels of metadata, as shown in Table 10. The results indicate that dataset-level metadata offers more distinctive contextual information in the multi-dataset joint training setting, playing a crucial role in the final performance. Other metadata types contribute additional valuable information, further enhancing the forecasting performance.

Table 8: Long-term forecasting ablations under single-dataset individual training.

| Design | ETTh1 | | ETTh2 | | ETTm1 | | ETTm2 | | Weather | | Traffic | | ECL | | Avg. | |
|---|---|---|---|---|---|---|---|---|---|---|---|---|---|---|---|---|
| | MSE | MAE | MSE | MAE | MSE | MAE | MSE | MAE | MSE | MAE | MSE | MAE | MSE | MAE | MSE | MAE |
| **W/o** Metadata information | 0.077 | 0.216 | 0.203 | 0.355 | 0.056 | 0.178 | 0.138 | 0.275 | **0.002** | 0.030 | 0.172 | 0.262 | 0.335 | 0.420 | 0.140 | 0.248 |
| **W/o** Exogenous series | 0.079 | 0.217 | 0.198 | 0.349 | 0.053 | 0.173 | 0.121 | 0.258 | **0.002** | 0.030 | 0.190 | 0.280 | 0.386 | 0.446 | 0.147 | 0.250 |
| **W/o** Endogenous series | 0.074 | 0.212 | 0.201 | 0.356 | 0.068 | 0.193 | 0.150 | 0.288 | **0.002** | 0.031 | 0.185 | 0.274 | 0.390 | 0.461 | 0.153 | 0.259 |
| **MetaTST** (**Ours**) | **0.069** | **0.203** | **0.182** | **0.335** | **0.051** | **0.170** | **0.118** | **0.254** | 0.002 | **0.029** | **0.146** | **0.227** | **0.308** | **0.402** | **0.125** | **0.231** |

Table 9: Short-term forecasting ablations under multi-dataset joint training.

| Design | NP | | PJM | | BE | | FR | | DE | | Avg. | |
|---|---|---|---|---|---|---|---|---|---|---|---|---|
| | MSE | MAE | MSE | MAE | MSE | MAE | MSE | MAE | MSE | MAE | MSE | MAE |
| **Only** Endogenous series | 0.261 | 0.275 | 0.092 | 0.195 | 0.350 | 0.241 | 0.366 | 0.197 | 0.465 | 0.432 | 0.307 | 0.268 |
| **Only** Exogenous series | 0.268 | 0.296 | 0.092 | 0.196 | 0.388 | 0.279 | 0.399 | 0.220 | 0.487 | 0.439 | 0.327 | 0.286 |
| **Only** Metadata information | 0.586 | 0.492 | 0.269 | 0.371 | 0.689 | 0.461 | 0.634 | 0.400 | 1.130 | 0.729 | 0.662 | 0.491 |
| **W/o** Metadata information | 0.237 | 0.264 | 0.088 | 0.187 | 0.324 | **0.234** | 0.338 | **0.192** | 0.487 | 0.421 | 0.296 | 0.263 |
| **W/o** Exogenous series | 0.255 | 0.272 | 0.090 | 0.192 | 0.355 | 0.242 | 0.366 | 0.195 | 0.459 | 0.428 | 0.305 | 0.266 |
| **W/o** Endogenous series | 0.267 | 0.298 | 0.100 | 0.201 | 0.393 | 0.272 | 0.399 | 0.222 | 0.481 | 0.438 | 0.328 | 0.286 |
| **MetaTST** (**Ours**) | **0.234** | **0.263** | **0.087** | **0.186** | **0.318** | **0.234** | **0.329** | 0.193 | **0.435** | **0.415** | **0.281** | **0.258** |

Table 10: Ablations on different levels of metadata under multi-dataset joint training.

| Design | NP | | PJM | | BE | | FR | | DE | | Avg. | |
|---|---|---|---|---|---|---|---|---|---|---|---|---|
| | MSE | MAE | MSE | MAE | MSE | MAE | MSE | MAE | MSE | MAE | MSE | MAE |
| **Only** Dataset-level metadata | 0.239 | 0.266 | 0.089 | 0.190 | 0.324 | 0.239 | 0.341 | 0.190 | 0.448 | 0.423 | 0.288 | 0.262 |
| **Only** Task-level metadata | 0.239 | 0.266 | 0.090 | 0.190 | 0.333 | 0.240 | 0.356 | 0.193 | 0.460 | 0.420 | 0.296 | 0.262 |
| **Only** Sample-level metadata | 0.291 | 0.313 | 0.105 | 0.209 | 0.406 | 0.279 | 0.432 | 0.237 | 0.524 | 0.473 | 0.352 | 0.302 |
| **MetaTST** (**Ours**) | **0.234** | **0.263** | **0.087** | **0.186** | **0.318** | **0.234** | **0.329** | 0.193 | **0.435** | **0.415** | **0.281** | **0.258** |

**Replacing Metadata Encoder with Different LLMs**    To further explore the generality of MetaTST, we conduct a comprehensive comparison of the model performance with different LLMs as the metadata encoder. In our main text, we presented experiments using the T5 model as the metadata encoder, demonstrating its effectiveness in generating valuable metadata representation. Here, we replace T5 with seven advanced LLMs, encompassing both Encoder-only and Decoder-only models,

to assess the impact of different language models on forecasting performance. As shown in Table 11, the differences in language models indeed lead to variations in prediction results. Notably, T5 emerges as the top-performing language model on average, underscoring its suitability for metadata encoding in the context of time series forecasting. These results collectively demonstrate the flexibility and adaptability of MetaTST, which can be easily integrated with various language models.

Table 11: Ablation Studies on different LLMs as the metadata Encoder.

| Design | Models | NP | | PJM | | BE | | FR | | DE | | Avg. | |
|---|---|---|---|---|---|---|---|---|---|---|---|---|---|
| | | MSE | MAE | MSE | MAE | MSE | MAE | MSE | MAE | MSE | MAE | MSE | MAE |
| Decoder-only | GPT2 (2019) | 0.250 | 0.267 | 0.089 | 0.188 | 0.328 | 0.234 | 0.349 | 0.193 | **0.429** | **0.414** | 0.289 | 0.259 |
| | GPT2M (2019) | 0.241 | 0.267 | 0.086 | 0.187 | 0.324 | 0.233 | 0.341 | 0.193 | 0.458 | 0.421 | 0.290 | 0.260 |
| | GPT2L (2019) | 0.234 | 0.264 | 0.086 | 0.187 | 0.323 | 0.235 | 0.340 | 0.194 | 0.464 | 0.422 | 0.289 | 0.260 |
| | Llama2 (2023) | 0.242 | 0.265 | 0.085 | 0.186 | 0.330 | 0.238 | 0.352 | 0.197 | 0.444 | 0.417 | 0.291 | 0.261 |
| | Llama3 | 0.244 | 0.267 | 0.088 | 0.189 | 0.320 | 0.236 | 0.341 | 0.196 | 0.443 | 0.419 | 0.287 | 0.261 |
| | LLM2Vec (2024) | 0.248 | 0.269 | 0.091 | 0.190 | 0.318 | 0.235 | 0.338 | 0.194 | 0.446 | 0.421 | 0.288 | 0.262 |
| Encoder-only | BERT (2018) | 0.247 | 0.268 | **0.085** | **0.185** | 0.324 | 0.235 | 0.343 | 0.193 | 0.435 | 0.416 | 0.287 | 0.259 |
| | T5 (2020) | **0.234** | **0.263** | 0.087 | 0.186 | **0.318** | **0.234** | **0.329** | **0.193** | 0.435 | 0.415 | **0.281** | **0.258** |

**Why is the LLMs-based Metadata Encoder?** LLM-based metadata embeddings offer a flexible approach to integrating context-specific metadata, making them highly adaptable to diverse time series analysis scenarios. Unlike one-hot encoding and learnable tokens, language-based meta-embeddings encode valuable prior knowledge, enabling more certain predictions beyond the capabilities of traditional methods. Below is a performance comparison between dataset-level one-hot encoding, learnable encoding, and our proposed MetaTST.

Table 12: Compared LLMs-based metadata encoding with one-hot and learnable encoding.

| Design | NP | | PJM | | BE | | FR | | DE | | Avg. | |
|---|---|---|---|---|---|---|---|---|---|---|---|---|
| | MSE | MAE | MSE | MAE | MSE | MAE | MSE | MAE | MSE | MAE | MSE | MAE |
| One-hot Encoding | 0.244 | 0.268 | 0.089 | 0.189 | 0.323 | 0.236 | 0.337 | 0.195 | 0.447 | 0.419 | 0.288 | 0.261 |
| Learnable Encoding | 0.242 | 0.268 | 0.090 | 0.190 | 0.328 | 0.237 | 0.345 | 0.199 | 0.458 | 0.422 | 0.293 | 0.263 |
| **MetaTST (Ours)** | **0.234** | **0.263** | **0.087** | **0.186** | **0.318** | **0.234** | **0.329** | **0.193** | **0.435** | **0.415** | **0.281** | **0.258** |

## D  MORE CERTAIN PREDICTION BY METATST

We conducted a validation experiment using Quantile Loss, setting the quantile parameters to $\tau = 0.9$ (Q90) and $\tau = 0.1$ (Q10), to evaluate model prediction certainty by introducing different types of information under complex multi-dataset joint training scenarios. The differences between Q90 and Q10 are calculated on the FR test set, and smaller discrepancies typically indicate higher predictive certainty. Results in Table 13 show that the predictive reliability of the model improves progressively as exogenous series and metadata are incrementally introduced. This finding provides additional experimental support for the conceptual illustration in Figure 1.

Table 13: Analysis of predictive certainty of different information types on MetaTST.

| Quantile Loss | En. | Ex. and Ex. | En., Ex. and Metadata |
|---|---|---|---|
| Q90 | 0.05207 | 0.05086 | 0.05103 |
| Q10 | 0.04022 | 0.03963 | 0.04032 |
| Interval of difference | 0.01185 | 0.01123 | **0.01071** |

## E  HYPER-PARAMETER ANALYSIS

We conduct a thorough evaluation of the hyperparameter analysis of MetaTST, exploring the impact of three key factors: the number of Transformer blocks ($e_{\text{layers}}$), the hidden dimension ($d_{\text{model}}$), and

the number of attention heads ($n_{\text{heads}}$). Besides, we fix the prediction length of at 24 and vary the look-back length in $\{144, 168, 192, 216\}$ based on the hourly record. Technologically, in Figure 8, we conduct multi-dataset joint training on electricity price datasets and perform zero-short short-term forecasting with different configurations to present the model property of MetaTST.

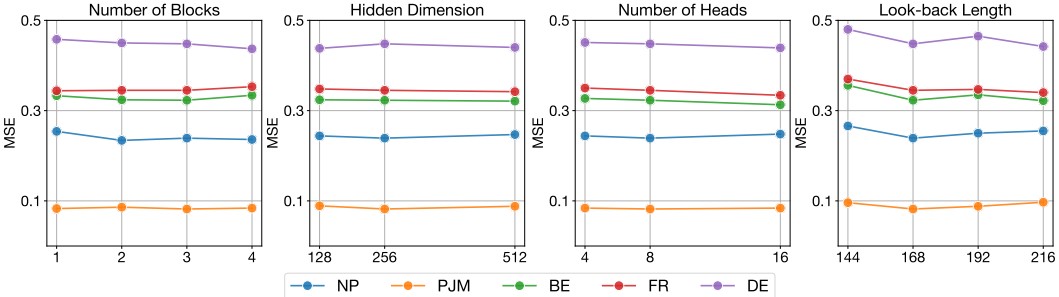

Figure 8: Hyperparameter analysis on the number of Transformer blocks ($e_{\text{layers}}$), the hidden dimension ($d_{\text{model}}$), the number of attention heads ($n_{\text{heads}}$), and the look-back length on short-term electricity price forecasting tasks.

## F  STANDARD DEVIATIONS

We repeat the experiment three times on the short-term prediction task in a multi-dataset joint training setting and provide the mean and standard deviation on each dataset to evaluate the robustness of MetaTST as follows:

Table 14: Standard Deviations of MetaTST.

| Experiment | NP | | PJM | | BE | | FR | | DE | |
|---|---|---|---|---|---|---|---|---|---|---|
| | MSE | MAE | MSE | MAE | MSE | MAE | MSE | MAE | MSE | MAE |
| No. 1 | 0.237 | 0.263 | 0.081 | 0.183 | 0.322 | 0.232 | 0.334 | 0.191 | 0.433 | 0.414 |
| No. 2 | 0.233 | 0.263 | 0.083 | 0.184 | 0.323 | 0.233 | 0.332 | 0.263 | 0.440 | 0.416 |
| No. 3 | 0.234 | 0.263 | 0.087 | 0.186 | 0.318 | 0.234 | 0.329 | 0.193 | 0.435 | 0.415 |
| Mean value$_{\pm\text{Standard deviation}}$ | $0.235_{\pm 0.002}$ | $0.263_{\pm 0.000}$ | $0.084_{\pm 0.003}$ | $0.184_{\pm 0.002}$ | $0.321_{\pm 0.003}$ | $0.233_{\pm 0.001}$ | $0.332_{\pm 0.003}$ | $0.191_{\pm 0.002}$ | $0.436_{\pm 0.004}$ | $0.415_{\pm 0.001}$ |

## G  METEOROLOGY FORECASTING

### G.1  SUPERVISED FORECASTING AND FAST ADAPTATION

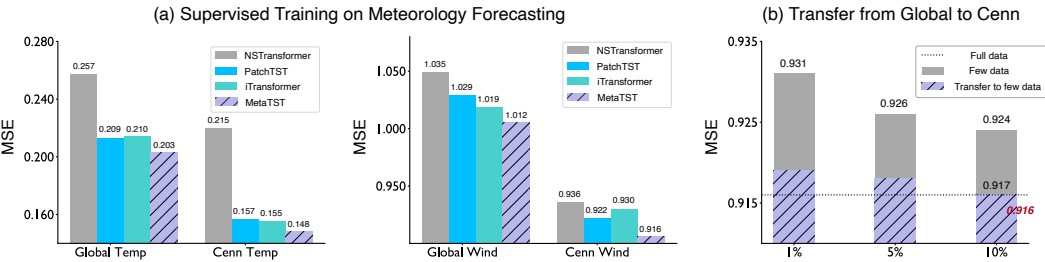

Figure 9: Comparison on meteorology forecasting (a) Supervised training with full data. (b) Transfer and fast adaptation. The input and prediction lenght are set 168 and 72 adhere to Wang et al..

We compare MetaTST with several advanced time series baselines on a large-scale meteorological forecasting dataset with exogenous series, as proposed by Wang et al.. This dataset includes an endogenous series of hourly temperature and wind data from 3,850 global stations and 2,500 local area stations. The exogenous series are meteorological indicators from the surrounding 3×3 grid

areas. Each region provides four types of information: temperature, pressure, and the u- and v-component of temperature and wind. We perform standard supervised training on four meteorological forecasting datasets: Global Temp, Global Wind, Cenn Temp, and Cenn Wind. The models are trained and evaluated on each respective dataset using the full data. As shown in Figure 9(a), MetaTST consistently outperforms other advanced models across all four meteorological forecasting tasks in the supervised training with full data. It's worth noting that the cold start problem is particularly significant in meteorological forecasting, where accurate predictions from newly established weather stations are challenging due to insufficient data. Thus we design another transfer experiments, training models on global wind data with full data and fine-tuning them in local regions using few-shot data (Global Wind → Cenn Wind). We found that MetaTST achieves comparable performance using only 10% of the downstream local area wind data compared to training with the full Cenn Wind dataset (*MSE*: 0.916 vs. 0.917) in Figure 9(b). This reinforces the design that by incorporating context-specific metadata, the model can learn transferable temporal variations from global wind data and rapidly adapt to the specific forecasting context of local stations, thus achieving effective predictions with fewer training samples.

## G.2 REPRESENTATION ANALYSIS OF METADATA IN METEOROLOGICAL FORECASTING

To further validate the impact of LLM in the representation learning of metadata, we design an interesting representation analysis experiment within a meteorological forecasting scenario. Specifically, we separately incorporate various meticulously crafted metadata descriptions and observe the distribution of their corresponding linguistic representations: (a) Basic metadata description with station numbers; (b) Basic metadata description with station numbers, latitude, longitude, and altitude; (c) Basic metadata description with station number, latitude, longitude, and climate zone.

As shown in Figure 10, we clearly observe that as more specific station information is progressively introduced into the metadata, the language model increasingly tends to classify the described information. This phenomenon effectively demonstrates that by incorporating useful context-specific information to construct metadata and leveraging the powerful representation capabilities of LLMs, we generate representations enriched with prior knowledge that benefit specific prediction scenarios.

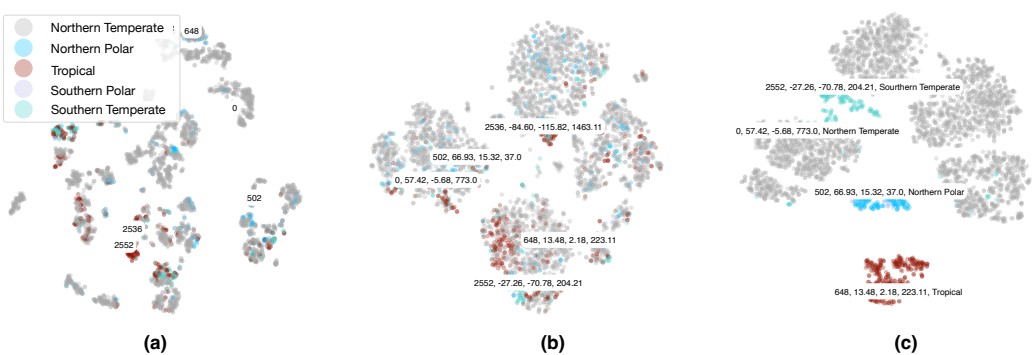

Figure 10: Representation analysis of different metadata in meteorology forecasting scenarios.

## H EXTENDING TO MULTIVARIATE FORECASTING SETTING

MetaTST is designed for informative forecasting by incorporating metadata along with exogenous series into the prediction of endogenous series, indicating its generalizability to multivariate forecasting tasks. By employing a channel independence mechanism, MetaTST can be seamlessly adapted to handle multivariate forecasting scenarios, where each variable can be treated as an endogenous variable while other variables are exogenous. In this section, we evaluate MetaTST on well-established public benchmarks for conventional multivariate long-term forecasting. As shown in Table 18, MetaTST achieves consistent state-of-the-art performance, underscoring its effectiveness and generalizability.

## I    CASE STUDIES

We present more case studies of the informative predictions in MetaTST with raw endogenous series (*En.*), exogenous series (*Ex.*), language-based metadata (*Meta*), and the corresponding learned attention maps. Informative prediction design enables MetaTST to learn discriminative attention maps that adapt to various temporal patterns and prediction scenarios in a multi-dataset joint training setting. See Figure 12 for more visualization cases.

## J    SHOWCASES

To visually compare different models, we present performance showcases for both short- and long-term prediction tasks in Figure 13 and Figure 14. These comparisons are conducted in a multi-dataset joint training setting, using a unified model configuration. In addition, we include visualizations of MetaTST's performance in individual training scenarios. The results demonstrate that the metadata-guided MetaTST predicts future values efficiently in both joint and individual training settings. It is important to note that MetaTST shows consistently more accurate predictions than other state-of-the-art time series models, especially for more challenging long-term predictions. This highlights our design that the metadata-guided informative prediction is crucial for models to adapt to diverse contexts and effectively learn temporal variations in large-scale, multi-dataset training scenarios. This enables models to achieve more deterministic predictions in dynamically changing training contexts, resulting in exceptional predictive performance.

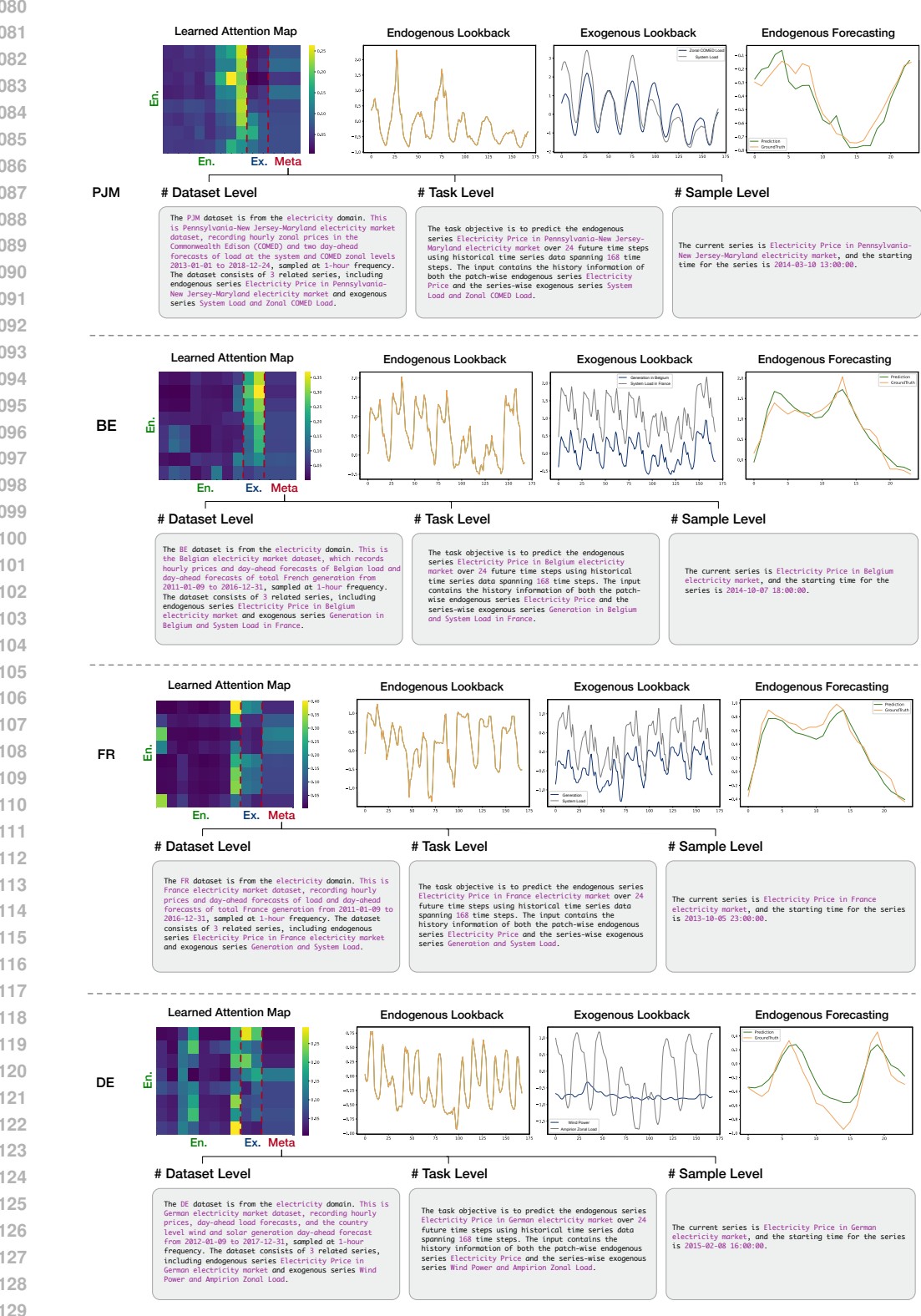

Figure 11: Showcases of short-term forecasting under the input-168-predict-24 setting, including raw endogenous series (*En.*), exogenous series (*Ex.*), language-based metadata (*Meta*) from the NP and PJM datasets, and the learned attention maps in MetaTST.

---

**Weather Metadata Example**

**# Dataset-Level**

The Weather dataset is from the Weather domain. This is a weather dataset, recording 21 meteorological indicators obtained from the Weather Station on Top of the Roof of the Institute Building of the Max-Planck-Institute Institute during 2020 whole year, sampled at a frequency of 10 minutes. The dataset Weather consists of 21 variables. The relation between variable dimension and variable description is as follows, 0. air pressure, 1. air temperature, 2. potential temperature, 3. dew point temperature, 4. relative humidity, 5. saturation water vapor pressure, 6. actual water vapor pressure, 7. water vapor pressure deficit, 8. specific humidity, 9. water vapor concentration, 10. air density, 11. wind velocity, 12. maximum wind velocity, 13. wind direction, 14. precipitation, 15. duration of precipitation, 16. short wave downward radiation, 17. photosynthetically active radiation, 18. maximum photosynthetically active radiation, 19. internal logger temperature, 20. $CO_2$-concentration of ambient air.

**# Task-Level**

The current variable is 20. $CO_2$-concentration of ambient air. The objective of this task is to forecast the target variable 20. $CO_2$-concentration of ambient air over 96 future time steps using historical time series data spanning 96 time steps. The input time series contains the history information of both the patch-wise target variable 20. $CO_2$-concentration of ambient air and the series-wise exogenous variables 0. air pressure, 1. air temperature, 2. potential temperature, 3. dew point temperature, 4. relative humidity, 5. saturation water vapor pressure, 6. actual water vapor pressure, 7. water vapor pressure deficit, 8. specific humidity, 9. water vapor concentration, 10. air density, 11. wind velocity, 12. maximum wind velocity, 13. wind direction, 14. precipitation, 15. duration of precipitation, 16. short wave downward radiation, 17. photosynthetically active radiation, 18. maximum photosynthetically active radiation, 19. internal logger temperature.

**# Sample-Level**

The current variable is 20. $CO_2$-concentration of ambient air and the starting time for the current time window is 2020-09-10 00:00:00

---

**ETTm1 Metadata Example**

**# Dataset-Level**

The ETTm1 dataset is from the Electricity domain. This is an electricity transformer temperature dataset, recording electrical transformers' oil temperature and corresponding external power load features in a region in China between July 2016 and July 2018, sampled at a frequency of 15 minutes. The dataset ETTm1 consists of 7 variables. The relation between variable dimension and variable description is as follows, 0. High UseFul Load, 1. High UseLess Load, 2. Middle UseFul Load, 3. Middle UseLess Load, 4. Low UseFul Load, 5. Low UseLess Load, 6. Oil Temperature.

**# Task-Level**

The current variable is 6. Oil Temperature. The objective of this task is to forecast the target variable 6. Oil Temperature over 96 future time steps using historical time series data spanning 96 time steps. The input time series contains the history information of both the patch-wise target variable 6. Oil Temperature and the series-wise exogenous variables 0. High UseFul Load, 1. High UseLess Load, 2. Middle UseFul Load, 3. Middle UseLess Load, 4. Low UseFul Load, 5. Low UseLess Load.

**# Sample-Level**

The current variable is 6. Oil Temperature and the starting time for the current time window is 2017-12-09 19:00:00.

---

**Traffic Metadata Example**

**# Dataset-Level**

The Traffic dataset is from the trasnportation domain. This is a traffic dataset, recording hourly road occupancy rate measured by 862 sensors on San Francisco Bay area freeways from January 2015 to December 2016. The data is range from 0 to 1, sampled at a frequency of 1 hour. The dataset Traffic consists of 862 variables. The variable dimension is as follows, 0 to 861.

**# Task-Level**

The current variable is 861. The objective of this task is to forecast the target variable 861 over 96 future time steps using historical time series data spanning 96 time steps. The input time series contains the history information of both the patch-wise target variable 861 and the series-wise exogenous variables 0 to 860.

**# Sample-Level**

The current variable is 861 and the starting time for the current time window is 2015-12-09 15:00:00.

---

**ECL Metadata Example**

**# Dataset-Level**

The ECL dataset is from the electricity domain. This is a electricity consuming load dataset, recording the hourly electricity consumption (Kwh) of 321 customers collected from 2016/7/1 2am to 2019/7/2 1 am, sampled at a frequency of 1 hour. The dataset Traffic consists of 321 variables. The variable dimension is as follows, 0 to 320.

**# Task-Level**

The current variable is 320. The objective of this task is to forecast the target variable 320 over 96 future time steps using historical time series data spanning 96 time steps. The input time series contains the history information of both the patch-wise target variable 320 and the series-wise exogenous variables 0 to 319.

**# Sample-Level**

The current variable is 320 and the starting time for the current time window is 2018-11-10 12:00:00.

---

Figure 12: Metadata cases of long-term forecasting tasks.

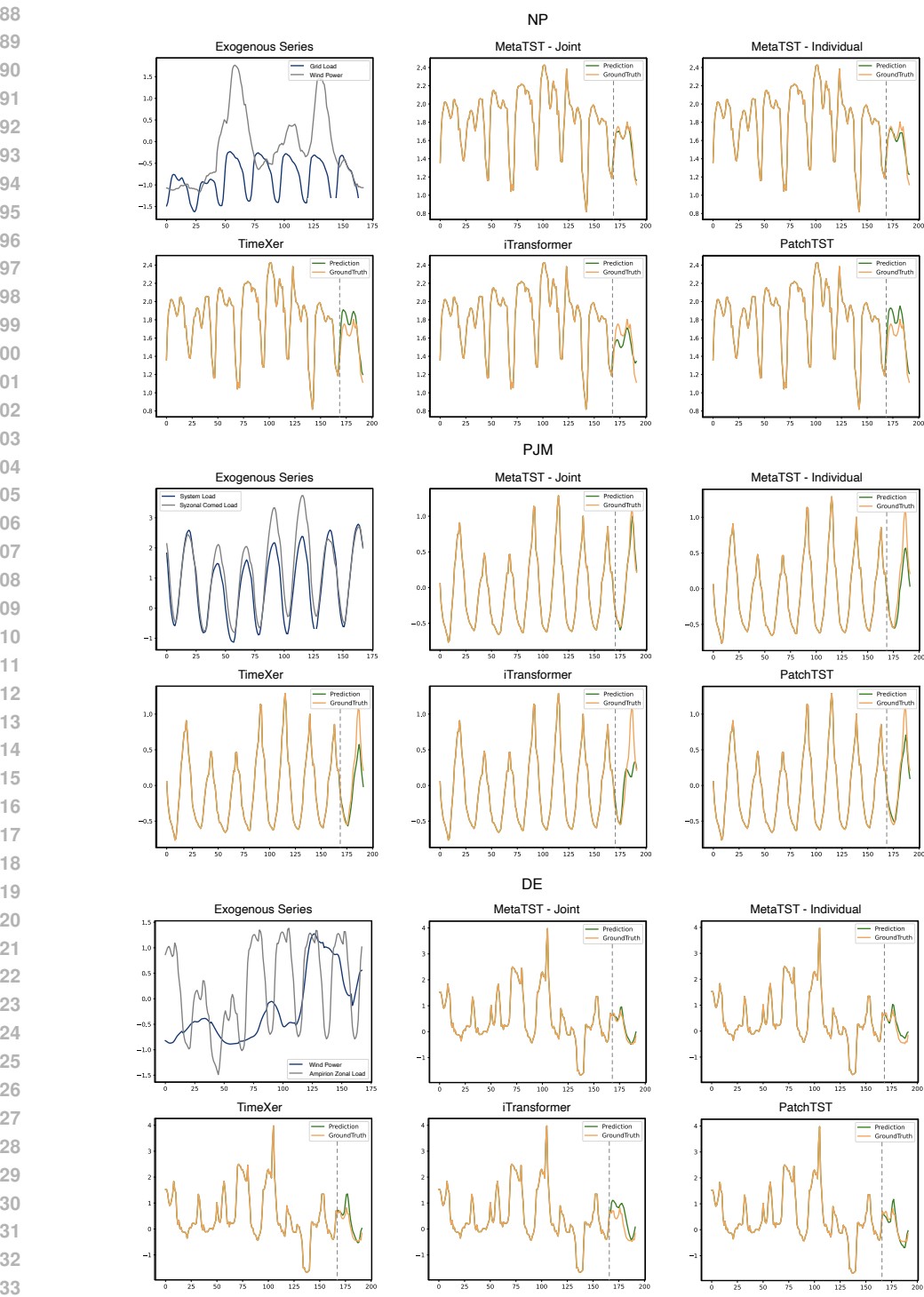

Figure 13: Visualization of short-term forecasting for NP, PJM, and DE predictions by different models under the input-168-predict-24 setting. The gray and blue lines stand for related exogenous series. The orange lines stand for the ground truth and the green lines stand for predicted values.

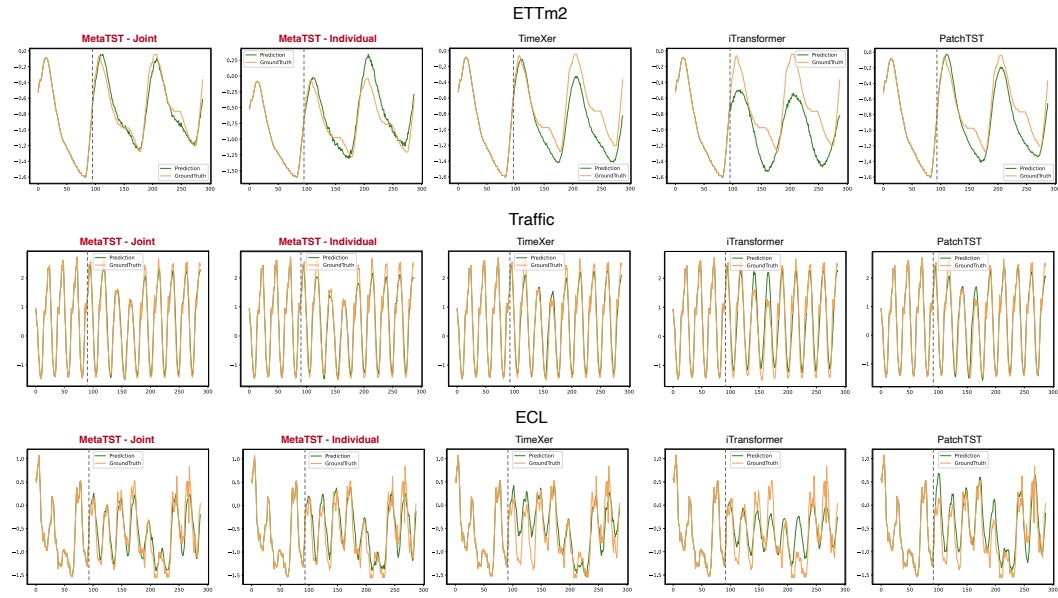

Figure 14: Visualization of long-term forecasting for ETTm2, Traffic, and ECL predictions by different models under the input-96-predict-192 setting. We did not visualize the exogenous series because of the inconsistencies in the numbers and types of exogenous variables in different datasets. The orange lines stand for the ground truth and the green lines stand for predicted values.

## K  FULL RESULTS

Due to the limited length of the text, we list the full results of the main experiments as follows:

Table 15: Full results of different token aggregating methods.

| Design | NP | | PJM | | BE | | FR | | DE | | Avg. | |
|---|---|---|---|---|---|---|---|---|---|---|---|---|
| | MSE | MAE | MSE | MAE | MSE | MAE | MSE | MAE | MSE | MAE | MSE | MAE |
| Routers-3 | 0.240 | 0.266 | 0.092 | 0.190 | 0.333 | 0.239 | 0.353 | 0.198 | 0.448 | 0.420 | 0.293 | 0.263 |
| Routers-6 | 0.242 | 0.268 | 0.091 | 0.192 | 0.324 | 0.236 | 0.346 | 0.196 | **0.434** | **0.414** | 0.287 | 0.261 |
| Routers-12 | 0.242 | 0.267 | **0.087** | 0.189 | 0.321 | 0.237 | 0.339 | 0.195 | 0.446 | 0.417 | 0.287 | 0.261 |
| Global Token | 0.247 | 0.268 | 0.089 | 0.187 | 0.324 | 0.235 | 0.343 | **0.193** | 0.436 | 0.416 | 0.288 | 0.260 |
| **Average Pooling** | **0.234** | **0.263** | **0.087** | **0.186** | **0.318** | **0.234** | **0.329** | **0.193** | 0.435 | 0.415 | **0.281** | **0.258** |

Table 16: Full results of the long-term forecasting with exogenous series under single-dataset individual training.

| Models | | MetaTST | | TimeLLM | | GPT4TS | | TimeXer | | Timer | | iTransformer | | PatchTST | | Crossformer | | TimesNet | | DLinear | | Autoformer | |
|---|---|---|---|---|---|---|---|---|---|---|---|---|---|---|---|---|---|---|---|---|---|---|---|
| Metric | | MSE | MAE | MSE | MAE | MSE | MAE | MSE | MAE | MSE | MAE | MSE | MAE | MSE | MAE | MSE | MAE | MSE | MAE | MSE | MAE | MSE | MAE |
| ECL | 96 | 0.250 | 0.365 | 0.315 | 0.396 | 0.288 | 0.384 | 0.265 | 0.370 | 0.275 | 0.370 | 0.299 | 0.403 | 0.339 | 0.412 | 0.265 | 0.364 | 0.342 | 0.437 | 0.387 | 0.451 | 0.432 | 0.502 |
| | 192 | 0.279 | 0.379 | 0.324 | 0.378 | 0.319 | 0.399 | 0.317 | 0.399 | 0.320 | 0.396 | 0.321 | 0.413 | 0.361 | 0.425 | 0.313 | 0.390 | 0.384 | 0.461 | 0.365 | 0.436 | 0.492 | 0.492 |
| | 336 | 0.330 | 0.415 | 0.381 | 0.403 | 0.378 | 0.436 | 0.371 | 0.429 | 0.392 | 0.439 | 0.379 | 0.446 | 0.393 | 0.440 | 0.380 | 0.431 | 0.439 | 0.493 | 0.391 | 0.453 | 0.508 | 0.548 |
| | 720 | 0.372 | 0.450 | 0.439 | 0.474 | 0.462 | 0.496 | 0.363 | 0.438 | 0.467 | 0.494 | 0.461 | 0.504 | 0.482 | 0.507 | 0.418 | 0.463 | 0.473 | 0.514 | 0.428 | 0.487 | 0.547 | 0.569 |
| | Avg. | **0.308** | **0.402** | 0.365 | 0.413 | 0.362 | 0.429 | 0.329 | 0.409 | 0.364 | 0.425 | 0.365 | 0.442 | 0.394 | 0.446 | 0.344 | 0.412 | 0.410 | 0.476 | 0.393 | 0.457 | 0.495 | 0.528 |
| Weather | 96 | 0.001 | 0.026 | 0.001 | 0.025 | 0.001 | 0.028 | 0.001 | 0.026 | 0.002 | 0.030 | 0.001 | 0.026 | 0.001 | 0.027 | 0.004 | 0.048 | 0.002 | 0.029 | 0.006 | 0.062 | 0.007 | 0.066 |
| | 192 | 0.001 | 0.028 | 0.003 | 0.028 | 0.001 | 0.030 | 0.002 | 0.030 | 0.002 | 0.033 | 0.002 | 0.029 | 0.002 | 0.030 | 0.005 | 0.053 | 0.002 | 0.031 | 0.006 | 0.066 | 0.007 | 0.061 |
| | 336 | 0.002 | 0.029 | 0.004 | 0.046 | 0.002 | 0.032 | 0.002 | 0.031 | 0.002 | 0.034 | 0.002 | 0.031 | 0.002 | 0.032 | 0.004 | 0.051 | 0.002 | 0.031 | 0.006 | 0.068 | 0.007 | 0.062 |
| | 720 | 0.002 | 0.033 | 0.005 | 0.044 | 0.002 | 0.036 | 0.002 | 0.035 | 0.003 | 0.039 | 0.002 | 0.036 | 0.002 | 0.036 | 0.007 | 0.067 | 0.381 | 0.368 | 0.007 | 0.070 | 0.005 | 0.053 |
| | Avg. | **0.002** | **0.029** | 0.003 | 0.036 | **0.002** | 0.031 | **0.002** | 0.031 | **0.002** | 0.034 | **0.002** | 0.031 | **0.002** | 0.031 | 0.005 | 0.055 | 0.097 | 0.115 | 0.006 | 0.066 | 0.006 | 0.060 |
| ETTh1 | 96 | 0.055 | 0.179 | 0.057 | 0.180 | 0.056 | 0.179 | 0.056 | 0.179 | 0.055 | 0.178 | 0.057 | 0.183 | 0.055 | 0.178 | 0.133 | 0.297 | 0.059 | 0.188 | 0.065 | 0.188 | 0.119 | 0.263 |
| | 192 | 0.070 | 0.202 | 0.075 | 0.210 | 0.073 | 0.206 | 0.071 | 0.205 | 0.073 | 0.208 | 0.074 | 0.209 | 0.072 | 0.206 | 0.232 | 0.409 | 0.080 | 0.217 | 0.088 | 0.222 | 0.132 | 0.286 |
| | 336 | 0.077 | 0.217 | 0.087 | 0.231 | 0.088 | 0.233 | 0.080 | 0.205 | 0.088 | 0.233 | 0.084 | 0.223 | 0.087 | 0.231 | 0.244 | 0.423 | 0.083 | 0.224 | 0.110 | 0.257 | 0.126 | 0.278 |
| | 720 | 0.072 | 0.214 | 0.091 | 0.238 | 0.090 | 0.236 | 0.084 | 0.229 | 0.108 | 0.259 | 0.084 | 0.229 | 0.098 | 0.247 | 0.530 | 0.660 | 0.083 | 0.231 | 0.202 | 0.371 | 0.143 | 0.299 |
| | Avg. | **0.069** | **0.203** | 0.077 | 0.215 | 0.077 | 0.214 | 0.073 | 0.209 | 0.081 | 0.220 | 0.075 | 0.211 | 0.078 | 0.215 | 0.285 | 0.447 | 0.076 | 0.215 | 0.116 | 0.259 | 0.130 | 0.282 |
| ETTh2 | 96 | 0.129 | 0.276 | 0.137 | 0.285 | 0.130 | 0.277 | 0.132 | 0.280 | 0.137 | 0.288 | 0.137 | 0.287 | 0.136 | 0.285 | 0.261 | 0.413 | 0.159 | 0.310 | 0.135 | 0.282 | 0.184 | 0.335 |
| | 192 | 0.176 | 0.333 | 0.191 | 0.345 | 0.178 | 0.330 | 0.181 | 0.333 | 0.177 | 0.336 | 0.187 | 0.341 | 0.185 | 0.337 | 1.240 | 1.028 | 0.196 | 0.351 | 0.188 | 0.335 | 0.214 | 0.364 |
| | 336 | 0.210 | 0.360 | 0.227 | 0.383 | 0.217 | 0.371 | 0.224 | 0.378 | 0.197 | 0.361 | 0.221 | 0.376 | 0.217 | 0.373 | 0.974 | 0.874 | 0.232 | 0.385 | 0.238 | 0.385 | 0.269 | 0.405 |
| | 720 | 0.211 | 0.370 | 0.242 | 0.396 | 0.232 | 0.386 | 0.220 | 0.376 | 0.233 | 0.390 | 0.253 | 0.403 | 0.229 | 0.384 | 1.633 | 1.177 | 0.254 | 0.403 | 0.336 | 0.475 | 0.303 | 0.440 |
| | Avg. | **0.182** | **0.335** | 0.199 | 0.352 | 0.189 | 0.341 | 0.189 | 0.342 | 0.186 | 0.344 | 0.199 | 0.352 | 0.192 | 0.345 | 1.027 | 0.873 | 0.210 | 0.362 | 0.224 | 0.369 | 0.242 | 0.386 |
| ETTm1 | 96 | 0.028 | 0.124 | 0.029 | 0.128 | 0.029 | 0.125 | 0.028 | 0.125 | 0.030 | 0.131 | 0.029 | 0.128 | 0.029 | 0.126 | 0.171 | 0.355 | 0.029 | 0.128 | 0.034 | 0.135 | 0.097 | 0.251 |
| | 192 | 0.043 | 0.158 | 0.044 | 0.160 | 0.043 | 0.158 | 0.043 | 0.158 | 0.045 | 0.162 | 0.045 | 0.163 | 0.045 | 0.160 | 0.293 | 0.474 | 0.044 | 0.160 | 0.055 | 0.173 | 0.062 | 0.197 |
| | 336 | 0.056 | 0.183 | 0.057 | 0.185 | 0.057 | 0.184 | 0.057 | 0.185 | 0.059 | 0.186 | 0.060 | 0.190 | 0.058 | 0.184 | 0.330 | 0.503 | 0.061 | 0.190 | 0.078 | 0.210 | 0.083 | 0.230 |
| | 720 | 0.078 | 0.216 | 0.081 | 0.219 | 0.080 | 0.218 | 0.079 | 0.217 | 0.079 | 0.215 | 0.079 | 0.218 | 0.082 | 0.221 | 0.852 | 0.861 | 0.083 | 0.223 | 0.098 | 0.234 | 0.100 | 0.245 |
| | Avg. | **0.051** | **0.170** | 0.053 | 0.173 | 0.052 | 0.171 | 0.052 | 0.171 | 0.053 | 0.173 | 0.053 | 0.175 | 0.053 | 0.173 | 0.411 | 0.548 | 0.054 | 0.175 | 0.066 | 0.188 | 0.085 | 0.230 |
| ETTm2 | 96 | 0.064 | 0.182 | 0.072 | 0.196 | 0.068 | 0.185 | 0.066 | 0.186 | 0.075 | 0.204 | 0.071 | 0.194 | 0.068 | 0.188 | 0.149 | 0.309 | 0.073 | 0.200 | 0.072 | 0.195 | 0.133 | 0.282 |
| | 192 | 0.099 | 0.234 | 0.103 | 0.239 | 0.101 | 0.235 | 0.100 | 0.235 | 0.109 | 0.249 | 0.108 | 0.247 | 0.100 | 0.236 | 0.686 | 0.740 | 0.106 | 0.247 | 0.105 | 0.240 | 0.143 | 0.294 |
| | 336 | 0.130 | 0.273 | 0.133 | 0.279 | 0.130 | 0.274 | 0.130 | 0.274 | 0.146 | 0.292 | 0.140 | 0.288 | 0.128 | 0.271 | 0.546 | 0.602 | 0.150 | 0.296 | 0.136 | 0.280 | 0.156 | 0.308 |
| | 720 | 0.180 | 0.328 | 0.182 | 0.331 | 0.183 | 0.331 | 0.182 | 0.332 | 0.227 | 0.375 | 0.188 | 0.340 | 0.185 | 0.335 | 2.524 | 1.424 | 0.186 | 0.338 | 0.191 | 0.335 | 0.184 | 0.333 |
| | Avg. | **0.118** | **0.254** | 0.122 | 0.261 | 0.120 | 0.256 | 0.120 | 0.257 | 0.139 | 0.280 | 0.127 | 0.267 | 0.120 | 0.258 | 0.976 | 0.769 | 0.129 | 0.271 | 0.126 | 0.263 | 0.154 | 0.305 |
| Traffic | 96 | 0.148 | 0.222 | 0.164 | 0.243 | 0.193 | 0.292 | 0.150 | 0.225 | 0.213 | 0.308 | 0.156 | 0.236 | 0.176 | 0.253 | 0.154 | 0.230 | 0.154 | 0.249 | 0.268 | 0.351 | 0.290 | 0.290 |
| | 192 | 0.144 | 0.225 | 0.182 | 0.252 | 0.185 | 0.285 | 0.152 | 0.228 | 0.244 | 0.346 | 0.156 | 0.237 | 0.162 | 0.243 | 0.180 | 0.256 | 0.164 | 0.255 | 0.302 | 0.387 | 0.291 | 0.291 |
| | 336 | 0.140 | 0.225 | 0.197 | 0.287 | 0.174 | 0.278 | 0.150 | 0.231 | 0.304 | 0.400 | 0.154 | 0.243 | 0.164 | 0.248 | 0.193 | 0.289 | 0.167 | 0.259 | 0.298 | 0.384 | 0.322 | 0.416 |
| | 720 | 0.153 | 0.237 | 0.201 | 0.302 | 0.189 | 0.290 | 0.172 | 0.253 | 0.599 | 0.581 | 0.177 | 0.268 | 0.189 | 0.267 | 0.199 | 0.295 | 0.197 | 0.292 | 0.340 | 0.416 | 0.307 | 0.414 |
| | Avg. | **0.146** | **0.227** | 0.186 | 0.271 | 0.185 | 0.286 | 0.156 | 0.234 | 0.340 | 0.409 | 0.161 | 0.246 | 0.173 | 0.253 | 0.182 | 0.268 | 0.171 | 0.264 | 0.323 | 0.404 | 0.302 | 0.353 |
| Benchmark Avg. | | **0.125** | **0.232** | 0.144 | 0.246 | 0.141 | 0.247 | 0.132 | 0.236 | 166 | 0.269 | 0.140 | 0.246 | 0.145 | 0.246 | 0.461 | 0.482 | 0.164 | 0.269 | 0.179 | 0.287 | 0.202 | 0.306 |

Table 17: Full results of the long-term forecasting with exogenous series under multi-dataset joint training.

| Models | | Joint Training with Specific Datasets | | | | | | | | Individual Training with Mixing Datasets | | | | | | | |
|---|---|---|---|---|---|---|---|---|---|---|---|---|---|---|---|---|---|
| | | MetaTST | | TimeXer | | iTransformer | | PatchTST | | MetaTST | | TimeXer | | iTransformer | | PatchTST | |
| | Metric | MSE | MAE | MSE | MAE | MSE | MAE | MSE | MAE | MSE | MAE | MSE | MAE | MSE | MAE | MSE | MAE |
| ECL | 96 | 0.271 | 0.376 | 0.332 | 0.421 | 0.591 | 0.587 | 0.304 | 0.405 | 0.274 | 0.376 | 0.291 | 0.387 | 0.394 | 0.469 | 0.313 | 0.400 |
| | 192 | 0.304 | 0.393 | 0.357 | 0.427 | 0.564 | 0.569 | 0.343 | 0.414 | 0.307 | 0.397 | 0.335 | 0.407 | 0.425 | 0.481 | 0.349 | 0.417 |
| | 336 | 0.356 | 0.427 | 0.405 | 0.455 | 0.597 | 0.584 | 0.366 | 0.431 | 0.373 | 0.440 | 0.396 | 0.457 | 0.457 | 0.499 | 0.383 | 0.443 |
| | 720 | 0.397 | 0.461 | 0.466 | 0.505 | 0.544 | 0.613 | 0.450 | 0.496 | 0.420 | 0.476 | 0.451 | 0.501 | 0.501 | 0.531 | 0.484 | 0.513 |
| | Avg. | **0.332** | **0.414** | 0.390 | 0.452 | 0.599 | 0.588 | 0.366 | 0.437 | 0.343 | 0.422 | 0.368 | 0.438 | 0.444 | 0.495 | 0.382 | 0.443 |
| Weather | 96 | 0.001 | 0.026 | 0.001 | 0.026 | 0.001 | 0.028 | 0.001 | 0.027 | 0.001 | 0.026 | 0.001 | 0.027 | 0.001 | 0.028 | 0.001 | 0.027 |
| | 192 | 0.001 | 0.029 | 0.002 | 0.029 | 0.002 | 0.031 | 0.002 | 0.003 | 0.002 | 0.029 | 0.002 | 0.029 | 0.002 | 0.003 | 0.002 | 0.029 |
| | 336 | 0.002 | 0.031 | 0.002 | 0.031 | 0.002 | 0.032 | 0.002 | 0.003 | 0.002 | 0.031 | 0.002 | 0.031 | 0.002 | 0.032 | 0.002 | 0.031 |
| | 720 | 0.002 | 0.035 | 0.002 | 0.035 | 0.002 | 0.037 | 0.002 | 0.036 | 0.002 | 0.035 | 0.002 | 0.036 | 0.002 | 0.036 | 0.002 | 0.037 |
| | Avg. | **0.002** | **0.030** | 0.002 | 0.030 | 0.002 | 0.032 | 0.002 | 0.031 | **0.002** | **0.030** | 0.002 | 0.031 | 0.002 | 0.032 | 0.002 | 0.031 |
| ETTh1 | 96 | 0.058 | 0.183 | 0.060 | 0.185 | 0.062 | 0.190 | 0.056 | 0.180 | 0.062 | 0.191 | 0.056 | 0.181 | 0.060 | 0.186 | 0.055 | 0.179 |
| | 192 | 0.073 | 0.206 | 0.073 | 0.208 | 0.080 | 0.219 | 0.075 | 0.209 | 0.074 | 0.208 | 0.073 | 0.207 | 0.077 | 0.214 | 0.073 | 0.208 |
| | 336 | 0.086 | 0.228 | 0.089 | 0.230 | 0.094 | 0.242 | 0.089 | 0.235 | 0.083 | 0.222 | 0.085 | 0.229 | 0.086 | 0.227 | 0.084 | 0.229 |
| | 720 | 0.089 | 0.234 | 0.090 | 0.236 | 0.097 | 0.246 | 0.092 | 0.240 | 0.093 | 0.239 | 0.098 | 0.246 | 0.091 | 0.239 | 0.094 | 0.241 |
| | Avg. | **0.077** | **0.213** | 0.078 | 0.215 | 0.083 | 0.224 | 0.078 | 0.216 | 0.078 | 0.215 | 0.078 | 0.216 | 0.079 | 0.217 | 0.077 | 0.214 |
| ETTh2 | 96 | 0.139 | 0.288 | 0.140 | 0.290 | 0.161 | 0.315 | 0.149 | 0.285 | 0.153 | 0.302 | 0.134 | 0.280 | 0.143 | 0.296 | 0.136 | 0.283 |
| | 192 | 0.186 | 0.339 | 0.189 | 0.340 | 0.208 | 0.362 | 0.187 | 0.343 | 0.181 | 0.334 | 0.185 | 0.348 | 0.189 | 0.344 | 0.186 | 0.338 |
| | 336 | 0.224 | 0.379 | 0.227 | 0.379 | 0.249 | 0.402 | 0.240 | 0.385 | 0.222 | 0.372 | 0.224 | 0.383 | 0.229 | 0.374 | 0.222 | 0.377 |
| | 720 | 0.236 | 0.390 | 0.238 | 0.392 | 0.273 | 0.423 | 0.234 | 0.389 | 0.254 | 0.403 | 0.243 | 0.399 | 0.256 | 0.412 | 0.258 | 0.413 |
| | Avg. | **0.196** | **0.349** | 0.199 | 0.350 | 0.223 | 0.376 | 0.203 | 0.351 | 0.202 | 0.353 | 0.197 | 0.353 | 0.204 | 0.357 | 0.201 | 0.353 |
| ETTm1 | 96 | 0.028 | 0.125 | 0.029 | 0.127 | 0.033 | 0.139 | 0.029 | 0.126 | 0.028 | 0.125 | 0.029 | 0.127 | 0.031 | 0.134 | 0.029 | 0.127 |
| | 192 | 0.043 | 0.158 | 0.044 | 0.160 | 0.047 | 0.166 | 0.043 | 0.158 | 0.045 | 0.161 | 0.045 | 0.161 | 0.047 | 0.167 | 0.044 | 0.159 |
| | 336 | 0.057 | 0.184 | 0.058 | 0.185 | 0.059 | 0.189 | 0.057 | 0.183 | 0.062 | 0.190 | 0.060 | 0.187 | 0.062 | 0.194 | 0.058 | 0.185 |
| | 720 | 0.078 | 0.215 | 0.081 | 0.218 | 0.081 | 0.220 | 0.080 | 0.218 | 0.081 | 0.218 | 0.084 | 0.222 | 0.081 | 0.220 | 0.083 | 0.220 |
| | Avg. | **0.052** | **0.171** | 0.053 | 0.173 | 0.055 | 0.179 | **0.052** | **0.171** | 0.054 | 0.174 | 0.055 | 0.174 | 0.055 | 0.179 | 0.054 | 0.173 |
| ETTm2 | 96 | 0.069 | 0.019 | 0.069 | 0.188 | 0.094 | 0.235 | 0.073 | 0.192 | 0.071 | 0.191 | 0.066 | 0.185 | 0.078 | 0.210 | 0.066 | 0.186 |
| | 192 | 0.103 | 0.239 | 0.113 | 0.251 | 0.121 | 0.267 | 0.111 | 0.240 | 0.112 | 0.248 | 0.100 | 0.235 | 0.110 | 0.252 | 0.099 | 0.234 |
| | 336 | 0.135 | 0.280 | 0.138 | 0.283 | 0.146 | 0.269 | 0.134 | 0.279 | 0.144 | 0.287 | 0.132 | 0.274 | 0.138 | 0.286 | 0.132 | 0.275 |
| | 720 | 0.188 | 0.339 | 0.190 | 0.340 | 0.194 | 0.346 | 0.191 | 0.341 | 0.193 | 0.343 | 0.187 | 0.336 | 0.199 | 0.350 | 0.183 | 0.332 |
| | Avg. | 0.124 | 0.262 | 0.128 | 0.266 | 0.139 | 0.286 | 0.127 | 0.263 | 0.130 | 0.267 | 0.121 | 0.258 | 0.131 | 0.275 | **0.120** | **0.257** |
| Traffic | 96 | 0.167 | 0.254 | 0.203 | 0.293 | 0.564 | 0.576 | 0.241 | 0.323 | 0.175 | 0.259 | 0.191 | 0.282 | 0.238 | 0.331 | 0.196 | 0.288 |
| | 192 | 0.169 | 0.257 | 0.192 | 0.282 | 0.531 | 0.557 | 0.210 | 0.292 | 0.177 | 0.263 | 0.185 | 0.273 | 0.259 | 0.358 | 0.188 | 0.279 |
| | 336 | 0.165 | 0.257 | 0.188 | 0.280 | 0.523 | 0.552 | 0.200 | 0.286 | 0.174 | 0.264 | 0.184 | 0.277 | 0.273 | 0.372 | 0.182 | 0.274 |
| | 720 | 0.184 | 0.277 | 0.208 | 0.297 | 0.544 | 0.562 | 0.221 | 0.306 | 0.193 | 0.283 | 0.201 | 0.291 | 0.310 | 0.399 | 0.199 | 0.290 |
| | Avg. | **0.171** | **0.261** | 0.198 | 0.288 | 0.541 | 0.562 | 0.218 | 0.302 | 0.180 | 0.267 | 0.190 | 0.281 | 0.270 | 0.365 | 0.191 | 0.283 |
| Benchmark Avg. | | **0.136** | **0.243** | 0.150 | 0.253 | 0.235 | 0.321 | 0.149 | 0.253 | 0.141 | 0.247 | 0.144 | 0.250 | 0.169 | 0.274 | 0.147 | 0.250 |

Table 18: Full results of the long-term multivariate forecasting task.

| Models | | MetaTST | | GPT4TS | | Timer | | TimeXer | | iTransformer | | RLinear | | PatchTST | | Crossformer | | TiDE | | TimesNet | | DLinear | | SCINet | | Stationary | | Autoformer | |
|---|---|---|---|---|---|---|---|---|---|---|---|---|---|---|---|---|---|---|---|---|---|---|---|---|---|---|---|---|---|
| Metric | | MSE | MAE | MSE | MAE | MSE | MAE | MSE | MAE | MSE | MAE | MSE | MAE | MSE | MAE | MSE | MAE | MSE | MAE | MSE | MAE | MSE | MAE | MSE | MAE | MSE | MAE | MSE | MAE |
| ECL | 96 | 0.145 | 0.238 | 0.185 | 0.272 | 0.160 | 0.245 | 0.141 | 0.244 | 0.148 | 0.240 | 0.201 | 0.281 | 0.195 | 0.285 | 0.219 | 0.314 | 0.237 | 0.329 | 0.168 | 0.272 | 0.197 | 0.282 | 0.247 | 0.345 | 0.169 | 0.273 | 0.201 | 0.317 |
| | 192 | 0.161 | 0.252 | 0.190 | 0.277 | 0.181 | 0.265 | 0.156 | 0.256 | 0.162 | 0.253 | 0.201 | 0.283 | 0.199 | 0.289 | 0.231 | 0.322 | 0.236 | 0.330 | 0.184 | 0.289 | 0.196 | 0.285 | 0.257 | 0.355 | 0.182 | 0.286 | 0.222 | 0.334 |
| | 336 | 0.179 | 0.271 | 0.204 | 0.292 | 0.208 | 0.290 | 0.173 | 0.272 | 0.178 | 0.269 | 0.215 | 0.298 | 0.215 | 0.305 | 0.246 | 0.337 | 0.249 | 0.344 | 0.198 | 0.300 | 0.209 | 0.301 | 0.269 | 0.369 | 0.200 | 0.304 | 0.231 | 0.338 |
| | 720 | 0.218 | 0.306 | 0.245 | 0.325 | 0.265 | 0.338 | 0.219 | 0.317 | 0.225 | 0.317 | 0.257 | 0.331 | 0.256 | 0.337 | 0.280 | 0.363 | 0.284 | 0.373 | 0.220 | 0.320 | 0.245 | 0.333 | 0.299 | 0.390 | 0.222 | 0.321 | 0.254 | 0.361 |
| | Avg. | 0.176 | **0.267** | 0.206 | 0.291 | 0.203 | 0.285 | **0.172** | 0.272 | 0.178 | 0.270 | 0.219 | 0.298 | 0.216 | 0.304 | 0.244 | 0.334 | 0.251 | 0.344 | 0.192 | 0.295 | 0.212 | 0.300 | 0.268 | 0.365 | 0.193 | 0.296 | 0.227 | 0.338 |
| Weather | 96 | 0.163 | 0.206 | 0.184 | 0.224 | 0.183 | 0.221 | 0.158 | 0.204 | 0.174 | 0.214 | 0.192 | 0.232 | 0.158 | 0.230 | 0.202 | 0.261 | 0.172 | 0.220 | 0.196 | 0.255 | 0.221 | 0.306 | | | 0.173 | 0.223 | 0.266 | 0.336 |
| | 192 | 0.214 | 0.252 | 0.231 | 0.264 | 0.231 | 0.263 | 0.204 | 0.248 | 0.221 | 0.254 | 0.240 | 0.271 | 0.225 | 0.259 | 0.206 | 0.277 | 0.242 | 0.298 | 0.219 | 0.261 | 0.237 | 0.296 | 0.261 | 0.340 | 0.245 | 0.285 | 0.307 | 0.367 |
| | 336 | 0.269 | 0.293 | 0.283 | 0.301 | 0.287 | 0.303 | 0.263 | 0.291 | 0.278 | 0.296 | 0.292 | 0.307 | 0.278 | 0.297 | 0.272 | 0.335 | 0.287 | 0.335 | 0.280 | 0.306 | 0.283 | 0.335 | 0.309 | 0.378 | 0.321 | 0.338 | 0.359 | 0.395 |
| | 720 | 0.347 | 0.346 | 0.361 | 0.350 | 0.368 | 0.355 | 0.343 | 0.345 | 0.358 | 0.349 | 0.364 | 0.353 | 0.354 | 0.348 | 0.398 | 0.418 | 0.351 | 0.386 | 0.365 | 0.359 | 0.345 | 0.381 | 0.377 | 0.427 | 0.414 | 0.410 | 0.419 | 0.428 |
| | Avg. | 0.248 | 0.274 | 0.265 | 0.285 | 0.267 | 0.285 | **0.242** | **0.272** | 0.258 | 0.279 | 0.272 | 0.291 | 0.259 | 0.315 | 0.271 | 0.320 | 0.259 | 0.317 | 0.292 | 0.363 | | | | | 0.288 | 0.314 | 0.338 | 0.382 |
| ETTh1 | 96 | 0.373 | 0.394 | 0.377 | 0.398 | 0.370 | 0.395 | 0.385 | 0.397 | 0.386 | 0.405 | 0.386 | 0.395 | 0.414 | 0.419 | 0.423 | 0.448 | 0.479 | 0.464 | 0.384 | 0.402 | 0.386 | 0.400 | 0.654 | 0.599 | 0.513 | 0.491 | 0.449 | 0.459 |
| | 192 | 0.425 | 0.429 | 0.438 | 0.427 | 0.421 | 0.425 | 0.432 | 0.431 | 0.441 | 0.436 | 0.437 | 0.424 | 0.460 | 0.445 | 0.471 | 0.474 | 0.525 | 0.492 | 0.436 | 0.429 | 0.437 | 0.432 | 0.719 | 0.631 | 0.534 | 0.504 | 0.500 | 0.482 |
| | 336 | 0.455 | 0.447 | 0.469 | 0.449 | 0.471 | 0.449 | 0.463 | 0.447 | 0.487 | 0.458 | 0.479 | 0.446 | 0.501 | 0.466 | 0.570 | 0.546 | 0.565 | 0.515 | 0.491 | 0.469 | 0.481 | 0.459 | 0.778 | 0.659 | 0.588 | 0.535 | 0.521 | 0.496 |
| | 720 | 0.475 | 0.475 | 0.496 | 0.476 | 0.521 | 0.473 | 0.486 | 0.474 | 0.503 | 0.491 | 0.481 | 0.470 | 0.500 | 0.488 | 0.653 | 0.621 | 0.594 | 0.558 | 0.521 | 0.500 | 0.519 | 0.516 | 0.836 | 0.699 | 0.643 | 0.616 | 0.514 | 0.512 |
| | Avg. | **0.432** | 0.436 | 0.445 | 0.437 | 0.446 | 0.436 | 0.441 | 0.437 | 0.454 | 0.447 | 0.446 | **0.434** | 0.469 | 0.454 | 0.529 | 0.522 | 0.541 | 0.507 | 0.458 | 0.450 | 0.456 | 0.452 | 0.747 | 0.647 | 0.570 | 0.537 | 0.496 | 0.487 |
| ETTh2 | 96 | 0.284 | 0.336 | 0.294 | 0.347 | 0.291 | 0.338 | 0.286 | 0.338 | 0.297 | 0.349 | 0.288 | 0.338 | 0.302 | 0.348 | 0.745 | 0.584 | 0.400 | 0.440 | 0.340 | 0.374 | 0.333 | 0.387 | 0.707 | 0.621 | 0.476 | 0.458 | 0.346 | 0.388 |
| | 192 | 0.361 | 0.389 | 0.386 | 0.404 | 0.370 | 0.387 | 0.364 | 0.389 | 0.380 | 0.400 | 0.374 | 0.390 | 0.388 | 0.400 | 0.877 | 0.656 | 0.528 | 0.509 | 0.402 | 0.414 | 0.477 | 0.476 | 0.860 | 0.689 | 0.512 | 0.493 | 0.456 | 0.452 |
| | 336 | 0.402 | 0.420 | 0.423 | 0.435 | 0.422 | 0.427 | 0.414 | 0.426 | 0.428 | 0.432 | 0.415 | 0.426 | 0.426 | 0.433 | 1.043 | 0.731 | 0.643 | 0.571 | 0.452 | 0.452 | 0.594 | 0.541 | 1.000 | 0.744 | 0.552 | 0.551 | 0.482 | 0.486 |
| | 720 | 0.400 | 0.440 | 0.424 | 0.467 | 0.438 | 0.448 | 0.411 | 0.426 | 0.427 | 0.445 | 0.420 | 0.440 | 0.431 | 0.446 | 1.104 | 0.763 | 0.874 | 0.679 | 0.462 | 0.468 | 0.831 | 0.657 | 1.249 | 0.838 | 0.562 | 0.560 | 0.515 | 0.511 |
| | Avg. | **0.362** | **0.394** | 0.382 | 0.408 | 0.380 | 0.400 | 0.369 | 0.397 | 0.383 | 0.407 | 0.374 | 0.398 | 0.387 | 0.407 | 0.942 | 0.684 | 0.611 | 0.550 | 0.414 | 0.427 | 0.559 | 0.515 | 0.954 | 0.723 | 0.526 | 0.516 | 0.450 | 0.459 |
| ETTm1 | 96 | 0.317 | 0.362 | 0.330 | 0.365 | 0.338 | 0.371 | 0.323 | 0.362 | 0.334 | 0.368 | 0.355 | 0.376 | 0.329 | 0.367 | 0.404 | 0.426 | 0.364 | 0.387 | 0.338 | 0.375 | 0.345 | 0.372 | 0.418 | 0.438 | 0.386 | 0.398 | 0.505 | 0.475 |
| | 192 | 0.362 | 0.384 | 0.368 | 0.383 | 0.405 | 0.409 | 0.355 | 0.368 | 0.387 | 0.391 | 0.391 | 0.392 | 0.367 | 0.385 | 0.450 | 0.451 | 0.398 | 0.404 | 0.374 | 0.387 | 0.380 | 0.389 | 0.426 | 0.441 | 0.459 | 0.444 | 0.553 | 0.496 |
| | 336 | 0.386 | 0.402 | 0.401 | 0.404 | 0.481 | 0.447 | 0.396 | 0.405 | 0.426 | 0.420 | 0.424 | 0.415 | 0.420 | 0.405 | 0.532 | 0.515 | 0.428 | 0.425 | 0.410 | 0.411 | 0.413 | 0.413 | 0.445 | 0.459 | 0.495 | 0.464 | 0.621 | 0.537 |
| | 720 | 0.454 | 0.439 | 0.461 | 0.439 | 0.642 | 0.510 | 0.454 | 0.442 | 0.491 | 0.459 | 0.487 | 0.450 | 0.454 | 0.439 | 0.666 | 0.589 | 0.487 | 0.461 | 0.478 | 0.450 | 0.474 | 0.453 | 0.595 | 0.550 | 0.585 | 0.516 | 0.671 | 0.561 |
| | Avg. | **0.380** | **0.387** | 0.390 | 0.398 | 0.466 | 0.434 | 0.385 | 0.400 | 0.407 | 0.410 | 0.414 | 0.407 | 0.387 | 0.400 | 0.513 | 0.496 | 0.419 | 0.419 | 0.400 | 0.406 | 0.403 | 0.407 | 0.485 | 0.481 | 0.481 | 0.456 | 0.588 | 0.517 |
| ETTm2 | 96 | 0.169 | 0.255 | 0.178 | 0.264 | 0.177 | 0.260 | 0.169 | 0.255 | 0.180 | 0.264 | 0.182 | 0.265 | 0.175 | 0.259 | 0.287 | 0.366 | 0.207 | 0.305 | 0.187 | 0.267 | 0.193 | 0.292 | 0.286 | 0.377 | 0.192 | 0.274 | 0.255 | 0.339 |
| | 192 | 0.237 | 0.300 | 0.245 | 0.306 | 0.245 | 0.305 | 0.237 | 0.300 | 0.250 | 0.309 | 0.246 | 0.304 | 0.241 | 0.302 | 0.414 | 0.492 | 0.290 | 0.364 | 0.249 | 0.309 | 0.284 | 0.362 | 0.399 | 0.445 | 0.280 | 0.339 | 0.281 | 0.340 |
| | 336 | 0.297 | 0.336 | 0.309 | 0.346 | 0.313 | 0.348 | 0.293 | 0.224 | 0.311 | 0.348 | 0.307 | 0.342 | 0.305 | 0.343 | 0.597 | 0.542 | 0.377 | 0.422 | 0.321 | 0.351 | 0.369 | 0.427 | 0.637 | 0.591 | 0.334 | 0.361 | 0.339 | 0.372 |
| | 720 | 0.394 | 0.394 | 0.411 | 0.409 | 0.423 | 0.411 | 0.392 | 0.394 | 0.412 | 0.407 | 0.407 | 0.398 | 0.402 | 0.400 | 1.730 | 1.042 | 0.558 | 0.524 | 0.408 | 0.403 | 0.554 | 0.522 | 0.960 | 0.735 | 0.417 | 0.413 | 0.433 | 0.432 |
| | Avg. | 0.274 | 0.321 | 0.286 | 0.331 | 0.290 | 0.331 | **0.273** | **0.320** | 0.288 | 0.332 | 0.286 | 0.327 | 0.281 | 0.326 | 0.757 | 0.610 | 0.358 | 0.404 | 0.291 | 0.333 | 0.350 | 0.401 | 0.571 | 0.537 | 0.306 | 0.347 | 0.327 | 0.371 |
| Traffic | 96 | 0.426 | 0.277 | 0.468 | 0.308 | 0.411 | 0.264 | 0.429 | 0.264 | 0.395 | 0.268 | 0.649 | 0.389 | 0.462 | 0.295 | 0.522 | 0.290 | 0.805 | 0.493 | 0.593 | 0.321 | 0.650 | 0.396 | 0.788 | 0.499 | 0.612 | 0.338 | 0.613 | 0.388 |
| | 192 | 0.435 | 0.284 | 0.477 | 0.311 | 0.440 | 0.281 | 0.463 | 0.278 | 0.417 | 0.276 | 0.601 | 0.366 | 0.466 | 0.296 | 0.530 | 0.293 | 0.756 | 0.474 | 0.617 | 0.336 | 0.598 | 0.370 | 0.789 | 0.505 | 0.613 | 0.340 | 0.616 | 0.382 |
| | 336 | 0.451 | 0.290 | 0.489 | 0.318 | 0.467 | 0.301 | 0.475 | 0.285 | 0.433 | 0.283 | 0.609 | 0.369 | 0.482 | 0.304 | 0.558 | 0.305 | 0.762 | 0.477 | 0.629 | 0.336 | 0.605 | 0.373 | 0.797 | 0.508 | 0.618 | 0.328 | 0.622 | 0.337 |
| | 720 | 0.269 | 0.303 | 0.522 | 0.333 | 0.530 | 0.348 | 0.520 | 0.305 | 0.467 | 0.302 | 0.647 | 0.387 | 0.514 | 0.322 | 0.589 | 0.328 | 0.719 | 0.449 | 0.640 | 0.350 | 0.645 | 0.394 | 0.841 | 0.523 | 0.653 | 0.355 | 0.660 | 0.408 |
| | Avg. | 0.445 | 0.289 | 0.489 | 0.318 | 0.462 | 0.298 | 0.472 | 0.273 | **0.428** | **0.282** | 0.626 | 0.378 | 0.481 | 0.304 | 0.550 | 0.304 | 0.760 | 0.473 | 0.620 | 0.336 | 0.625 | 0.383 | 0.804 | 0.509 | 0.624 | 0.340 | 0.628 | 0.379 |

