# OpenReview forum: "Metadata Matters for Time Series: Informative Forecasting with Transformers"
_ICLR.cc/2025/Conference — ICLR 2025 Conference Withdrawn Submission_

### Official Review · Reviewer_GUpQ · 2024-10-27

**Soundness:** 3
**Presentation:** 3
**Contribution:** 3
**Rating:** 6
**Confidence:** 3

**Summary:**

This manuscript introduces a framework for time-series forecasting, which integrates dataset-, task-, and sample-level descriptions with endogenous and exogenous series to jointly consider semantic and temporal representations. These representations are concatenated to serve as the inputs of the Transformer, which is different from conventional LLM4TS methods that utilize LLMs as the backbone. The experiments are evaluated on both well-known short-term and long-term time-series forecasting tasks.

**Strengths:**

1. The major motivation of this work is clearly stated. The idea of incorporating context-specific metadata and exogenous series provides clear contributions.
2. The paper is generally well-written and the proposed framework is intuitive with sufficient discussions in the literature review and intuitive paradigms to provide readability.
3. The experiments on various ranges of time-series forecasting support the authors' claim, especially for the potential to serve as the time-series foundation model.

**Weaknesses:**

=== AFTER REBUTTAL ===
The authors has resolved most of my concerns and questions. The remaining concerns lie in the need of sample-level metadata as well as the reason that MetaTST could have effective performance in terms of low-resource scenarios.


1. The proposed method may not be able to generalize to the unseen task that does not have metadata. Maybe it has task-level metadata, but dataset- and sample-level seem to have additional efforts to obtain.
2. The descriptions of the method should be improved. The paper does not describe related details for exogenous series. For example, how are the exogenous series selected for each sequence? How many exogenous series are used for each target series? Do exogenous series contain future records or only historical behaviors? Do the authors use positional encoding when feeding to the Transformer?
3. In my opinion, some experiments and discussions should be further improved. The authors should compare their proposed method with more advanced LLM4TS methods, e.g., [1, 2]. In addition, it would be better to include only w/ En., w/ Ex., and w/ Meta to understand where the improvements come from. As one of the main contributions is the incorporation of metadata, the authors should provide the analysis only including single-level metadata.
5. [Minor] There are some numbers different from the original TimeXer paper, e.g., 0.238 -> 0.236 MSE for NP, 0.211 -> 0.208 MAE for FR, 0.418 -> 0.415 MAE for DE. It is not an issue if the authors replicated the experiments. In addition, the number of the long-term forecasting results is significantly different from those baselines (e.g., Table 3 of TimeXer and also significant difference compared with the TimeLLM paper although the given length is slightly different). Please correct me if I misunderstood anything since I didn't find the descriptions in the main text and appendix.


[1] UNITS: A Unified Multi-Task Time Series Model
[2] Unified Training of Universal Time Series Forecasting Transformers

**Questions:**

1. Regarding sample-level metadata, is it only computed based on the historical records for the target series? Would it be recomputed during forecasting? Given that time-series domains often have periodic patterns (e.g., electricity and weather) [1], I suspect that sample-level could provide an advantage for future forecasting.
2. Could the authors elaborate more on the reason that the embedding processes are different in Eq. 4? How would the performance behave if using PatchEmbed for exogenous series?
3. As the authors state the adjustment in L323 for experiments, do the baselines used in the experiments follow the settings from their papers? I only found the parameters for MetaTST in the appendix, but none for the baselines.

[1] LLM4TS: Aligning Pre-Trained LLMs as Data-Efficient Time-Series Forecasters.

---

> ### Author Response · Authors · 2024-11-20
> **Response to Reviewer GUpQ (part 1)**
>
> We would like to sincerely thank Reviewer GUpQ for providing a very insightful review.
>
> ### About the Weakness
>
> > **W1：** The proposed method may not be able to generalize to the unseen task that does not have metadata. Maybe it has task-level metadata, but dataset- and sample-level seem to have additional efforts to obtain.
>
> Thank you for your valuable insight. We have carefully considered the difficulties of metadata generation in the design of MetaTST. Therefore, **we select only the most fundamental and essential elements to compose the metadata descriptions at each level**. As shown $\underline{\text{in Fig. 2 of the main text}}$, dataset-level metadata includes basic information such as dataset name, domain, and frequency, **which is readily available from the description of the data in public datasets, or from application contexts in real-world scenarios**. For variable descriptions, the raw column information of the data can also be used as a source. In sample-level metadata, **time stamps are an intrinsic property of time series data and statistical metrics can be pre-computed based on the data itself.**
>
> In summary, as we highlighted in $\underline{\text{Lines 193-197 of main text}}$, metadata can be easily obtained and combined with our metadata parsing module to enable straightforward generation of new task-specific and language-based metadata descriptions with minimal effort.
>
> > **W2：** The descriptions of the method should be improved. The paper does not describe related details for exogenous series. For example, how are the exogenous series selected for each sequence? How many exogenous series are used for each target series? Do exogenous series contain future records or only historical behaviors? Do the authors use positional encoding when feeding to the Transformer?
>
> Thank you for your constructive feedback. In fact, **all our prediction tasks, both in the long-term and short-term experimental settings, are designed to follow the design of prediction with the exogenous series established by TimeXer.** **In $\underline{\text{Lines 702-748 and Table 6 of Appendix A}}$, we provide an overview of the detail selection of endogenous and exogenous series for each dataset**.
>
> Since each dataset is constructed from a real-world prediction scenario. Therefore, we stick to the original dataset setup, choosing the target series for the dataset itself as the endogenous series and all other related series as the exogenous series.
>
> **We use only historical endogenous and exogenous series to predict future values of the endogenous series.** Each series is processed through a distinct embedding layer tailored to its type. **Positional encoding is then applied to the patchified endogenous series before the data is fed into a Transformer-based time series model for further processing.**
>
> Thank you for your valuable feedback. We have improved the description by detailing the selection and details of endogenous and exogenous series for each dataset in $\underline{\text{Appendix A}}$ and highlighting the usage of positional encoding in $\underline{\text{Appendix B}}$.

---

> ### Author Response · Authors · 2024-11-20
> **Response to Reviewer GUpQ (part 2)**
>
> > **W3：** In my opinion, some experiments and discussions should be further improved. The authors should compare their proposed method with more advanced LLM4TS methods, e.g., [1, 2]. In addition, it would be better to include only w/ En., w/ Ex., and w/ Meta to understand where the improvements come from. As one of the main contributions is the incorporation of metadata, the authors should provide the analysis only including single-level metadata.
>
> Thank you for your valuable suggestion, and we will include them as our related works in our revision. Following your suggestions, we conduct ablation with different design and add these reuslts in $\underline{\text{Table 9, 10 of Appendix}}$, including w/ En., w/ Ex., w/ Meta, and single-level metadata.
>
> | MSE/MAE                           | BE          | DE          | FR          | NP          | PJM         |
> | --------------------------------- | ----------- | ----------- | ----------- | ----------- | ----------- |
> | W/ En.                            | 0.350/0.241 | 0.465/0.432 | 0.366/0.197 | 0.261/0.275 | 0.092/0.195 |
> | W/ Ex.                            | 0.388/0.270 | 0.487/0.439 | 0.399/0.220 | 0.268/0.296 | 0.092/0.196 |
> | W/ Metadata                       | 0.689/0.461 | 1.130/0.729 | 0.634/0.400 | 0.586/0.492 | 0.269/0.371 |
> | W/ En.+Ex.+Dataset-level metadata | 0.324/0.239 | 0.448/0.423 | 0.341/0.190 | 0.239/0.266 | 0.089/0.190 |
> | W/ En.+Ex.+Task-level metadata    | 0.333/0.240 | 0.460/0.420 | 0.356/0.193 | 0.239/0.266 | 0.090/0.190 |
> | W/ En.+Ex.+Sample-level metadata  | 0.406/0.279 | 0.524/0.473 | 0.432/0.237 | 0.291/0.313 | 0.105/0.209 |
> | MetaTST (Ours)                    | 0.318/0.234 | 0.435/0.415 | 0.329/0.193 | 0.234/0.263 | 0.087/0.186 |
>
> Regarding the baseline comparison you mentioned, **we would like to emphasize that these two methods are not LLM4TS approaches.** Instead, they are large-scale time series models pre-trained on extensive time series data. Their experimental setup and validity validation differ significantly to our work, who can also benefit from large-scale training data. Thus, the comparison between MetaTST and them is unfair.
>
> Further, as we stated in $\underline{\text{Conclusion section}}$, **MetaTST, with its innovative approach of incorporating metadata to enhance time series forecasting, can be used as a foundation backbone in the reviewer mentioned large-scale time series models.** We will leavel this direction as future work.
>
> > **W4：** There are some numbers different from the original TimeXer paper, e.g., 0.238 -> 0.236 MSE for NP, 0.211 -> 0.208 MAE for FR, 0.418 -> 0.415 MAE for DE. It is not an issue if the authors replicated the experiments. In addition, the number of the long-term forecasting results is significantly different from those baselines (e.g., Table 3 of TimeXer and also significant difference compared with the TimeLLM paper although the given length is slightly different). Please correct me if I misunderstood anything since I didn't find the descriptions in the main text and appendix.
>
> **We report the reuslts TimeXer form its previous version (arXiv: [v1] Thu, 29 Feb 2024 11:54:35 UTC)**, which has been updated after our submission. We have revised them in our revision. Regarding the long-term forecasting, we would like to highlight that **we follow the experimental setup in TimeXer and perform long-term forecasting with exogenous variables, the correpoding results are listed $\underline{\text{in Table 11 of TimeXer}}$.**

---

> ### Author Response · Authors · 2024-11-20
> **Response to Reviewer GUpQ (part 3)**
>
> ### About the Questions
>
> > **Q1：** Regarding sample-level metadata, is it only computed based on the historical records for the target series? Would it be recomputed during forecasting? Given that time-series domains often have periodic patterns (e.g., electricity and weather) [1], I suspect that sample-level could provide an advantage for future forecasting.
>
> Your understanding is correct that sample-level metadata is derived only from historical target series, using timestamps or statistical information to distinguish between various samples. For each inference, the sample-level metadata relies solely on the look-back series (e.g. 96 time points for long-term prediction). In fact, the statistical information already exists in the raw data; We simply highlight this statistical information and present it to the LLMs. Therefore, the sample-level metadata does not incorporate any external information than the other baselines.
>
> > **Q2：** Could the authors elaborate more on the reason that the embedding processes are different in Eq. 4? How would the performance behave if using PatchEmbed for exogenous series?
>
> From a design perspective, different embeddings naturally allow for differentiation in the granularity and importance of modeling endogenous and exogenous series. Too many exogenous patches may distract the model's attention on the most important endogenous series.
>
> From a complexity point of view, exogenous series typically outnumber endogenous series, and applying patch embedding to exogenous series significantly increases the computational complexity.
>
> Considering computational efficiency, we chose short-term forecasting tasks for performance validation. As shown in the results, MetaTST achieves superior predictive performance compared to using PatchEmbed for both endogenous and exogenous series.
>
> | MSE/MAE                             | BE          | DE          | FR          | NP          | PJM         |
> | ----------------------------------- | ----------- | ----------- | ----------- | ----------- | ----------- |
> | patch embedding to exogenous series | 0.325/0.242 | 0.464/0.420 | 0.340/0.200 | 0.251/0.271 | 0.093/0.194 |
> | MetaTST (Ours)                      | 0.318/0.234 | 0.435/0.415 | 0.329/0.193 | 0.234/0.263 | 0.087/0.186 |
>
> > **Q3：** As the authors state the adjustment in L323 for experiments, do the baselines used in the experiments follow the settings from their papers? I only found the parameters for MetaTST in the appendix, but none for the baselines.
>
> This is a very insightful question. For standard individual training on a single time series dataset, all results are reported $\underline{\text{in Table 2,3 of main text}}$, with the hyper-parameters searching for each method following the same way as reported in their own papers, resulting in **varying hyper-parameter settings and model sizes across different datasets and prediction lengths. Thus, if we follow their official setting, it is impossible to conduct joint training experiments.**
>
> To ensure a fair and valuable comparison, **we tried our best to construct joint training experiments on these baselines, where we have to use unified hyperparameter settings in all forecasting lengths and datasets**. For $\underline{\text{the results in Tables 4,5 of the main text}}$, **all baseline methods and MetaTST are configured with unified hyper-parameters to ensure them under a comparable model size.** For further details, please refer to $\underline{\text{Table 7 of Appendix}}$.

---

> ### Comment · Reviewer_GUpQ · 2024-11-25
>
> Thank the authors for their feedback! Most of my concerns and questions are resolved, and my remaining concerns lie in the generalizability for the usage of metadata as well as the incorporation of sample-level metadata (and slightly the selected baselines). Therefore, I have increased my score to reflect this.
>
> > A1.
>
> Thanks for the clarification. Although the authors only select the most findamental metadata, my major concern is for the generalizability for the unseen task, where some of the existing time-series forecasting methods can achieve (e.g., [1, 2]). For instance, the statistical metrics would have large variance when using only 5% of training data.
>
> > A3.
>
> Thank you for providing additional results and to point out the difference of the mentioned baselines. Given the ablation study, the results of `W/ En.+Ex.+Sample-level metadata` match my hypothesis as mentioned in Q1 that adding sample-level metadata could not improve the performance compared with `W/o Metadata information` in Table 9.
>
> A follow-up question: In terms of the baselines, is there any reason that the authors do not compare with the methods in [2]? Note that they are not trained with additional datasets.
>
>
> [1] LLM4TS: Aligning Pre-Trained LLMs as Data-Efficient Time-Series Forecasters
>
> [2] Time-LLM: Time Series Forecasting by Reprogramming Large Language Models

---

> ### Author Response · Authors · 2024-11-26
> **Thanks for your response and raising score**
>
> We sincerely thank you for your response and for raising the score. Your feedback means a lot to us.
> Here we further answer your follow-up questions separately.
>
> > About A1: About the generalizability of the model on the few-shot scenario
>
> For the generalizability task you mentioned, we followed the experimental settings from Time-LLM [1] to validate the effectiveness of our method under a 5% few-shot scenario. The specific results are as follows:
>
> | Avg MSE/MAE | MetaTST         | Time-LLM        | LLM4TS      |
> | ----------- | --------------- | --------------- | ----------- |
> | ETTh1       | 0.644/**0.542** | **0.627**/0.543 | 0.651/0.551 |
> | ETTh2       | **0.333/0.386** | 0.382/0.418     | 0.359/0.405 |
> | ETTm1       | **0.406/0.416** | 0.425/0.434     | 0.413/0.417 |
> | ETTm2       | **0.262/0.321** | 0.274/0.323     | 0.286/0.332 |
> | Weather     | **0.231/0.276** | 0.260/0.309     | 0.256/0.292 |
>
> The results show that our method outperforms Time-LLM and LLM4TS overall. This advantage may stem from the fact that **these methods rely solely on the embedding layer of LLMs for language embedding, while our MetaTST utilizes the overall architecture of LLMs for metadata encoding.** The better utilization of LLM in MetaTST allows for more effective handling of complex distribution shifts.
>
> > About A3 [Part-1]: About the effectiveness of sample-level metadata
>
> Excellent remark! First, we want to highlight that our key motivation is **using multi-level metadata**. Focusing only on sample-level metadata may lead to some misunderstandings. Here are the details.
>
> We further validate the experimental results using both dataset-level and task-level metadata in a joint training scenario as follows.
>
> | MSE/MAE                           | BE          | DE          | FR          | NP          | PJM         |
> | --------------------------------- | ----------- | ----------- | ----------- | ----------- | ----------- |
> | En.+Ex. W/o Sample-level metadata | 0.324/0.239 | 0.445/0.419 | 0.332/0.195 | 0.237/0.265 | 0.089/0.188 |
> | En.+Ex. W/ Sample-level metadata  | 0.406/0.279 | 0.524/0.473 | 0.432/0.237 | 0.291/0.313 | 0.105/0.209 |
> | MetaTST (Ours)                    | 0.318/0.234 | 0.435/0.415 | 0.329/0.193 | 0.234/0.263 | 0.087/0.186 |
>
> We observe that removing sample-level metadata (En.+Ex. W/o Sample-level metadata) leads to a drop in performance, highlighting the importance of jointly utilizing multi-level metadata and the specific effectiveness of sample-level metadata.
>
> However, as the reviewer mentioned, only using sample-level metadata (En.+Ex. W/ Sample-level metadata) results in poor prediction performance. This is most likely because, when using only sample-level metadata, similar statistical information across different datasets can make it more difficult to distinguish distinct temporal patterns, especially in more challenging multi-dataset joint training scenarios. This highlights the effectiveness of our design of multi-level metadata.
>
> > About A3 [Part-2]: About the follow-up question of comparing with Time-LLM
>
> Good question! In fact, we have performed a comprehensive comparison with Time-LLM in the single-dataset training setting in $\underline{\text{Tables 2, 3 of the main text}}$, where it was found not to be the most effective single-dataset model, comparing to TimeXer. Therefore, we mainly experiment with TimeXer on the joint training dataset, considering the higher computational cost of Time-LLM.
>
> As per your request, we have added a comparison with Time-LLM in the joint training setting and also provided results in $\underline{\text{A1}}$ for the 5% few-shot scenario. The results show that MetaTST outperforms Time-LLM in terms of prediction performance.
>
> | Joint Training MSE/MAE | BE          | DE          | FR          | NP          | PJM         |
> | ---------------------- | ----------- | ----------- | ----------- | ----------- | ----------- |
> | Time-LLM               | 0.420/0.285 | 0.577/0.486 | 0.428/0.239 | 0.332/0.329 | 0.109/0.213 |
> | MetaTST (Ours)         | 0.318/0.234 | 0.435/0.415 | 0.329/0.193 | 0.234/0.263 | 0.087/0.186 |

---

> > ### Comment · Reviewer_GUpQ · 2024-11-27
> >
> > I appreciate the follow-up responses with additional experiments from the authors.
> >
> > > A1:
> >
> > This is indeed helpful to see the performance of MetaTST in terms of low-resource scenarios, but I was wondering why MetaTST could improve the performance. Could the authors elaborate more on the effective performance of MetaTST? From my understanding, the pre-trained LLM is also trained in LLM4TS (Fig. 2 in their paper). I am also curious about if the sample-level metadata would significantly vary between 5% and full-shot scenarios if the authors stored the metadata (it would be acceptable if not since this is a follow-up question).
> >
> > > A3 [Part-1]:
> >
> > I agree that the contribution lies in the use of multi-level metadata; however, I still have concerns about the incorporation of sample-level metadata, e.g., to what extent sample-level metadata could really help.

---

> > > ### Author Response · Authors · 2024-11-27
> > >
> > > > A3 [Part-1]：I agree that the contribution lies in the use of multi-level metadata; however, I still have concerns about the incorporation of sample-level metadata, e.g., to what extent sample-level metadata could really help.
> > >
> > > In fact, by comparing the results we presented in A3 [Part-1], we can observe that using **only dataset-level and task-level metadata, without sample-level metadata, leads to a performance drop** ($\underline{\text{W/o Sample-level metadata < MetaTST (Ours)}}$). Furthermore, **using only sample-level metadata yields even worse results** ($\underline{\text{Only W/ Sample-level metadata < W/o Sample-level metadata < MetaTST (Ours)}}$). This further demonstrates that **sample-level metadata needs to be used in conjunction with dataset-level and task-level metadata to achieve the desired performance**. Additionally, it is important to note that sample-level metadata not only captures sample statistics but also includes timestamp information of current sample, which is a critical feature in time-series analysis.

---

> ### Author Response · Authors · 2024-11-27
> **Thanks for your insight review**
>
> We sincerely thanks for your patience and valuable feedback.
>
> > A1-1：This is indeed helpful to see the performance of MetaTST in terms of low-resource scenarios, but I was wondering why MetaTST could improve the performance. Could the authors elaborate more on the effective performance of MetaTST? From my understanding, the pre-trained LLM is also trained in LLM4TS (Fig. 2 in their paper).
>
> In fact, **previous LLM for TS models (LLM4TS) [1, 2] primarily rely on the LLMs as their main body. Whether the language model is frozen or trained, the core motivation is to align the time series to the language modality**, attempting to leverage the powerful sequence modeling capabilities of LLMs to enhance time-series forecasting. However, this has been questioned by [3] that **there is no evidence the LLMs successfully transfer sequence modeling abilities from text to time series**.
>
> As stated $\underline{\text{in lines 70-74 of our paper}}$, metadata is critically important for time series forecasting. How effectively to leverage metadata information through LLMs to enhance prediction performance is precisely the central question that MetaTST aims to address. It is worth emphasizing that **the primary framework of MetaTST is a time series model. We only incorporate LLMs as an auxiliary metadata encoder to build informative time series embeddings**. This design offers the following advantages:
>
> 1. **Fully Leveraging the Prior Knowledge of Pre-trained LLMs：**  We use the entire LLM as the encoder for language-based metadata, rather than relying on a simple embedding layer as other LLM4TS methods. This approach enables us to leverage the prior knowledge of pre-trained LLMs more effectively, resulting in more adequate and robust metadata encoding in the input phase of the model.
> 2. **Enhancing Forecasting Performance in Single-Dataset Training Scenarios：** In single-dataset training scenarios, metadata serves as supplementary information to aid in predicting the target variable. Benefiting from the powerful self-attention mechanism, the model can adaptively extract valuable insights from the encoded metadata tokens to enhance the prediction of the target variable. The experimental results $\underline{\text{in Tables 2, 3 of the main text}}$ demonstrate that MetaTST consistently outperforms, even on tasks close to saturation in the optimized hyperparameter setting.
> 3. **The Potential to be a Time Series Foundational Model：** In the multi-dataset joint training scenario, metadata can not only provide external information for prediction, but also effectively distinguish between different prediction contexts. In complex training scenarios where datasets, tasks, and samples vary dynamically, metadata can guide the predictive behavior of the model and prevent confusion caused by similar time series patterns across different training scenarios. This addresses a key challenge in large-scale training for time-series models, and MetaTST offers an effective solution.
>
> The above points highlight why MetaTST could improve the performance and is more effective than previous LLM4TS methods and native time series models, further validating its utility and advantages in real-world applications.
>
> [1] LLM4TS: Aligning Pre-Trained LLMs as Data-Efficient Time-Series Forecasters
>
> [2] Time-LLM: Time Series Forecasting by Reprogramming Large Language Models
>
> [3] Are language models actually useful for time series forecasting?
>
> > A1-2：I am also curious about if the sample-level metadata would significantly vary between 5% and full-shot scenarios if the authors stored the metadata (it would be acceptable if not since this is a follow-up question).
>
> Regarding the issue of sample-level metadata variation, since time-series forecasting tasks are always normalized using instance normalization techniques like RevIN (ICLR, 2022), **the sample-level metadata constructed by MetaTST is derived from the statistics of the normalized time-series samples**. As a result, sample-level metadata generally does not vary significantly between few-shot and full-shot scenarios.

---

> ### Comment · Reviewer_GUpQ · 2024-11-28
>
> Thank the authors for the discussions. It would be better to also include the ablation study of each level metadata as shown above into the paper.
>
> > A1-1
>
> This is not convincing for me since those goals the authors mentioend could be also achieved by existing works. Finetuning LLMs to time-series domains may be debatable, but those works did showcase the effectiveness.
>
> > A1-2
>
> Thanks for sharing this information. If the statistics do not change significantly, this still indicates the potential issue of generalizability. I agree that it may be difficult to find a benchmark to actually reflect this, so I would not consider this as a disadvantage of this paper.
>
> > A3
>
> This sounds confusing. In addition to empirical results (however single-training scenario does not work), I still suspect the motivation of using sample-level metadata. For instance, why and what is the motivation that the sample-level metadata is required with other-level metadata?

---

> ### Author Response · Authors · 2024-11-28
> **Thanks for your insight response**
>
> We sincerely appreciate the reviewer’s constructive insights and suggestions, which have helped us significantly improve the quality of our paper.
>
>
>
> > About A1-1
>
> As you mentioned, fine-tuning LLMs is a subject of debate [1]. However, our approach offers a new, efficient, and effective training strategy for leveraging LLMs. We believe that the idea of "informative time series forecasting" can serve as a valuable supplement to current research. While the effectiveness of previous models has been discussed, we would like to emphasize the advantages of MetaTST in terms of training efficiency. We compare the running time of different models in a single iteration (ms/iter) using the BE dataset, as shown below.
>
>
> | Method                  | MetaTST (Ours) | TimeLLM | GPT4TS |
> | ----------------------- | -------------- | ------- | ------ |
> | Training Time (ms/iter) | 27ms           | 228ms   | 77ms   |
>
> [1] Are language models actually useful for time series forecasting?
>
> > About A3
>
>
> The key contribution of this paper is **proposing "informative time series forecasting" as a new forecasting paradigm**. However, **the lack of sample-level data prevents the model from intuitively recognizing the differences between individual samples**. For further details, please refer to $\underline{\text{Lines 215–225 of the main text}}$.
>
> ​Indeed, in the single-dataset individual training setting, the dataset-level and task-level metadata are consistent across different training samples under the same dataset and task objectives, with the only distinction being the sample-level metadata (including timestamp information). In this context, the results $\underline{\text{in Table 10 of the Appendix}}$ further illustrate the effectiveness of sample-level metadata. For clarity, we have included the primary results below.
>
>
> | Avg MSE/MAE   | ETTh1       | ETTh2       | ETTm1       | ETTm2       | Weather     | Traffic     | ECL         |
> | ------------- | ----------- | ----------- | ----------- | ----------- | ----------- | ----------- | ----------- |
> | W/o Metadata  | 0.077/0.216 | 0.203/0.355 | 0.056/0.178 | 0.138/0.275 | 0.002/0.030 | 0.172/0.262 | 0.335/0.420 |
> | MetaTST(Ours) | 0.069/0.203 | 0.182/0.335 | 0.051/0.170 | 0.118/0.254 | 0.002/0.029 | 0.146/0.227 | 0.308/0.402 |

---

### Official Review · Reviewer_wMyv · 2024-11-04

**Soundness:** 2
**Presentation:** 2
**Contribution:** 2
**Rating:** 5
**Confidence:** 5

**Summary:**

The paper presents the Metadata-informed Time Series Transformer (MetaTST), a model that incorporates metadata in time series forecasting using a novel approach where metadata is embedded as language tokens processed through Large Language Models (LLMs). The proposed model aims to improve forecast accuracy by enriching the context of time series data with structured metadata.

**Strengths:**

Use of Metadata: By integrating metadata through structured natural language tokens, MetaTST significantly enriches the contextual representation of time series data, addressing gaps left by traditional time series models.

LLM Integration as Metadata Encoder: The approach of using LLMs as fixed encoders allows MetaTST to leverage the semantic understanding of LLMs without extensive fine-tuning, thereby maintaining computational efficiency.

Robust Performance: MetaTST achieves state-of-the-art results across several datasets, both in single-dataset and multi-dataset training settings, showcasing its adaptability to various forecasting scenarios.

**Weaknesses:**

Complexity in Implementation: The use of LLMs as metadata encoders and the structured metadata embedding mechanism might increase the model's complexity, which could be challenging for practitioners to implement. The $h_0$ dimension is too high consisting of three components, a better idea is to try cross attention between these components.

Limited Explanation of Metadata Selection: The paper provides limited detail on the criteria for selecting specific metadata fields and their impact on forecasting accuracy, leaving questions about its generalizability to other domains.

Interpretability Challenges: With multiple embedding and alignment processes, understanding the contribution of each metadata token may be challenging, which affects the model's interpretability.

**Questions:**

what are the exogenous series in your experiments? In the paper, the model is trained on mixed data from all datasets and then probed in the target dataset.

The paper's objective is to talk about informative forecasting with metadata extends series representation for higher accurate forecasting. But the performance may be from the exogenous series, there is no ablation study ablating exogenous series.

Detaily, the metadata is the same from TimeLLM, the related news or reports would be preferred.

---

> ### Author Response · Authors · 2024-11-20
> **Response to Reviewer wMyv (part 1)**
>
> We would like to sincerely thank Reviewer wMyv for reviewing and coming up with detailed questions.
>
> ### About the Weakness
>
> > **W1：** Complexity in Implementation: The use of LLMs as metadata encoders and the structured metadata embedding mechanism might increase the model's complexity, which could be challenging for practitioners to implement. The h0 dimension is too high consisting of three components, a better idea is to try cross attention between these components.
>
> MetaTST emphasizes the informative forecasting by introducing context-specific metadata embeddings to improve prediction performance. However, **the LLM-based metadata encoder will be frozen and different levels of metadata representation can be pre-generated and directly loaded during model training, which minimally impacts model complexity.**
>
> Regarding the \(h_0\) dimension issue, metadata embedding is the core contributions of our method. As shown $\underline{\text{in Figure 3 and Line 232 of the main text}}$, MetaTST aggregates word-level metaata representations at different levels into global metadata tokens using a mean aggregation method. As a result, **only three additional metadata tokens are added to the prediction process, which does not significantly increase the model complexity.**
>
> Furthermore, as illustrated $\underline{\text{in Figure 6 (b) of our paper}}$, the efficiency experiments have also demonstrated that **MetaTST outperforms all baselines with favorable efficiency.**
>
> Regarding cross-attention, TimeXer first attempts to use cross-attention between exogenous and endogenous series, and cross-attention based router approaches for aggregating word-level meta-representations have also been validated in our paper, **all of which perform less effectively compared to the current design of MetaTST.**
>
> > **W2：** Limited Explanation of Metadata Selection: The paper provides limited detail on the criteria for selecting specific metadata fields and their impact on forecasting accuracy, leaving questions about its generalizability to other domains.
>
> Sorry for missing the explanation. In fact, we do not implement any specific filtering of the metadata; Instead, we strive to incorporate as much metadata information as possible, including domain information, variable descriptions, and sampling frequencies. Due to the unstructured nature of metadata, we develop a textual template to effectively organize all relevant information.
>
> Regarding your concerns about generalizability, we believe our approach demonstrates strong potential for applicability across various domains and datasets. Metadata, by definition, provides essential context and external information about primary data sources, and it is inherently present in any real-world scenario involving time series data.
>
> > **W3：** Interpretability Challenges: With multiple embedding and alignment processes, understanding the contribution of each metadata token may be challenging, which affects the model's interpretability.
>
> First, **none of the previous methods based on the design of LLMs have performed an interpretability analysis, so interpretability should not be considered as a unique weakness of our approach.**
>
> Moreover, we have performed a detailed visual analysis to validate the contribution of the various meta-tokens.
>
> **(1) Dataset-level token**: $\underline{\text{In Figure 5 (b) of the main text}}$, the metadata representations from different datasets are separated from each other, while representations from similar datasets are closer, demonstrating the value of dataset-level metadata representation.
>
> **(2) Task-level token:** $\underline{\text{In Figure 10  of the Appendix}}$, task tokens exhibit clustering properties based on latitude and longitude, indicating that task-level metadata can provide reasonable priors.
>
> **(3) Sample-leve token:** $\underline{\text{in Figure 5 (b) of the main text}}$, different samples within the same dataset are separated from one another, indicating that sample-level metadata can capture distinctions between samples.
>
> Since our work is not theoretical, we can only provide experimental insights into interpretability. We strongly believe that our design in **adopting LLM for text encoding is much more interpretable than previous usage of fine-tuning the LLM parameters**. Thus, we respectively suggest the reviewer can reconsider this issue.

---

> ### Author Response · Authors · 2024-11-20
> **Response to Reviewer wMyv (part 2)**
>
> ### About the Question
>
> > **Q1：** what are the exogenous series in your experiments? In the paper, the model is trained on mixed data from all datasets and then probed in the target dataset.
>
> We follow the experimental settings established in the advanced time series forecasting model with exogenous series, TimeXer. **We also provide detailed descriptions and choose of the endogenous and exogenous series for different datasets in Lines 702-744 and Table 6**. It is worth to note that we will keep the same setting covering all individual single dataset and joint multiple datasets training scenarios. More details refer to $\underline{\text{Appendix A of our paper}}$.
>
> > **Q2：** The paper's objective is to talk about informative forecasting with metadata extends series representation for higher accurate forecasting. But the performance may be from the exogenous series, there is no ablation study ablating exogenous series.
>
> **In $\underline{\text{Lines 429-448 and Figure 4 of the main text}}$, we have presented comprehensive ablation studies on endogenous, exogenous, and metadata series**, including an in-depth discussion of the importance of each series type across different datasets and their impact on forecasting performance in both individual and joint training scenarios. **More than 12 ablation studies about exogenous series have been provided in $\underline{\text{Table 8,9 of Appendix}}$.**
>
>
>
> > **Q3：** Detaily, the metadata is the same from TimeLLM, the related news or reports would be preferred.
>
> In fact, compared to TimeLLM, our metadata is different in both construction principles and usage.
>
> * Different construction principles：As we mentioned $\underline{\text{in Line 48 of the main text}}$, metadata is a widely recognized concept in the field of time series analysis that aids in the understanding of data. TimeLLM leverages manually observed characteristics of data as domain knowledge. For instance, they note that "electricity consumption typically peaks at noon, accompanied by a significant increase in transformer load." **In contrast, MetaTST adheres closely to the formal definition of metadata, focusing on data source descriptions with minimal human intervention.**
>
> * Different utilize method：In TimeLLM, the textual information is fed into an LLM embedding layer concatenated with reprogrammed patch embeddings and fed into an LLM backbone. In MetaTST, **the metadata information is learned by utilizing the hidden states of LLM's last layer, and an Adapter is further introduced to align them to time series space.**
>
> We need to emphasize that MetaTST leverages the powerful representation learning capabilities of LLMs to introduce context-specific metadata into the time series prediction process, which guides the model to efficiently utilize multi-source data for efficient training in a large-scale, multi-dataset time series training scenario. This provides a meaningful approach to the construction of fundamental time series models. **The idea you mention of using news or reports to assist in time series forecasting is promising but different from the motivation of this work**. Specific differences are below:
>
> |                   | News or report         | Metadata                                                     |
> | ----------------- | ---------------------- | ------------------------------------------------------------ |
> | Motivation        | Multi-modal Model      | Time series foundation model training                        |
> | Language features | Related News or Report | Domain Knowledge、Task Detail、Sample Statistic              |
> | Data acquisition  | Difficult              | Easy ($\underline{\text{aleardy highlighted in Lines 193-197 of main text}}$) |

---

> ### Author Response · Authors · 2024-11-26
> **Request of Reviewer’s attention and feedback**
>
> Dear Reviewer wMyv,
>
> We kindly remind you that it has been around 1 week since we uploaded our rebuttal. We would be very grateful if you could confirm whether our responses adequately address your concerns. Please share any additional issues/questions you may have, and we will be happy to address them promptly.
>
> We have made every effort to address your concerns and look forward to further valuable feedback. Specifically:
>
> - **Clarify model complexity:** We provided a detailed explanation of the model complexity problem.
> - **Model generalizability:** We further clarified the criteria for selecting specific meta-data and the generalizability of our approach.
> - **Ablations on exogenous series:** We emphasized that ablation studies of exogenous series are already included in the main text.
> - **Compare with LLM4TS models in interpretability:** We highlighted the interpretability challenges of the current LLM4TS approach and made a concerted effort to analyze the effectiveness of our approach through experimental results and visual analysis.
> - **Compare with Time-LLM in processing text data:** We analyzed the differences between TimeLLM and our approach, elaborating on the distinct motivations, language features, and data acquisition for informative prediction with metadata and integration of external news or reports.
>
> All these updates have been incorporated into $\underline{\text{the revised paper}}$. We sincerely thank you for your thoughtful review of our paper.
>
> Best regards!

---

> > ### Comment · Reviewer_wMyv · 2024-12-03
> >
> > Thanks for your rebuttal. I revised the score since you answered my Q1&Q2 well. But I don't agree with your answer to Q3, different construction principles, you do have a MetaParse( ) in the paper with some fancy explanation, but it looks same as in TimeLLM.  Authors can develop further research on this topic with real-world data, like forecast temperature aligned with weather reports [1].
> > [1]: Beyond Trend and Periodicity: Guide Time Series Forecasting with Textual Cues

---

> > > ### Author Response · Authors · 2024-12-03
> > > **Thanks for your response and raising score. Weather forecasting experiments have been included Appendix G.**
> > >
> > > Dear Reviewer wMyv,
> > >
> > > We sincerely appreciate your response and raising the score. **Regarding your suggestions for further research using real-world data, such as temperature forecasting, we would like to highlight that we have already conducted a similar experiment $\underline{\text{in Appendix G}}$ to evaluate the effectiveness of our method in meteorological forecasting**.
> > >
> > > The key experimental conclusions are summarized as follows.
> > >
> > > * **Significant performance in global weather station forecasting:** MetaTST can effectively learn transferable temporal variations from global data, allowing it to adapt rapidly to the specific forecasting context of local stations. This enables efficient prediction even with smaller training samples, mitigating the cold start problem – a particularly significant challenge in meteorological forecasting.
> > > * **Visualization of metadata token in weather forecasting:** A novel representation analysis experiment clearly demonstrates that incorporating useful, context-specific metadata and leveraging LLMs' powerful representation capabilities enrich the generated representations with prior knowledge. This visual phenomenon is consistent with meteorological features and enhances the accuracy of the prediction task.
> > >
> > > For more details, please refer to $\underline{\text{Appendix G}}$. **Regarding your suggestion about temperature forecasting aligned with weather reports, we think exploring metadata-guided prediction and multimodal prediction are not mutually exclusive.** Since there are only a few hours left before the deadline of the reviewer-author discussion phase, we will explore this in future work.
> > >
> > > We believe that the informative time series forecasting approach proposed in MetaTST can be a good supplement to current research.
> > >
> > > Thank you again for your thoughtful review of our paper. Hopefully, you will reconsider your rating for a positive attitude.

---

> ### Author Response · Authors · 2024-11-27
> **We are anticipating your feedback**
>
> Dear Reviewer wMyv,
>
> Thank you once again for your valuable and constructive review, which has greatly inspired us to improve our paper further.
>
> Following your suggestions, **we discussed all your mentioned weaknesses in every detail**, including complexity analysis, metadata selection, and interpretability. In addition, **we provided an in-depth comparison between our method and TimeLLM**, highlighting differences in construction principles and utilizing method of LLMs. **We have also emphasized the distinct motivation behind MetaTST and your suggestion of using news or reports to enhance time series prediction**.
>
> We eagerly await your reply and are happy to answer any further questions. We kindly remind you that **the deadline for uploading the revised PDF is November 27th, just a few hours away.** After that, we may not have the opportunity to incorporate any further revisions based on your feedback.
>
> Sincere thanks for your dedication!
>
> Authors

---

> ### Author Response · Authors · 2024-11-29
> **Eagerly Await Your Response**
>
> Dear Reviewer wMyv,
>
> Thank you once again for your valuable and constructive review. As the author-reviewer discussion period is coming to a close, we kindly request your feedback on our rebuttal provided above. Please let us know if our responses have sufficiently addressed your concerns. If there are any remaining issues, we would be happy to engage in further discussion, as we believe that an in-depth exchange will help strengthen our paper.
>
> Best regards!
>
> Authors

---

### Official Review · Reviewer_Rm51 · 2024-11-11

**Soundness:** 2
**Presentation:** 3
**Contribution:** 2
**Rating:** 3
**Confidence:** 5

**Summary:**

This work explores the forecasting of endogenous time series using both endogenous and exogenous data, alongside metadata. The authors categorize metadata into three types and leverage a pretrained LLM for time series forecasting. The study includes short-term forecasting experiments on five electricity price datasets and long-term forecasting on seven public datasets. The proposed approach is evaluated through comparisons with naive time series models (e.g., PatchTST, iTransformer, Timer) and LLM-based models (e.g., GPT4Ts, TimeLLM).

**Strengths:**

- The proposed method achieves state-of-the-art performance.
- Leveraging meta data for time series forecasting is an interesting topic. The authors propose a method that employs pretrained large language models to effectively utilize meta data.
- The study includes extensive experimental analysis.

**Weaknesses:**

- The performance improvements over SOTA models do not appear significant. Including standard deviation metrics is crucial for a more robust evaluation. If it is unavailable, conducting statistical tests would have strengthened the claims.
- The novelty of the proposed method is limited, as similar concepts and methodologies have been presented in recent works such as TimeLLM, UniTime, PatchTST, and iTransformer. This overlap with other recent methods its novelty.
- While the paper includes numerous experiments, some key experiments supporting the main claims are absent. For instance, a quantitative comparison between models using all three types of meta data and models using only a subset of these types is not provided.
- The paper does not sufficiently analyze how the use of meta data enhances specific aspects of model predictions. A more in-depth examination is needed to identify scenarios where the use of meta data shows clear benefits and where it does not.
- The absence of supplementary code hinders reproducibility, making it challenging for others to replicate the results.
- It is unclear how meta data was used for datasets such as ETT, ECL, and weather in long-term forecasting. This lack of clarity can reduce reproducibility and is a weakness.

**Questions:**

1. TimeLLM (ICLR, 2024) also uses pretrained LLMs as encoders to input meta data into the model. What are the distinct novel aspects of MetaTST compared to this approach?
2. In Figure 8, the authors present a sensitivity analysis indicating that MetaTST's performance remains relatively stable with respect to hyperparameter changes. However, in the main text, the reported performance differences among models appear at the third decimal place. Could the authors clarify whether such minimal performance variations imply a significant difference in model effectiveness?
3. The performance improvement appears minimal. It would be beneficial to conduct a statistical analysis across multiple experimental runs to confirm whether the observed improvement is substantial and consistent.
4. UniTime (WWW, 2024) and TimesFM (ICML, 2024) are both methodologies that employ joint training across multiple domains. UniTime, in particular, incorporates domain-specific instructions to guide the model. It would be helpful to have a clear distinction regarding how MetaTST differs from these techniques. Additionally, while UniTime and TimesFM are designed for univariate input leading to univariate time series forecasting, MetaTST utilizes multivariate input for univariate time series forecasting. It would be insightful to understand whether MetaTST demonstrates superior performance in this context.
5. Many recently proposed time series forecasting methods incorporate instance normalization techniques like RevIN (ICLR, 2022). Could the authors clarify whether the proposed method also utilizes such normalization techniques? Providing clarity on this point would be helpful.
6. Most of the performance evaluations in the main text use T5 as the encoder, but only GPT-2 is used for efficiency analysis. Could the authors clarify the rationale behind this choice? A comparison using T5 would provide a fairer assessment in this context. Additionally, please specify the dataset used for the efficiency measurements. Based on the reported performances, I think the results come from joint training on the BE dataset.
7. Is Figure 1 based on actual results, or is it simply an illustrative concept? If it is just intended as a conceptual figure, please provide experimental analyses and evidence that support this concept clearly.
8. How was the proposed metadata template constructed? If the same information is provided with variations in the prompt template, does this lead to significant performance differences?
9. For long-term forecasting (Table 3), the input length is set to 96, while the output length is 192 or greater. Could the authors clarify whether this input length might be too short relative to the output length? Would increasing the input length to match or exceed the output length improve performance, potentially reducing the need for metadata? It would be insightful to see the impact on performance when the input length is extended to match the output length.
10. In the ablation study, some datasets show substantial performance differences with and without metadata, while others do not. Could the authors discuss the characteristics of these datasets and possible reasons for this variation? Given that the use of metadata is core component to the paper’s contributions, a detailed analysis would be valuable.
11. Since existing baselines do not include modules for handling metadata, could the authors provide more details on how they adapted the experimental setup for joint training? Specifically, was the setup designed to indicate that data originated from different sources? For instance, would using methods such as one-hot encoding or dataset-specific learnable encodings to identify different data sources enable the baselines to achieve similar performance improvements?

---

> ### Author Response · Authors · 2024-11-20
> **Response to Reviewer Rm51 (part 1)**
>
> We would like to sincerely thank Reviewer Rm51 for providing the detailed and insightful suggestions.
>
> ### About the Weakness
>
> > **W1：** The performance improvements over SOTA models do not appear significant. Including standard deviation metrics is crucial for a more robust evaluation. If it is unavailable, conducting statistical tests would have strengthened the claims.
>
> First, we would like to highlight that MetaTST has demonstrated superior long- and short-term forecasting capabilities in 10 competitive state-of-the-art baselines across 12 standard benchmarks including standard individual single dataset training and valuable joint multiple dataset training, which should not be underestimated.
>
> **(1) All the Baselines of Standard Single Dataset Training are under Comprehensive Hyperparameter Searching**
>
> As we stated $\underline{\text{in Lines 319-321 of the main text}}$, the results $\underline{\text{in Table 2-3 of main text}}$ for baselines are based on the **hyperparameter searching strategy**, which makes the results close to the saturation on all tasks. However, **even under this strict comparison, our MetaTST still consistently outperforms all baselines, covering both long-term and short-term predictions.** This achievement is particularly challenging and highlights the effectiveness of MetaTST.
>
> **(2) MetaTST Significantly Outperforming Other Baselines in Joint Training with Unified Hyperparameters**
>
> To provide a more intuitive understanding of the effectiveness of MetaTST, we construct individual and joint training scenarios with the same hyperparameter settings to maintain a fair and valuable performance comparison ($\underline{\text{Lines 407-409 of main text}}$).
>
> As shown $\underline{\text{in Tables 4-5 of the main text}}$, **MetaTST achieves an average MSE reduction of 9.6% across five short-term prediction datasets and 9.3% across seven long-term prediction datasets compared to the state-of-the-art model TimeXer in the joint training scenario with fair and the same hyperparameter settings.** This is very challenging and highlights the potential of MetaTST as a fundamental model for time series. Notely, we presented the average mean squared error (MSE) for four forecasting lengths (96, 192, 336, 720) in long-term forecasting tasks. However, **it is important to note that relying solely on the mean metric value sometimes gives the false impression that the improvement is not significant.**
>
> **(3) Newly add Standard Deviations in the Revised Paper**
>
> Due to relatively robust effects, most works in the field of time series usually fix random seeds instead of enumerating standard deviations. As per reviewer's request, we repeat the experiment three times and list the standard deviation of the short-term prediction task as follows. We also add new results in $\underline{\text{Section F of Appendix}}$.
>
> | MSE/MAE (Table 4 Joint Training) | BE          | DE          | FR          | NP          | PJM         |
> | -------------------------------- | ----------- | ----------- | ----------- | ----------- | ----------- |
> | 1                                | 0.322/0.232 | 0.433/0.414 | 0.334/0.191 | 0.237/0.263 | 0.081/0.183 |
> | 2                                | 0.323/0.233 | 0.440/0.416 | 0.332/0.190 | 0.233/0.263 | 0.083/0.184 |
> | 3                                | 0.318/0.234 | 0.435/0.415 | 0.329/0.193 | 0.234/0.263 | 0.087/0.186 |
> | Average                          | 0.321/0.233 | 0.436/0.415 | 0.332/0.191 | 0.235/0.263 | 0.084/0.184 |
> | Standard deviation               | 0.003/0.001 | 0.004/0.001 | 0.003/0.002 | 0.002/0.00  | 0.003/0.002 |
>
> **The confidence of MetaTST surpassing other baselines is over 99% in BE, DE, FR, NP.**

---

> > ### Author Response · Authors · 2024-11-20
> > **Response to Reviewer Rm51 (part 5)**
> >
> > ### About the Questions
> >
> > > **Q1：** TimeLLM (ICLR, 2024) also uses pretrained LLMs as encoders to input meta data into the model. What are the distinct novel aspects of MetaTST compared to this approach?
> >
> > We have analyzed TimeLLM $\underline{\text{in Section 2.1 of the main text}}$. As we stated $\underline{\text{in Line 155 of the main text}}$, Firstly, the main contribution of TimeLLM is reprogramming the input time series with text prototypes and leveraging LLMs’ reasoning capability for time series modeling. Therefore, the major distinct lies in the motivation of MetaTST, which is using metadata of time series data to provide valuable context-specific information to the prediction. **Different from taking LLM as a backbone model for time series modeling, MetaTST is a time series dominated transformer-based model training from scratch**. Secondly, there is a clear distinction between TimeLLM and MetaTST in the encoding of textual information. In TimeLLM, the textual information is fed into an LLM embedding layer concatenated with reprogrammed patch embeddings and fed into an LLM backbone. **In MetaTST, the metadata information is learned by utilizing the hidden states of LLM's last layer, and an Adapter is further introduced to align them to time series space**.
> >
> > The above analyses are summarized as  follow:
> >
> > |                      | Design Principle                             | Usage of LLM                   | Detailed Design              |
> > | -------------------- | -------------------------------------------- | ------------------------------ | ---------------------------- |
> > | TimeLLM (ICLR, 2024) | Align Time Series to Language Representation | Main body is LLM               | Use Embedding Layer for Text |
> > | MetaTST              | Encoding Metadata Text Information           | Main body is Time Series Model | Use the LLM Encoder for Text |
> >
> > Also, we need to emphasize that **MetaTST achieves significantly better predictive performance than TimeLLM while noticeably lower computational efficiency, which have shown $\underline{\text{in Figure 6 (b) of the main text}}$**. This further demonstrates that using LLMs as the backbone to align time series with language space still requires further exploration [1].
> >
> > [1] Tan, Mingtian, et al. Are language models actually useful for time series forecasting?. NeurIPS 2024.

---

> ### Author Response · Authors · 2024-11-20
> **Response to Reviewer Rm51 (part 2)**
>
> > **W2：** The novelty of the proposed method is limited, as similar concepts and methodologies have been presented in recent works such as TimeLLM, UniTime, PatchTST, and iTransformer. This overlap with other recent methods its novelty.
>
> We would like to highlight that the design principal of MetaTST is distinct from previous methods in the following aspects.
>
> **(1) New Usage of LLM.**
>
> Notbaly, **previous LLM-based approaches like TimeLLM and UniTime is leveraging pre-trained language model parameters to time series forecasting. The main body of the model is LLMs**. However, this has been questioned by [1] that **there is no evidence the LLMs successfully transfer sequence modeling abilities from text to time series**. In constrast, **MetaTST is a time series centric model that introduce metadata information formalized into natural languages with the LLMs only for metadata embedding. The main body of MetaTST is Transformer-based time series model.**
>
> [1] Tan, Mingtian, et al. Are language models actually useful for time series forecasting?. NeurIPS 2024.
>
> **(2) Defining Informative Time Series Forecasting.**
>
> To the best of our knowledge, MetaTST is the first forecasting model that incroporate both exogenous variables and metadata to perform informative time series forecasting. **The definition of "informative forecasting" rethinks the time series forecasting paradigm**, which is different from PatchTST and iTransformer that only consider the model architecture design.
>
> As highlighted $\underline{\text{in Figure 6 (b) of the main text}}$, MetaTST consistently achieves superior predictive performance compared to these mentioned methods, while maintaining significantly better or comparable efficiency, especially for LLMs as the main body model.

---

> ### Author Response · Authors · 2024-11-20
> **Response to Reviewer Rm51 (part 3)**
>
> > **W3：** While the paper includes numerous experiments, some key experiments supporting the main claims are absent. For instance, a quantitative comparison between models using all three types of meta data and models using only a subset of these types is not provided.
>
> Thank you for your insightful suggestion. As per your request, we newly conduct an ablation study on the different types of metadata. Specifically, we keep the endogenous and exogenous tokens and use different types of metadata token to validate the effectiveness of each token. As shown below, each meta token can bring benefits to the final results. We also provide additional analys $\underline{\text{in Lines 825-830}}$ and results $\underline{\text{in Table 10 of Appendix}}$.
>
> | MSE/MAE (Table 4 Joint Training) | BE          | DE          | FR          | NP          | PJM         |
> | -------------------------------- | ----------- | ----------- | ----------- | ----------- | ----------- |
> | only dataset-level               | 0.324/0.239 | 0.448/0.423 | 0.341/0.190 | 0.239/0.266 | 0.089/0.190 |
> | only task-level                  | 0.333/0.240 | 0.460/0.420 | 0.356/0.193 | 0.239/0.266 | 0.090/0.190 |
> | only sample-level                | 0.406/0.279 | 0.524/0.473 | 0.432/0.237 | 0.291/0.313 | 0.105/0.209 |
> | MetaTST（Ours）                  | 0.318/0.234 | 0.435/0.415 | 0.329/0.193 | 0.234/0.263 | 0.087/0.186 |

---

> ### Author Response · Authors · 2024-11-20
> **Response to Reviewer Rm51 (part 4)**
>
> > **W4:** The paper does not sufficiently analyze how the use of meta data enhances specific aspects of model predictions. A more in-depth examination is needed to identify scenarios where the use of meta data shows clear benefits and where it does not.
>
> **(1) New stastical results for supporting our motivation.**
>
> As shown in $\underline{\text{Figure 1 of the main text}}$, the core motivation of our method is to incorporate metadata to reduce uncertainty in future predictions. This concept is further validated in the analysis experiment for **Q7 (Please see below)**, where results indicate that **adding metadata narrows the model's predictive uncertainty interval, thereby enhancing prediction accuracy.**
>
> **(2) Emperical explanation based on our experiments.**
>
> Since this paper does not focus on theoretical analysis, we can only provide emperical explanations. Generally, if time series exhibits complex patterns and lacks stable periodicity (e.g. $\underline{\text{DE dataset in Figure 12 of Appendix}}$), the current series information may be insufficient to support future predictions. In such cases, metadata can significantly improve prediction accuracy (as shown in $\underline{\text{Table 9 of Appendix}}$). However, when the series displays clear periodicity and the prediction task is relatively simple (e.g. $\underline{\text{NP and PJM datasets in Figure 12 of Appendix}}$), metadata's contribution diminishes if sufficient information is already available.
>
> **Further, the above experimental results exactly support our motivation in "informative time series forecasting",** which is highlighted as one of our main contributions.
>
> > **W5：** The absence of supplementary code hinders reproducibility, making it challenging for others to replicate the results.
>
> Following your request, we have provide our code here: https://anonymous.4open.science/r/MetaTST_Git-CE2E
>
> > **W6：** It is unclear how meta data was used for datasets such as ETT, ECL, and weather in long-term forecasting. This lack of clarity can reduce reproducibility and is a weakness.
>
> We have provided metadata templates $\underline{\text{in Figure 2 of the main text}}$, along with practical use cases $\underline{\text{in Figure 7 of the main text and Figure 11 of the Appendix}}$. Per your request, we have re-provided comprehensive metadata examples for the long-term prediction dataset ETTm1, Weather, Traffic, and ECL $\underline{\text{in Figure 12 of Appendix}}$.

---

> ### Author Response · Authors · 2024-11-20
> **Response to Reviewer Rm51 (part 6)**
>
> > **Q2：** In Figure 8, the authors present a sensitivity analysis indicating that MetaTST's performance remains relatively stable with respect to hyperparameter changes. However, in the main text, the reported performance differences among models appear at the third decimal place. Could the authors clarify whether such minimal performance variations imply a significant difference in model effectiveness?
>
> We've responded that question $\underline{\text{in W1}}$. Here is a highlighted point: $\underline{\text{Table 2-3}}$ are with comprehensive hyperparameter searching, where the model performance is close to the saturation on these tasks. The advantage of MetaTST is more clear in the more challenging joint training tasks: $\underline{\text{Table 4-5}}$.
>
> > **Q3：** The performance improvement appears minimal. It would be beneficial to conduct a statistical analysis across multiple experimental runs to confirm whether the observed improvement is substantial and consistent.
>
> We have accounted for this issue by repeating multiple experiments to show stable average results and standard deviations $\underline{\text{in W1}}$.
>
> > **Q4：** UniTime (WWW, 2024) and TimesFM (ICML, 2024) are both methodologies that employ joint training across multiple domains. UniTime, in particular, incorporates domain-specific instructions to guide the model. It would be helpful to have a clear distinction regarding how MetaTST differs from these techniques. Additionally, while UniTime and TimesFM are designed for univariate input leading to univariate time series forecasting, MetaTST utilizes multivariate input for univariate time series forecasting. It would be insightful to understand whether MetaTST demonstrates superior performance in this context.
>
> Thank you for your insightful questions, we have included them as our related works $\underline{\text{in Lines 128-131 of the main text}}$. However, we have to pointed out that the contribution of MetaTST is orthogonal to UniTime and TimesFM.
>
> **(1) The primary objective of MetaTST is to conduct informative time series forecasting, rather than developing large-scale time series models (LTSMs)**.
>
> Previous LTSMs, such as UniTime or TimesFM, aim to use large-scale time series datasets collected from multiple domains to empower the pre-trained model to handle numerous or even unseen temporal variations. Thus, these two models are trained based on extensive and large-scale datasets, which makes their comparion to MetaTST unfair.
>
> **(2) MetaTST aims to demonstrate the importance of metadata in conventional time series models.**
>
> As we have discussed $\underline{\text{in Line 77 of the main text}}$, we contend that incorporating informative metadata can provide valuable extern information to the prediction. Considering the practical situation discussed $\underline{\text{in Line 42 of the main text}}$, we believe that this design also enables the model to distinguish between distinct forecasting scenarios and achieve accurate forecasts when training on multiple datasets. To substantiate this, we empirically verify the effectiveness of including metadata tokens through joint training on existing benchmark datasets.
>
> Our experimental results listed $\underline{\text{in Table 5 of the main text}}$ demonstrate that, **compared to conventional time series models like PatchTST and iTransformer, MetaTST achieves better performance in the multi-dataset joint training setting.** Further, As we stated in conclusion, this underscores the potential of metadata inclusion as an integral component of a large-scale time series foundation model, where **the exploration of LTSMs is left as our future work.**
>
> > **Q5：** Many recently proposed time series forecasting methods incorporate instance normalization techniques like RevIN (ICLR, 2022). Could the authors clarify whether the proposed method also utilizes such normalization techniques? Providing clarity on this point would be helpful.
>
> Yes, as shown in our code (https://anonymous.4open.science/r/MetaTST_Git-CE2E), we also adopt RevIN, which is a commonly-used pre-processing module in deep time series models. We have added the details $\underline{\text{in Lines 756-758 of the revised paper}}$.

---

> ### Author Response · Authors · 2024-11-20
> **Response to Reviewer Rm51 (part 7)**
>
> > **Q6：** Most of the performance evaluations in the main text use T5 as the encoder, but only GPT-2 is used for efficiency analysis. Could the authors clarify the rationale behind this choice? A comparison using T5 would provide a fairer assessment in this context. Additionally, please specify the dataset used for the efficiency measurements. Based on the reported performances, I think the results come from joint training on the BE dataset.
>
> **As mentioned $\underline{\text{in Line 501 of the main text}}$, the baseline models LLM4TS, which includes GPT4TS and TimeLLM, utilize a six-layer GPT-2 as their backbone.** **We would also like to emphasize that this configuration is the official setup for these two baselines**. Given that these models rely on the LLM architecture, specifically the GPT-2 model, for time series analysis, we believe that arbitrarily substituting the GPT backbone would be unjust to the baseline models and could potentially compromise their performance on the task.
>
> Regarding the dataset, our efficiency analysis is conducted on the BE dataset within a single-dataset individual training framework. We would like to clarify, as stated $\underline{\text{in Line 377 of the main text}}$, that joint training refers to the training of models using multiple datasets. The BE dataset is one of our benchmark datasets for short-term forecasting with exogenous variables. We think there may be a misunderstanding regarding the concept of joint training. Thank you for allowing us to clarify these points.
>
> > **Q7：** Is Figure 1 based on actual results, or is it simply an illustrative concept? If it is just intended as a conceptual figure, please provide experimental analyses and evidence that support this concept clearly.
>
> Figure 1 is a conceptual illustration. Following your suggestions, we designed a validation experiment using Quantile Loss, setting quantile parameters τ = 0.9 (Q90) and τ = 0.1 (Q10) under the joint training setup. To evaluate the model prediction uncertainty, we computed the difference in Quantile Loss between Q90 and Q10 on the FR test set. Typically, larger differences indicate higher prediction uncertainty, with predictions varying over a wider range; In turn, a smaller discrepancy implies greater predictive reliability.
>
> The experimental results are presented in the following table. We observe that the predictive reliability of the model gradually increases as endogenous variable information (En. + Ex.) and metadata (En. + Ex. + Metadata) are progressively introduced (0.01185 → 0.01123 → 0.01071). This conclusion is in agreement with the analysis illustrated in Figure 1. We also add these analys and results $\underline{\text{in Section D of Appendix}}$.
>
> | Test Quantile Loss  | En.     | En.+Ex. | En.+Ex.+Metadata |
> | ------------------- | ------- | ------- | ---------------- |
> | Q90                 | 0.05207 | 0.05086 | 0.05103          |
> | Q10                 | 0.04022 | 0.03963 | 0.04032          |
> | Interval difference | 0.01185 | 0.01123 | **0.01071**      |
> > **Q8：** How was the proposed metadata template constructed? If the same information is provided with variations in the prompt template, does this lead to significant performance differences?
>
> Similar to previous studies, the metadata template is manually constructed. Since MetaTST is a time series domaint forecasting model, the slight distinction in the prompt template will not lead to signficant performance difference. Following your suggestion, we attempted sentence-level shuffling within the original three prompt templates of metadata, as shown below, and found that the final prediction performance showed minimal differences.
>
> | MSE/MAE                             | BE          | DE          | FR          | NP          | PJM         |
> | ----------------------------------- | ----------- | ----------- | ----------- | ----------- | ----------- |
> | Sentence-level shuffle for metadata | 0.315/0.232 | 0.438/0.416 | 0.331/0.193 | 0.236/0.264 | 0.086/0.187 |
> | MetaTST（Ours）                     | 0.318/0.234 | 0.435/0.415 | 0.329/0.193 | 0.234/0.263 | 0.087/0.186 |

---

> ### Author Response · Authors · 2024-11-20
> **Response to Reviewer Rm51 (part 8)**
>
> > **Q9：** For long-term forecasting (Table 3), the input length is set to 96, while the output length is 192 or greater. Could the authors clarify whether this input length might be too short relative to the output length? Would increasing the input length to match or exceed the output length improve performance, potentially reducing the need for metadata? It would be insightful to see the impact on performance when the input length is extended to match the output length.
>
> In this paper, we follow **a conventional setting: long-term time series forecasting with the input length set as 96**, which is widely used in many recent works, such as iTransformer (ICLR 2024), Patchformer (ICLR 2024), CARD (ICLR 2024), and UniTime (WWW 2024). There has been a lot of work demonstrating that increasing the look-back length of time series can improve forecasting performance. **It is important to note, however, that while extending the input length enriches the temporal information of the series, metadata offers context-specific insights that come from two distinct perspectives.**
>
> Furthermore, in the context of multi-dataset joint training, as we highlighted in the main text and our previous response, metadata is crucial for enabling the model to differentiate between various datasets effectively. Following your suggestion, we extend the input length and compare the performance under the unified model hyper-parameters setting with (MetaTST) and without metadata (W/o metadata) to verify the necessity of including metadata.
>
> | MSE/MAE (input-output) | ETTh2       | ETTm2       | Traffic     | ECL         |
> | ---------------------- | ----------- | ----------- | ----------- | ----------- |
> | MetaTST (96-192)       | 0.181/0.334 | 0.112/0.248 | 0.177/0.263 | 0.307/0.397 |
> | MetaTST (192-192)      | 0.178/0.332 | 0.104/0.244 | 0.156/0.240 | 0.278/0.375 |
> | W/o Metadata (192-192) | 0.188/0.346 | 0.109/0.247 | 0.165/0.249 | 0.286/0.385 |
>
> > **Q10：** In the ablation study, some datasets show substantial performance differences with and without metadata, while others do not. Could the authors discuss the characteristics of these datasets and possible reasons for this variation? Given that the use of metadata is core component to the paper’s contributions, a detailed analysis would be valuable.
>
> Thanks for the detailed question. The difference of improvement comes from the forecastability of dataset.
>
> For intutive understanding, we have provided some showcases $\underline{\text{in Figure 12 of Appendix}}$, where we can observe that forecastability varis from different datasets. For datasets with clear periodicity such as NP and PJM, the endogenous variables can be well-predicted using information from both the endogenous and exogenous variables. In these cases, the metadata contributes limited additional information, resulting in relatively modest improvements in forecasting accuracy. In contrast, for more complex datasets like DE, where periodicity is less pronounced, the inclusion of metadata, such as sampling frequency, provides the model with crucial external periodicity information that can significantly enhance predictions. As detailed $\underline{\text{in Table 9 of the main text}}$, when metadata is utilized for the DE dataset, we observe a substantial improvement in the prediction results.

---

> ### Author Response · Authors · 2024-11-20
> **Response to Reviewer Rm51 (part 9)**
>
> > **Q11：** Since existing baselines do not include modules for handling metadata, could the authors provide more details on how they adapted the experimental setup for joint training? Specifically, was the setup designed to indicate that data originated from different sources? For instance, would using methods such as one-hot encoding or dataset-specific learnable encodings to identify different data sources enable the baselines to achieve similar performance improvements?
>
> Thank you for your thoughtful question. We have elaborated on the concept of multi-dataset joint training $\underline{\text{in Lines 377-408 of the main text}}$. The primary goal of multi-dataset joint training is to develop a unified model that can effectively learn from multiple datasets, necessitating the model's ability to handle a varying number of input variables. Although the baseline models, PatchTST, iTransformer, and TimeXer, were not specifically designed to accommodate metadata, they can still be utilized in joint training scenarios without requiring any architectural modifications. In our experiment , these models are trained on a merged dataset using the same strategy as MetaTST, though they do not account for the fact that the data is sourced from different datasets.
>
> While one-hot and learnable encodings can serve to differentiate datasets, however, as shown $\underline{\text{in lines 471-481 of the main text and in E.2 of the Appendix}}$. **LLM-based metadata embeddings allow for flexible incorporation of context-specific metadata. This enables not only distinguishing between different prediction scenarios, but also embedding valuable prior knowledge to support the prediction**, which cannot be achieved with one-hot encoding and learnable tokens. We provide the performance of dataset-level one-hot encoding and learnable encoding compared to our MetaTST as follows:
>
> |                        | BE              | DE              | FR              | NP              | PJM             |
> | ---------------------- | --------------- | --------------- | --------------- | --------------- | --------------- |
> |                        | MSE/MAE         | MSE/MAE         | MSE/MAE         | MSE/MAE         | MSE/MAE         |
> | One-hot token          | 0.323/0.236     | 0.447/0.419     | 0.337/0.195     | 0.244/0.268     | 0.089/0.189     |
> | Learned token          | 0.328/0.237     | 0.458/0.422     | 0.345/0.199     | 0.242/0.268     | 0.090/0.190     |
> | Metadata token（Ours） | **0.318/0.234** | **0.435/0.415** | **0.329/0.193** | **0.234/0.263** | **0.087/0.186** |
>
> We also add these results $\underline{\text{in Lines 862-880 and Table 12 of Appendix}}$ in our revision.

---

> ### Author Response · Authors · 2024-11-26
> **Request of Reviewer’s attention and feedback**
>
> Dear Reviewer Rm51,
>
> We kindly remind you that the rebuttal period began 2 weeks ago. Please let us know if our response has addressed your concerns. We will be happy to discuss any further issues/questions.
>
> Every effort has been made to address your concerns. Specifically:
>
> - **Compare with LLM4TS models:** We highlighted the effectiveness and novelty of our approach compared to prior LLM4TS models and also emphasized the differences between our method and these LLM4TS methods in design principles, utilization of LLMs, and specific implementation details.
> - **Standard deviations:** We added experiments that include the calculation of standard deviations over multiple runs to ensure robustness.
> - **Ablations on metadata:** We conducted ablation studies to assess the impact of single-level metadata on model performance. We include additional experiments in which one-hot encodings or learned tokens replace metadata to assess their relative effectiveness.
> - **Longer inputs:** We extended the input length with or without metadata to analyze its effect on prediction accuracy.
> - **Stastical analysis for intuitive understanding:** We analyzed the relationship between informative predictions and predictive certainty to gain a deeper insight into our model's behavior.
>
> We have included all these updates in $\underline{\text{the revised paper}}$. Thank you again for your thoughtful and thorough review of our submission.
>
> Best regards!

---

> ### Comment · Reviewer_Rm51 · 2024-11-26
>
> Thank you for your response. Since the authors uploaded their responses last week, I have thoroughly read and reviewed the authors' responses. However, I still have some remaining concerns that need further clarification.
>
> ---
> **W1.** The standard deviation values listed in Appendix F appear to have been incorrectly calculated. Specifically, they are approximately ten times smaller than those provided in the table from the OpenReview comment.
>
> ---
> **W2, Q1.** The statement regarding MetaTST as a "new usage of LLM" appears to hinge on its use of LLMs solely for embedding metadata. However, the model structure of MetaTST seems to simply **incorporate the metadata embedding from TimeLLM into PatchTST**. The structures for RevIN, Patching, Transformer blocks, and the Forecastor all appear similar to those of PatchTST. Even the **series embedding seems to be the same as the iTransformer model**.
>
> This **combination of existing methods** raises concerns about the novelty of the approach. Moreover, the practical results of this combination do not appear to demonstrate significant improvement, offering only marginal improvement. (See Q2)
>
> Also, **could the authors clarify the distinction between conventional multivariate-to-univariate time series forecasting and the use of exogenous and endogenous values in this context?**
>
> ---
> **Q2.** My earlier question remains unresolved, so I would like to clarify the main concern.
>
> The **performance differences reported in Table 4 between MetaTST and the closest baseline models are minimal**, ranging from **-0.002 to 0.031** across different datasets. The authors claimed that MetaTST introduces a significant improvement in forecasting performance.
>
> On the other hand, the **sensitivity analysis results in Fig. 8 indicate that MetaTST with different hyperparameters** exhibits a performance difference range of approximately **0.01 to 0.04. The authors insisted that this performance gap is marginal**, and MetaTST is robust to the hyperparameter change.
>
>
> The **performance gap between MetaTST and baselines seems similar to the gap between MetaTST and MetaTST with different hyperparameters**. If hyperparameter changes result in larger error differences than the gap between MetaTST and the previous baselines, **this suggests that the performance in Table 4 is also marginal**.
>
> **Additional issue**: The current sensitivity analysis (Fig. 8) uses a y-axis grid spacing of 0.2, which may be too coarse to accurately assess the error differences. It would be helpful to present this data in a table with more precise values.
>
> ---
> **Q4.** First, I am unclear about why the setting I mentioned is considered unfair. Comparing zero-shot performance on data not used for training in TimesFM seems to be an unfavorable setting for TimesFM. Based on the paper, it seems that MetaTST should show better performance in this comparison. Could the authors provide specific reasons for labeling this setting as unfair?
> Second, if the authors want to make comparisons based on training, it might be difficult to compare TimesFM because they do not provide the code for training. However, UniTime provides their official code, so a comparison is definitely possible. I suggest **evaluating not only the quantitative performance but also analyzing the impact of exogenous variables**. Since **UniTime also uses text-based domain instructions to differentiate between different datasets, such an analysis could support the claims made in this paper**.
>
> ----
> **W4-(2).** Fig. 12 does not clearly demonstrate the benefit of how metadata influences the results. To substantiate the claims, it would be helpful to compare the same examples **before and after** incorporating metadata information. No intuition has been provided about which cases were initially incorrect and how the inclusion of metadata improves the results. Additionally, the number of examples shown is insufficient to support the claim.
>
> Furthermore, **the selected examples do not convincingly showcase the influence of metadata or exogenous variables**. For instance, in cases where ground truth values drop to -1.0, the model predicts a drop to 0.5, which **seems to reflect only the seasonal pattern of the input endogenous variable rather than direct contributions from metadata or exogenous lookback**. This suggests that metadata or exogenous lookback may not have directly contributed to the accuracy of this prediction. A deeper analysis of how metadata and exogenous variables affect the model's prediction of endogenous variables is required.
>
> ---
> **Q7.** I find it somewhat unclear how the quantile loss interval added in Appendix D measures uncertainty. **A visualization, such as a prediction interval, is needed to show whether it works as motivated by Fig. 1**.
>
> Otherwise, it should be **explicitly stated in the caption of Fig. 1 that it is a conceptual illustration**, as, in its current form, it could be interpreted as an overclaim or potentially misleading.

---

> ### Author Response · Authors · 2024-11-26
> **Thanks for your response**
>
> First, we would like to thank the reviewer for the detailed response. However, it seems like that there are some important parts in our response are overlooked. Here are our clarifications.
>
> > About **W1**:
>
> We apologize for the misunderstanding caused by the incorrect transcription of the results. We have corrected the standard deviation values in $\underline{\text{Table 14 of the main text}}$ to be consistent with the OpenReview.
>
>
>
> > About **W2, Q1:**
>
> **We have already clearly highlighted the differences between our approach and TimeLLM, so we kindly point out that it is not "simply take the metadata embedding from Time-LLM."** We respectfully suggest the reviewer review our rebuttal in Q1 again. MetaTST is distinct from Time-LLM in performance, design, and efficiency.
>
> In addition, PatchTST does not use text or exogenous variables, whose main contribution lies in tokenizing time series data. **In this context, MetaTST first defines the token set essential for informative time-series forecasting and explores various methods to align metadata with the time-series model. These contributions should not be overlooked.**
>
> > About **Q2:**
>
> In our original rebuttal, we have provided a clear explanation for this question. Here are the highlights:
>
> The performance on a single dataset has already reached saturation. However, in the joint training task that is much more challenging, MetaTST can obtain a significant improvement over other methods.
>
> We hope the reviewer can recheck our model's performance in Tables 4 and 5 in the paper.
>
> > About **Q4:**
>
> As per your request, we compare UniTime and report the results of "Models Trained Across Datasets" of their paper as follows:
>
> | Avg MSE/MAE   | ETTh1           | ETTh2           | ETTm1           | ETTm2           | Weather         | Traffic     | ECL             |
> | ------------- | --------------- | --------------- | --------------- | --------------- | --------------- | ----------- | --------------- |
> | UniTime       | 0.442/0.448     | 0.378/0.403     | 0.385/0.399     | 0.293/0.334     | 0.253/0.276     | -           | 0.216/0.305     |
> | MetaTST(Ours) | **0.432/0.436** | **0.362/0.394** | **0.380/0.387** | **0.274/0.321** | **0.248/0.274** | 0.445/0.289 | **0.176/0.267** |
>
> As shown above, MetaTST still performs best.
>
> > About **W4-(2):**
>
> We have to point out that we have provided two types of results in our rebuttal of W4 to demonstrate the benefit of metadata: **Statistical Results and Emperical Explanation**.
>
> The Statistical Results can clearly explain and demonstrate that incorporating metadata can reduce prediction uncertainty and improve forecasting performance. We kindly ask you to revisit our previous rebuttal. Also, **since deep learning models are often considered "black boxes", and none of the previous works have provided a similar case study you mentioned, we believe that in terms of explanation depth, we have already surpassed previous papers.**
>
>
> > About **Q7:**
>
> There might be some misunderstanding about Quantile Loss. We respectfully suggest the reviewer check the definition of quantile loss outlined on Wikipedia: https://en.wikipedia.org/wiki/Quantile_regression, which can be used to model the possible distribution of model prediction. In Q7, we have shown that incorporating metadata would bring the 90% and 10% boundaries of the model's output closer, thereby better representing forecasting uncertainty.
>
> Thanks again for your dedication in reviewing our paper. We believe that the **informative time series forecasting** setting proposed in this paper can be a good supplement to current research. Hope you can reconsider your original rating.

---

> ### Comment · Reviewer_Rm51 · 2024-11-27
>
> I have noticed that some of my previous concerns may not have been fully addressed. To facilitate clarification, I have highlighted the key points in bold in my previous response, emphasizing the issues that remain unresolved.
>
> ---
> For example, the following points have not been addressed by the authors:
>
> In (Q2), I referenced the specific values presented in the paper and raised a concern regarding the contradiction between the two interpretations of these values provided by the authors. I would appreciate further clarification from the authors on this matter.
>
> In (W2, Q1), my intention was not merely to suggest that MetaTST is similar to other approaches, but rather to highlight that the "combination of existing work" may not offer significant novelty. As noted in my initial review, joint training is also a framework that builds on existing work.
>
> (W4-(2)) Regarding the question in Fig. 12, the intent was not to request a definition of quantile loss, but rather to emphasize that qualitative results, such as those in Fig. 1, are necessary to substantiate the authors' claims. Could the authors clarify why Fig. 1 was presented conceptually rather than as an actual prediction? I would appreciate further explanation and solid evidence confirming that the real qualitative results align with the claims made.
>
> ---
> These are some examples of the issues, though not all, that have not yet been fully addressed. I kindly request that the authors review my previous response and provide clarification.

---

> ### Author Response · Authors · 2024-11-27
> **Thanks for your patient clarification**
>
> Thanks for your detailed response and highlighting some key points. Your patience helps us a lot in fully understanding your question.
>
> > About **Q2**: contradiction between the two interpretations of these values provided by the authors
>
> Thanks for the detailed check. However, we respectfully point out that **comparison of the absolute value between different models and hyperparameters is meaningless**.
>
> As we stated in $\underline{\text{Line 323 of main text, end of Page 6}}$, "for the joint training setting, **we adjust the hyperparameters to ensure all the models have a comparable parameter**". Note that changing model hyperparameters, like the number of blocks and the size of hidden dimensions ($\underline{\text{1st and 2nd subfigures of Figure 8}}$), can seriously affect the model parameter size. Especially for the joint training, this task requires the model to handle complex data, where model capacity is essential. As for $\underline{\text{3rd subfigure of Figure 8}}$, you can find that changing the number of heads will only affect the final performance by a small margin. As for $\underline{\text{4th subfigure of Figure 8}}$, we change the look-back length, which will completely change the setting of the forecasting task.
>
> That is why the performance perturbation of MetaTST is larger in $\underline{\text{1st, 2nd and 4th subfigure of Figure 8}}$, but smaller in $\underline{\text{3rd subfigure of Figure 8}}$.
>
> Thanks for your great remark. We have rephrased the description of "model robustness" in Figure 8 as "model properties under different configurations".
>
> > About **W2, Q1**: The combination of existing work may not offer significant novelty.
>
> Thanks for this clear clarification. We want to recall our paper's title "Metadata Matters for Time Series: **Informative Forecasting with Transformers**". In this paper, our key contribution is firstly **proposing "informative time series forecasting" as a new forecasting paradigm**.
>
> According to your review, we think that the reviewer is quite familiar with the time series forecasting community. Maybe you hold the same opinion as us, that is, this community has done an exhaustive exploration of model architectures. Thus, **our paper is not trying to define a new model architecture but wants to define a more reasonable forecasting paradigm**, that is why we use "Rethinking the key factors that drive accurate time series forecasting" in $\underline{\text{Line 100, the first sentence of listed contribution}}$.
>
> Thus, we do hope the reviewer can reconsider our paper's contribution from the perspective of "defining a new paradigm", not limited by model architectures. We believe that the idea of "informative time series forecasting" can be a good supplement to current research, considering that most of the current papers are about pure time-series modality or with channel-independence training.
>
> > About **W4-(2)**:  why Fig. 1 was presented conceptually rather than as an actual prediction?
>
> We sincerely hope the reviewer can **consider the common property of deep neural networks, whose outputs cannot be precisely controlled.** Even if you are using GPT-4, the same prompt can bring quite different answers.
>
> **Although the deep models' output of a single case cannot be controlled, the statistical behavior can be analyzed.** (That is why researchers call deep models as statistical models sometimes). What we do in calculating quantile loss is to show the statistical behavior of MetaTST. From our provided results, you can clearly find that in the statistical context, adding more information can reduce the uncertainty of the future. We believe that this observation can give strong support to our conceptual figure 1.
>
> Also, since MetaTST is a deterministic model, we cannot provide the forecast interval as you requested in the prior response. Sincerely hope you can understand that, it is impossible to pursue a completely controllable behavior of deep models. The only thing we can do is just statistics.
>
> Following your suggestion, we have changed the title of Figure 1 to "A conceptual illustration" for clarity.
>
> > Potential Questions: Comparing with UniTime.
>
> Since you mentioned that the listed question is only part of your concern. We guess you may still not be satisfied with our answer for comparing with UniTime.
>
> In the last response, we have shown that MetaTST surpasses UniTime. **Fairly speaking, it is unreasonable and endless to compare the ablation variant of our methods to other models.** For example, if we remove all the designs for the metadata and exogenous series, MetaTST is just a univariate Transformer.
>
> We do hope the reviewer can consider MetaTST as a whole. **Without any part of MetaTST, we cannot achieve "informative time series forecasting." This foundational idea enables our method better than UniTime.**
>
> Thanks again for your detailed clarification. Looking forward to your reply. We will be happy to answer any further questions.
>
> Best regards.

---

> ### Author Response · Authors · 2024-12-03
> **Discussion Period Ending Soon. Eagerly Await Your Response.**
>
> Dear Reviewer Rm51,
>
> We sincerely thank you for your insightful and detailed review. Your valuable suggestions have been instrumental in improving our paper.
>
> In our last response, we provided a detailed explanation for your remained concerns, including
>
> - **Explain and revise the hyperparameter analysis**, which highlights that the relative improvement of MetaTST is not minor and should not be underestimated.
> - **Highlight our contribution of defining a new forecasting paradigm "information time series forecasting".** Also, we have to point out that overlooking our overall design and mainly arguing detailed ablations is too hair-splitting.
> - **Revise and explain why Fig.1 is a conceptual figure.** Since deep models are mainly statistical models, we believe that our statistical results of quantile loss are sufficiently convincing.
> - **Comparing the ablation variant of MetaTST with UniTime.** We would like to point out that without any part of MetaTST, we cannot do information time series forecasting (the key contribution of this paper).
>
> **With only a few hours remaining before the deadline for the reviewer-author discussion phase**, we kindly ask if our responses have adequately addressed your concerns. We look forward to your feedback and would happily answer any further questions. We also hope you will reconsider your original rating.

---

### Author Response · Authors · 2024-11-22
**Summary of Revisions**

We sincerely thank all the reviewers for their insightful reviews and valuable comments, which are instructive for improving our paper further.

this paper presents a Metadata-informed Time Series Transformer (MetaTST), which incorporates multiple levels of context-specific metadata into Transformer forecasting models to enable informative time series forecasting. **MetaTST can adaptively learn context-specific patterns across various scenarios, which is particularly effective in handling large-scale, diverse-scenario forecasting tasks, posing a potential solution for time series foundation models**. Experimentally, MetaTST achieves state-of-the-art short- or long-term forecasting performance compared to 10 advanced baselines, encompassing both single-dataset individual and multi-dataset joint training settings on 12 well-established short- and long-term benchmarks.



The reviewers generally expressed positive opinions of our paper, noting that the method proposed "**an interesting topic**," is "**clearly motivated and well-written**," demonstrates "**robust performance**," achieves "**state-of-the-art results**," and "**maintains computational efficiency**." Additionally, they highlighted its "**potential to serve as the time-series foundation model**."



The reviewers also raised insightful and constructive concerns. We have made every effort to address all the concerns by providing sufficient evidence and requested results. Here is the summary of the major revisions:

* **Clarify the effectiveness (Reviewers Rm51, wMyv):** We have emphasized that our method significantly outperforms other baselines under both comprehensive hyperparameter tuning and unified hyperparameter settings in standard single-dataset training as well as the more challenging joint-training scenarios.

* **Compare our method and previous LLMSTS models (Reviewers Rm51, wMyv):** We provide a comprehensive analysis of the differences of our method and other LLM4TS models, focusing on motivation, design principles, usage of LLMs, and detailed design.

* **Add ablation studies (Reviewers Rm51, GUpQ):** We further investigated the impact of using single-level metadata, only endogenous series, exogenous series, or metadata, as well as replacing metadata with one-hot or learned encodings. The results demonstrate that the current design of MetaTST is both reasonable and effective.

* **Analysis of Interpretability (Reviewers Rm51, wMyv):** We have explored the effectiveness of different levels of LLM-based metadata through representation visualization and conducted a validation experiment using Quantile Loss to demonstrate that the introduction of metadata enhances predictive certainty.

* **Open-sourcing and Standard Deviation (Reviewer Rm51):** We have open-sourced our code on anonymous github and repeated the experiments three times, reporting the standard deviation for the short-term prediction task.

The valuable suggestions from reviewers are very helpful for us to improve our paper. $\underline{\text{All the changes I've made in our revision (highlighted in blue)}}$. We'd be very happy to answer any further questions.

Looking forward to the reviewer's feedback.

---

### Note · Authors · 2024-12-30

**Comment:**

We express our gratitude to AC and all the reviewers for providing us with valuable constructive feedback.

Having taken into consideration the limitations of this version, we have decided to withdraw it. Nevertheless, we remain fully committed to enhancing the quality of our work and will continue working towards further improvements.

**Withdrawal Confirmation:**

I have read and agree with the venue's withdrawal policy on behalf of myself and my co-authors.